# CAMP: Coherent Alignment of Multimodal Prototypes for Explainable Complementary Learning

**Alvaro Lopez Pellicer**[† 1 2]  **Eoin M. Kenny**[1]  **Simran Lamba**[1]  **Shubham Sharma**[1]  **Plamen P Angelov**[2]  **Saumitra Mishra**[1]

## Abstract

Most multimodal learning assumes redundant views (such as image–caption pairs), yet many applications require combining complementary modalities that provide distinct evidence (such as an X-ray and medical history). We term this setting *Complementary Multimodal Classification* (CMC). In CMC, existing explainable-by-design methods often force an accuracy–interpretability trade-off because single shared similarity metrics fail under asymmetric, class-conditional evidence. We propose Coherent Alignment of Multimodal Prototypes (CAMP). CAMP enforces coherent multimodal reasoning by aligning classwise evidence via optimal transport and imposing geometric constraints to counter modality dominance and representation collapse. We provide theoretical guarantees showing that these mechanisms eliminate such degeneracies without restricting expressivity. Empirically, across 16 public CMC datasets, CAMP matches or exceeds large ($>$100M parameter) AutoML baselines with fewer than 1M trainable parameters, and when fine-tuned end-to-end it achieves state-of-the-art performance. To the best of our knowledge, this work is the first modality-agnostic prototype-learning framework designed for complementary multimodal tasks.

*Figure 1.* **CAMP Overview. (Left) CMC Task:** The system takes a multimodal input (e.g., PetFinder with Image, Text, and Tabular data) where modalities provide complementary evidence. **(Top Right) CAMP Framework:** The inputs are processed via **Adaptive Gating**, which dynamically assigns importance weights (e.g., 80% Image vs 50% Text) to fused features, followed by a query to a **Retrieved Prototype** bank in the latent space. **(Bottom Right) Prediction & Evidence:** The model outputs a prediction (e.g., Fast Adoption) supported by **Case-Based Evidence**, specifically displaying the nearest retrieved prototype to make the decision interpretable by design.

## 1. Introduction

Multimodal learning has achieved impressive results when modalities form *descriptive* paired views (e.g.,

[†]Corresponging author. Work done during an internship at JPMorganChase AI Research. [1]JPMorgan AI Research [2]School of Computing and Communications, Lancaster University, Lancaster, United Kingdom. Correspondence to: Alvaro Lopez Pellicer <a.lopezpellicer@lancaster.ac.uk>.

*Proceedings of the 43$^{rd}$ International Conference on Machine Learning*, Seoul, South Korea. PMLR 306, 2026. Copyright 2026 by the author(s).

vision↔language) and can be trained end-to-end on large-scale aligned corpora (Radford et al., 2021). However, many deployed decision systems operate in a *complementary* regime where each modality contributes distinct, label-relevant evidence. For example, an image of a patient's x-ray may be complemented by a textual description of their medical history, both offering distinct and important information for task performance, as opposed to a textual description which simply describes the image. We term such a setting *Complementary Multimodal Classification* (CMC), which is an area relatively underexplored in the

community.

CMC is common in high-stakes applications (e.g., healthcare, fraud, online safety), where stakeholders often require not only accuracy but also *auditable evidence* for a decision. However, prevailing CMC systems are typically large multimodal transformers or AutoML ensembles (Shi et al., 2021; Tang et al., 2024b), which can be expensive to fine-tune and rely on post-hoc explanation tools not faithful to the model's internal decision logic (Rudin, 2019). Prototype-based models offer predictions based on case-based reasoning that are explicit functions of similarity to a small set of learned representatives and thus expose decision evidence by design (Snell et al., 2017; Chen et al., 2019). However, in CMC, a single shared notion of similarity can be brittle, because complementary modalities can contribute asymmetric class-conditional evidence, which can yield class-impure or redundant prototypes, collapse, dead prototypes, and drift under rare/outlier cases.

To solve these problems, we introduce CAMP (see Figure 1), a modality-agnostic prototype learning framework for CMC. CAMP maps each available modality into a shared space, fuses them with lightweight instance-wise gates, and predicts using a compact class-partitioned prototype bank with a top-$r$ retrieval head (Sec. 3.2–3.3). During training, we align class-wise modality evidence via Sinkhorn-regularized optimal transport and regularize geometry, allocation, and anchor robustness to avoid collapse, dominance, and outlier drift (Sec. 3.4–4). This yields a multimodal prototype model that uses complementary evidence from the available modalities while retaining faithful, case-based explanations by design.

To summarize, our contributions are:

- We formalize CMC as classification from arbitrary observed subsets of complementary modalities and identify the prototype failure modes that arise in this setting (Sec. 3.1).

- We propose CAMP, a modality-agnostic prototype architecture that can be trained end-to-end or as a sub-1M trainable head on frozen encoders (Sec. 3.2).

- We introduce a loss-by-loss objective that aligns class-wise prototype usage across modalities while controlling collapse, dead units, class impurity, and prototype drift (Sec. 3.4).

- We give a theory analysis that composes classical margin, allocation, anchoring, and OT arguments into a CMC-specific guarantee for coherent prototype alignment (Sec. 4; Appendix B).

We evaluate CAMP on 16 public CMC datasets in frozen and end-to-end regimes, including detailed ablations, mechanistic prototype diagnostics, missing-modality stress tests, and a case study of CAMP's multimodal explanations with faithfulness guarantees (Sec. 5–6.2; Appendix D–G).

## 2. Related Work

Multimodal learning studies how to represent, align, and fuse heterogeneous modalities (vision, language, tabular, audio), with early work and surveys formalizing core fusion challenges (Ngiam et al., 2011; Baltrušaitis et al., 2019). Much recent progress targets *descriptive* paired settings (e.g., image–text pretraining and retrieval), where instance-level correspondence is explicitly optimized (Radford et al., 2021). We focus on complementary multimodal classification (CMC), where each instance is a data-table row with structured covariates and optional unstructured signals (text/images) and modalities are complementary rather than mutually descriptive (Sec. 1) (Shi et al., 2021; Tang et al., 2024b).

The current works builds upon prototype neural networks. Perhaps the first introduction of such architectures was by Li et al. (2018), which prompted followup works in e.g. text (Ming et al., 2019), prototype "parts" (Chen et al., 2019), and deep reinforcement learning (Kenny et al., 2023). This area of research represents the most popular form of interpretable-by-design neural network architecture (Ma et al., 2024), but to the best of our knowledge their usage in multimodal tasks has not been explored before the present work.

Prototype models predict by similarity to a small set of learned representatives, enabling case-based reasoning and intrinsic interpretability (Snell et al., 2017; Chen et al., 2019; Ming et al., 2019; Nauta et al., 2021; Rymarczyk et al., 2022). Training often combines a supervised term with auxiliary objectives (e.g., pull/cluster and push/separation) and optional projection steps to anchor prototypes to observed examples (Chen et al., 2019). However, learned prototypes can exhibit *collapse* (multiple prototypes converging), *dead* prototypes that are never selected/updated, and *redundancy* across classes, motivating balanced assignment and selection mechanisms (Caron et al., 2020; Zheng & Vedaldi, 2023; Nauta et al., 2021; Rymarczyk et al., 2022). Prototypes can also *drift* under non-stationarity/concept drift (Shao et al., 2014) and may under-represent *rare outliers* or long-tail classes without explicit balancing (Wei et al., 2022). To overcome these issues, we enforce class-wise entropic optimal transport alignment of modality-specific prototype-usage distributions to reduce cross-modal disagreement, and add geometry and allocation constraints to prevent collapse, redundancy, and dead prototypes, yielding coherent multimodal evidence while preserving interpretability.

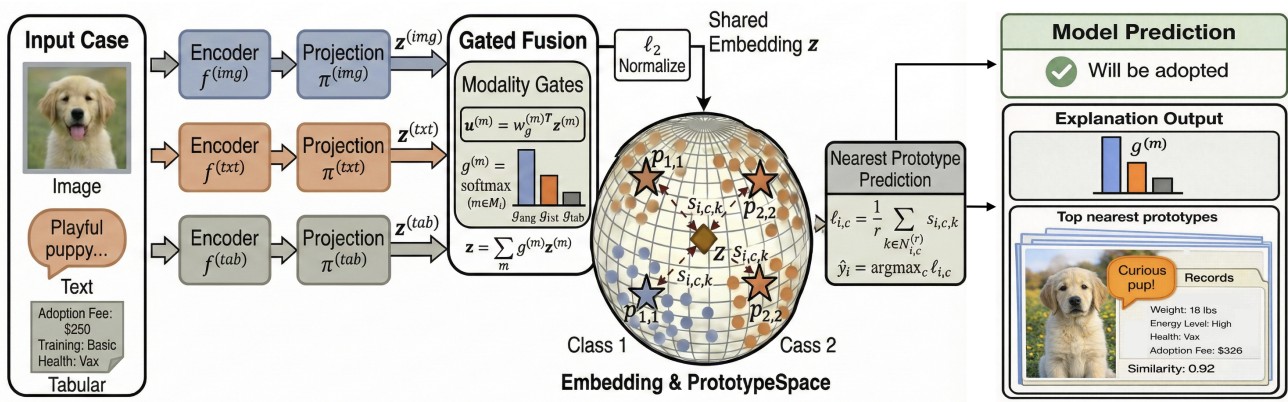

*Figure 2.* **Overview of the CAMP Architecture.** The model takes a multi-modal input case consisting of Image, Text, and Tabular data, processing them through modality-specific encoders $f^{(m)}$ and projections $\pi^{(m)}$ to produce embeddings $z^{(m)}$. A **Gated Fusion** mechanism dynamically computes attention weights $g^{(m)}$ to aggregate these inputs into a normalized shared embedding $z$. In the **Embedding & Prototype Space**, this case embedding is compared against learned class prototypes (stars, $p_{c,k}$) residing on a hypersphere. The system yields a **Model Prediction** based on prototype similarity and generates an **Explanation Output** displaying the computed modality gates and the top nearest prototypes (retrieved historical cases).

Perhaps the most related work to ours is ProtoMedX (Pellicer et al., 2025), a recent prototype-based approach for multimodal medical classification that explains predictions via nearest neighbors. It combines modality evidence via a weighted sum of prototype distances, but it targets a fixed two-modality medical setting and does not explicitly align per-class prototype *usage distributions* across modalities or support arbitrary modality subsets. We include ProtoMedX in our empirical results for comparison.

## 3. Method

We first formalize CMC (Sec. 3.1), then describe CAMP's architecture (Sec. 3.2) and training objective (Sec. 3.4).

### 3.1. Problem setting

Let $\mathcal{M} = \{1, \dots, M\}$ be a modality universe. Instances are tuples $(\{x_i^{(m)}\}_{m \in \mathcal{M}_i}, y_i)$, where $\mathcal{M}_i \subseteq \mathcal{M}$ denotes the present modalities and $y_i \in \{1, \dots, C\}$ the class label.

We term this setting **Complementary Multimodal Classification** (CMC): tasks where inputs comprise arbitrary subsets of loosely coupled modalities (e.g., tabular, text, image). Unlike the mutually descriptive pairs in vision–language pretraining, these modalities provide non-redundant signals. We formalize this complementarity by noting that for at least one modality $m$, the conditional mutual information is positive: $I(Y; X^{(m)} \mid X^{(\mathcal{M} \setminus \{m\})}) > 0$.

This distinguishes CMC from *descriptive* paired multimodality (Sec. 2), where instance-level alignment is inherently well-posed (Radford et al., 2021).

In descriptive regimes, contrastive objectives succeed be-

cause inter-modality mutual information $I(X^{(A)}; X^{(B)})$ is intrinsically high. Conversely, in CMC, modalities may occupy entirely distinct semantic spaces (e.g., a categorical patient history versus a high-dimensional continuous scan). This means inter-modality mutual information can be exceedingly low even while joint mutual information with the target label, $I(Y; X^{(A)}, X^{(B)})$, remains high. Consequently, blindly applying standard instance-level alignment to CMC data often forces an artificial bottleneck, erasing necessary complementary signals.

Our goal is to learn (i) a shared case embedding $z_i \in \mathbb{R}^d$ and (ii) a prototype-grounded score vector $\ell(z_i) \in \mathbb{R}^C$ that can be evaluated for any subset $\mathcal{M}_i$ (*modality-agnostic inference*). At inference, the gates and prototype query are computed only over the observed set, meaning no modality imputation is required (yielding strong missing-modality robustness, see Appendix F.6). While modality-agnostic, our experiments instantiate $\mathcal{M} = \{\text{tab}, \text{txt}, \text{img}\}$ following AutoML-style CMC benchmarks (Shi et al., 2021; Tang et al., 2024b).

We target **discrete** outcomes (binary/multiclass/ordinal classification), matching standard CMC benchmarks (Shi et al., 2021; Tang et al., 2024b); continuous regression extensions remain future work.

### 3.2. Encoders, shared space, and gated fusion

As illustrated in Figure 2, CAMP processes a heterogeneous input tuple via distinct encoding branches. Each modality $m \in \mathcal{M}_i$ uses an encoder $f^{(m)}$ and projection $\pi^{(m)}$ to map

raw inputs to a common latent dimension:

$$h_i^{(m)} = f^{(m)}\big(x_i^{(m)}\big) \in \mathbb{R}^{d_m}, \tag{1}$$

$$z_i^{(m)} = \pi^{(m)}\big(h_i^{(m)}\big) \in \mathbb{R}^{d}. \tag{2}$$

To synthesize these signals, the **Gated Fusion** module (Eq. 4) calculates a scalar importance gate $g_i^{(m)}$ per branch, aggregating them into a single case embedding $z_i = \sum_m g_i^{(m)} z_i^{(m)}$.

This embedding is projected onto the unit hypersphere (**Embedding & Prototype Space**) and compared against the learned prototype bank $\{p_{c,k}\}$. The final prediction $\hat{y}_i$ is derived from the average similarity to the $r$ nearest prototypes (Eq. 5), ensuring the decision is locally supported by representative regions of the feature space.

**Choosing late fusion.** We use *late fusion* (unimodal encoders preceding a lightweight shared-space fusion module), a robust default for multimodal tables (Shi et al., 2021; Tang et al., 2024b). Encoder choices follow these best practices (Appendix E).

**Choosing gated fusion.** To represent a *case* with partial observations, we fuse embeddings via per-instance gates estimating each modality's relevance. Unlike dense cross-attention mechanisms that deeply entangle feature spaces and obscure learned signals, our gating preserves independent channel integrity. Let $u_i^{(m)} = w_g^{(m)\top} z_i^{(m)}$ be a learned scalar relevance score. We compute

$$g_i^{(m)} = \text{softmax}_{m \in \mathcal{M}_i}\big(u_i^{(m)}\big), \tag{3}$$

$$z_i = \sum_{m \in \mathcal{M}_i} g_i^{(m)} z_i^{(m)}. \tag{4}$$

The gates form a simplex over observed modalities, so they can be inspected directly as modality reliance. This linear combination forces the network to dynamically route evidence in a transparent manner. This structure handles missing deployment data natively; if a modality is unavailable, the softmax seamlessly redistributes evidential mass across surviving modalities without complex imputation heuristics. We keep gated fusion to preserve interpretability, and it is empirically strongest: it wins on 15/16 datasets vs. concat/attention fusion (Appendix F, Table 18).

To ground the abstract latent space in human-understandable concepts, we adopt the projection strategy common to standard prototypical models. Although $p_{c,k}$ are learned as continuous vectors, the **Explanation Output** (Figure 2) visualizes each by projecting it onto its nearest training instance (e.g., the "Curious pup" record). This justifies predictions by surfacing concrete historical cases semantically similar to the input query.

### 3.3. Prototype bank and prototype head

We learn $K$ prototypes per class, $\{p_{c,k}\}_{k=1}^{K}$ for each $c \in \{1, \ldots, C\}$, and write $p_{c,k} \in \mathbb{R}^{d}$. Let $S(z, p)$ be a bounded similarity that is monotone with distance and Lipschitz in $z$ (e.g., negative squared distance on the unit sphere). We compute per-prototype similarities $s_{i,c,k} = S(z_i, p_{c,k})$.

**Classification head.** We use a simple *r-NN prototype head*: each class logit is the mean similarity to the $r$ most similar prototypes of that class. Let $\mathcal{N}_{i,c}^{(r)} = \text{Top}_r\big(\{s_{i,c,k}\}_{k=1}^{K}\big)$ be the indices of the $r$ largest similarities for class $c$. We define

$$\ell_{i,c} = \frac{1}{r} \sum_{k \in \mathcal{N}_{i,c}^{(r)}} s_{i,c,k}. \tag{5}$$

The prediction is $\hat{y}_i = \arg\max_c \ell_{i,c}$. At inference, we return the modality weights $\{g_i^{(m)}\}$ and the retrieved prototypes in $\mathcal{N}_{i,\hat{y}_i}^{(r)}$ as a case-based explanation. Inference formats and formal explanation faithfulness guarantees are fully defined in Appendices E and G. Setting $r=1$ recovers the nearest-prototype rule used in our theory.

### 3.4. Training objective

We start from Cross-Entropy (CE), the standard classification loss. From here onwards defined as the prediction/task loss for our method, symbolized as $\mathcal{L}_{\text{pred}}$.

Prototype learning traditionally augments CE with pull/push and supervised diversity terms (Chen et al., 2019; Khosla et al., 2020). In CMC, these alone do not prevent redundancy, dead prototypes, cross-modal disagreement, or drift. CAMP addresses these via four complementary loss blocks.

**(1) Multimodal alignment via class-wise entropic OT.** To reduce cross-modal disagreement and modality dominance *without* instance-level alignment, we match modalities at the level of *class-conditioned prototype usage* using entropic OT (Cuturi, 2013; Peyré & Cuturi, 2019) (Sec. 4 and Appendix B.3). OT resolves within-class prototype permutation ambiguity by providing a differentiable coupling over prototypes.

For each modality $m$ we compute modality-specific similarities $s_{i,c,k}^{(m)} = S(z_i^{(m)}, p_{c,k})$ and assignments $\alpha_{i,c,k}^{(m)} = \text{softmax}_k(s_{i,c,k}^{(m)}/\tau_a)$. For each class $c$, define a discrete measure on that class's prototypes:

$$\nu_c^{(m)} = \sum_{k=1}^{K} w_{c,k}^{(m)} \delta_{p_{c,k}}, \qquad w_{c,k}^{(m)} = \frac{1}{|\mathcal{I}_c|} \sum_{i \in \mathcal{I}_c} \alpha_{i,c,k}^{(m)}, \tag{6}$$

where $\mathcal{I}_c = \{i : y_i = c\}$. Note that we align the usage distributions ($\nu$), not specific instances. This allows modalities to contribute complementary evidence while agreeing on the

aggregate class definition. We penalize pairwise entropic OT distances between these measures,

$$\mathcal{L}_{\text{mm}} = \frac{1}{C|\mathcal{P}|} \sum_{c=1}^{C} \sum_{(m,m')\in\mathcal{P}} \quad (7)$$
$$W_\varepsilon\big(\nu_c^{(m)}, \nu_c^{(m')}\big).$$

where $\mathcal{P}$ is the set of observed modality pairs in a batch, $W_\varepsilon$ is the Sinkhorn-regularized OT cost (Cuturi, 2013; Peyré & Cuturi, 2019) with ground cost $c(p, p') = \|p - p'\|_2^2$. We compute $W_\varepsilon$ via $L$ Sinkhorn iterations with entropic regularization $\varepsilon$. This term is training-time only and adds no inference overhead. Intuitively, it encourages modalities to agree on *which prototypes* represent each class in aggregate while preserving complementary per-instance evidence.

**(2) Prototype geometry with class-conditioned diversity.** While standard prototype networks rely on instance-to-prototype pull and push terms (Chen et al., 2019), this block adapts to the CMC regime by introducing explicit class-conditioned prototype–prototype margins. Specifically, $\mathcal{L}_{\text{pp-intra}}$ forces within-class prototypes to maintain minimum separation ($\rho_{\text{in}}$) preventing redundant collapse, while $\mathcal{L}_{\text{pp-inter}}$ pushes cross-class prototypes apart ($\rho_{\text{out}}$). This guarantees diverse, non-overlapping spatial anchors that distinct complementary modalities can reliably align to (see Theorem B.1 in Appendix B.1). We combine these into a single block:

$$\mathcal{L}_{\text{pull}} = \frac{1}{N} \sum_{i=1}^{N} \Big[ \rho_+ - \max_k s_{i,y_i,k} \Big]_+,$$

$$\mathcal{L}_{\text{rep}} = \frac{1}{N} \sum_{i=1}^{N} \Big[ \max_{c\neq y_i,\, k} s_{i,c,k} - \rho_- \Big]_+,$$

$$\mathcal{L}_{\text{pp-intra}} = \frac{1}{CK(K-1)} \sum_{c=1}^{C} \sum_{k\neq k'} [S(p_{c,k}, p_{c,k'}) - \rho_{\text{in}}]_+,$$

$$\mathcal{L}_{\text{pp-inter}} = \frac{1}{C(C-1)K^2} \sum_{c\neq c'} \sum_{k,k'} [S(p_{c,k}, p_{c',k'}) - \rho_{\text{out}}]_+,$$

$$\mathcal{L}_{\text{geo}} = \mathcal{L}_{\text{pull}} + \mathcal{L}_{\text{rep}} + \lambda_{\text{pp}}\big(\mathcal{L}_{\text{pp-intra}} + \mathcal{L}_{\text{pp-inter}}\big),$$
$$(8)$$

with margins $\rho_+ > \rho_-$ and $\rho_{\text{out}} < \rho_{\text{in}}$.

**(3) Allocation: coverage and purity.** This block prevents *dead* (unused) prototypes and class-impure prototypes that dilute explanations and degrade robustness (Chen et al., 2019). Let $\alpha_{i,k}$ be a soft assignment of sample $i$ to prototype $k$ (over all $CK$ prototypes), e.g., $\alpha_{i,k} = \text{softmax}_k(s_{i,k}/\tau_a)$. We define average usage $\bar{\alpha}_k = \frac{1}{N} \sum_i \alpha_{i,k}$. Coverage encourages all prototypes to be used:

$$\mathcal{L}_{\text{cov}} = D_{\text{KL}}\big(\bar{\boldsymbol{\alpha}} \,\|\, \mathbf{u}\big), \qquad \mathbf{u} = \frac{1}{CK}\mathbf{1}. \quad (9)$$

Purity encourages each prototype to specialize to one class. Let $q_k(c) \propto \sum_{i=1}^{N} \alpha_{i,k} \mathbf{1}\{y_i = c\}$ be the induced class distribution for prototype $k$. We minimize entropy:

$$\mathcal{L}_{\text{pure}} = \frac{1}{CK} \sum_{k=1}^{CK} H\big(q_k\big). \quad (10)$$

The allocation block is

$$\mathcal{L}_{\text{alloc}} = \mathcal{L}_{\text{cov}} + \lambda_{\text{pure}}\mathcal{L}_{\text{pure}}. \quad (11)$$

These allocation constraints theoretically guarantee full bank utilization and strict class specialization (Theorems B.2 and B.3, Appendix B.2), preventing a common bottleneck where a redundant minority of units monopolizes classification and dilutes explainability.

**(4) Anchor-based robustness for unrepresented cases.** Case-based prototype models can behave erratically for rare or unrepresented cases far from every same-class prototype, causing unstable gradients and drift (Snell et al., 2017; Chen et al., 2019). We maintain class anchors $\{a_c\}_{c=1}^{C}$ (learnable parameters softly tied to prototype means) and apply an additional stabilizer to samples whose best same-class prototype similarity is below a threshold. Let $s_i^\star = \max_k s_{i,y_i,k}$ and define the outlier indicator $\omega_i = \mathbb{1}\{s_i^\star < \rho_{\text{det}}\}$. We use

$$\mathcal{L}_{\text{anc}} = \frac{1}{N} \sum_{i=1}^{N} \omega_i \Bigg[ \text{ReLU}\big(\rho_a - S(z_i, a_{y_i})\big)$$
$$+ \frac{1}{CK} \sum_{c=1}^{C} \sum_{k=1}^{K} \text{ReLU}\big(S(z_i, p_{c,k}) - \rho_{\text{ol}}\big) \Bigg], \quad (12)$$

and softly tie anchors to the prototype configuration with $\mathcal{L}_{\text{anc-cons}} = \frac{1}{C} \sum_c \big\| a_c - \frac{1}{K} \sum_k p_{c,k} \big\|_2^2$. We denote the full anchor block by $\mathcal{L}_{\text{anchor}} = \mathcal{L}_{\text{anc}} + \lambda_{\text{cons}}\mathcal{L}_{\text{anc-cons}}$. Anchors act as a conservative class-level fallback, provably bounding outlier gradient influence and target drift (Theorem B.6, Appendix B.4).

**Final objective.** The full training loss is

$$\mathcal{L} = \mathcal{L}_{\text{pred}} + \lambda_{\text{mm}}\mathcal{L}_{\text{mm}} + \lambda_{\text{geo}}\mathcal{L}_{\text{geo}}$$
$$+ \lambda_{\text{alloc}}\mathcal{L}_{\text{alloc}} + \lambda_{\text{anchor}}\mathcal{L}_{\text{anchor}}. \quad (13)$$

Training minimizes Eq. (13) with standard minibatch optimization; auxiliary losses are linearly ramped and computed only on observed modalities/pairs in each minibatch. See Appendix E (Algorithm E.2) for full training pseudocode and hyperparameter selection. The frozen-encoder regime is obtained by holding $\{f^{(m)}\}$ fixed and optimizing only the projections, gates, prototypes, and anchors.

# 4. Coherent Alignment of Multimodal Prototypes

CMC differs from paired-view multimodal learning: modalities can carry *distinct* label evidence, so unconstrained fusion can achieve low training loss while still producing *multimodal degeneracy* at test time. Biased toward simpler predictive paths, unconstrained networks often overfit to a dominant modality, discarding the sparse, complementary signals critical for edge cases.

We focus on four prototype failure modes amplified by this CMC degeneracy: **(i) collapse** (redundant representations stripping discriminative capacity), **(ii) dominance and disagreement** (modalities monopolizing latent geometry or clustering classes inconsistently), **(iii) dead or impure prototypes** (under-utilized banks or weak class specialization diluting explanations), and **(iv) drift** (gradient instability driven by rare multimodal outliers).

Our theory provides a structural composition rather than a new isolated primitive. We combine Optimal Transport (OT) duality (Edwards, 2011; Villani, 2009; Peyré & Cuturi, 2019), margin bounds (Bartlett & Mendelson, 2002; Mohri et al., 2018), Pinsker-style allocation control (Pinsker, 1964; Cover & Thomas, 2006), and geometric anchor stability to show their *joint application* forms a well-posed regularizer for CMC. Since individual constraints remain vulnerable to degeneracies (e.g., distributional alignment fails without separated support), coupling geometry, allocation, anchors, and OT is necessary to ensure a stable, modality-agnostic prototype head.

The formal statements below characterize the prototype head and frozen-encoder regime; Sec. 6 shows this structural bias remains effective when encoders are fine-tuned.

## 4.1. Minimizing Alignment Bias via Coherent Transport

Instance-level multimodal alignment ($z^{(A)} \approx z^{(B)}$) can introduce an *alignment bias* that erases complementary information (Dufumier et al., 2025). CAMP instead aligns *class-wise prototype-usage distributions*. Thus modalities become **Coherent** at the aggregate class level while remaining **Complementary** at the instance level.

**Theorem 4.1** (OT alignment enforces coherent evidence). *Fix class c. For any semantic feature $\varphi$ (1-Lipschitz), the cross-modal disagreement is bounded by the transport cost:*

$$\left| \mathbb{E}_{\nu_c^{(m)}}[\varphi] - \mathbb{E}_{\nu_c^{(m')}}[\varphi] \right| \leq W_2\big(\nu_c^{(m)}, \nu_c^{(m')}\big). \qquad (14)$$

*(Full proof in Appendix C.1.)*

While theorem 4.1 relies on classical Kantorovich–Rubinstein duality, the CMC-specific point is that the OT bound is meaningful only on well-behaved shared support:

$\mathcal{L}_{\text{geo}}$ prevents trivial collapse by enforcing within-class separation (Theorem B.1); $\mathcal{L}_{\text{alloc}}$ keeps that support active and class-specialized, avoiding dead or redundant units (Theorems B.2 and B.3); and $\mathcal{L}_{\text{anchor}}$ bounds target drift so alignment tracks persistent evidence rather than transient batch artifacts (Theorem B.6).

Under these coupled conditions, small OT cost gives strict agreement of prototype-usage histograms across modalities (**Lemma B.5**) and stable inference over arbitrary observed modality subsets (**Proposition C.1**).

# 5. Experiments

We evaluate CAMP on 16 public CMC datasets: Tab+Text (6), Image+Text (4), Image+Tab (3), and Image+Text+Tab (3), following the standardized suite, splits, and preprocessing from the AutoGluon multimodal benchmarks (Shi et al., 2021; Tang et al., 2024b). We exclude AEP because it does not meet our complementarity definition and three Tab+Text datasets due to broken access; details and licenses are in Appendix D.

## 5.1. Competitors and Baselines

We compare with: (1) **High-performance references**: AutoGluon (AG) multimodal ensembles (Tang et al., 2024b) and the Tabular-Text Transformer (TTT) (Bonnier, 2024); (2) **Lightweight frozen heads**: linear probe (LP), CE-only MLP, and XGBoost; and (3) **Prototype baselines**: PROTO-RAW, PROTO-CLASSIC, and ProtoMedX (Pellicer et al., 2025).

## 5.2. Evaluation Protocol

To isolate objective structure from backbone capacity, all frozen-head methods (CAMP-F and baselines) use identical Swin-L and DeBERTa-v3 embeddings. CAMP-F trains only a compact prototype head (~660K parameters); CAMP-E fine-tunes end-to-end. We report mean±std over 5 seeds with Optuna tuning (50 trials/dataset) and summarize by unweighted average (Avg.) and Mean Reciprocal Rank (MRR).

# 6. Results

## 6.1. Quantitative Results

Table 1 isolates the impact of CAMP's structural objective from encoder capacity confounds. In the **frozen-encoder regime** (CAMP-F), training solely a sub-1M parameter prototype head with fixed Swin-L and DeBERTa-v3 embeddings achieves a suite average of 0.814 and a Mean Reciprocal Rank (MRR) of 0.448. This significantly outperforms an identical-feature cross-entropy MLP (0.767 average, 0.174

*Table 1.* **Performance across 16 multimodal classification benchmarks.** Mean $\pm$ standard deviation over five seeds. **Bold** marks the best result; $^\star$ indicates CAMP-F matches or exceeds AutoGluon, and underline is within 1%. AvG/MRR report the suite mean/mean reciprocal rank. Abbreviations: XGB (XGBoost); LP (Linear Probe); PR (Proto-Raw); PB (Proto-Base); PMX (ProtoMedX); MLP (Frozen backbone + MLP); TTT (Tabular-Text Transformer); AG (AutoGluon); CAMP-F/CAMP-E (Ours, frozen/end-to-end).

| Dataset | Metric | LIGHTWEIGHT | | PROTOTYPE BASELINES | | | BLACK-BOX | | | Ours (CAMP) | |
|---|---|---|---|---|---|---|---|---|---|---|---|
| | | XGB | LP | PR | PB | PMX | MLP | TTT | AG | **Frozen** | **End-to-End** |
| *Tabular + Text* | | | | | | | | | | | |
| IMDB | AUC | $0.806_{\pm0.001}$ | $0.801_{\pm0.002}$ | $0.795_{\pm0.005}$ | $0.813_{\pm0.003}$ | $0.843_{\pm0.003}$ | $0.839_{\pm0.004}$ | $0.886_{\pm0.004}$ | $0.878_{\pm0.005}$ | $0.899_{\pm0.001}^\star$ | $0.911_{\pm0.002}$ |
| FakeJob | AUC | $0.948_{\pm0.001}$ | $0.915_{\pm0.003}$ | $0.920_{\pm0.007}$ | $0.942_{\pm0.005}$ | $0.970_{\pm0.005}$ | $0.951_{\pm0.007}$ | $0.977_{\pm0.003}$ | $0.974_{\pm0.015}$ | $0.992_{\pm0.001}^\star$ | $0.997_{\pm0.004}$ |
| Kick | AUC | $0.763_{\pm0.001}$ | $0.745_{\pm0.002}$ | $0.710_{\pm0.006}$ | $0.736_{\pm0.004}$ | $0.766_{\pm0.004}$ | $0.763_{\pm0.006}$ | $0.767_{\pm0.005}$ | $0.799_{\pm0.010}$ | $0.811_{\pm0.001}^\star$ | $0.844_{\pm0.002}$ |
| Jigsaw | AUC | $0.941_{\pm0.001}$ | $0.912_{\pm0.002}$ | $0.905_{\pm0.006}$ | $0.921_{\pm0.004}$ | $0.954_{\pm0.004}$ | $0.955_{\pm0.005}$ | $0.945_{\pm0.003}$ | $0.967_{\pm0.008}$ | $0.974_{\pm0.001}^\star$ | $0.986_{\pm0.002}$ |
| Wine | Acc | $0.802_{\pm0.001}$ | $0.737_{\pm0.003}$ | $0.780_{\pm0.008}$ | $0.795_{\pm0.005}$ | $0.818_{\pm0.005}$ | $0.809_{\pm0.007}$ | $0.821_{\pm0.006}$ | $0.842_{\pm0.012}$ | $0.844_{\pm0.003}^\star$ | $0.872_{\pm0.004}$ |
| Airbnb | Acc | $0.445_{\pm0.001}$ | $0.361_{\pm0.004}$ | $0.355_{\pm0.010}$ | $0.378_{\pm0.006}$ | $0.401_{\pm0.006}$ | $0.395_{\pm0.008}$ | $0.383_{\pm0.011}$ | $0.498_{\pm0.017}$ | $\underline{0.488}_{\pm0.004}$ | $0.529_{\pm0.012}$ |
| *Image + Text* | | | | | | | | | | | |
| PTech | AUC | $0.726_{\pm0.002}$ | $0.688_{\pm0.005}$ | $0.690_{\pm0.012}$ | $0.703_{\pm0.008}$ | $0.716_{\pm0.008}$ | $0.720_{\pm0.011}$ | $0.718_{\pm0.008}$ | $0.732_{\pm0.032}$ | $0.744_{\pm0.004}^\star$ | $0.800_{\pm0.006}$ |
| Food101 | Acc | $0.896_{\pm0.001}$ | $0.913_{\pm0.002}$ | $0.885_{\pm0.005}$ | $0.903_{\pm0.003}$ | $0.925_{\pm0.003}$ | $0.916_{\pm0.004}$ | $0.892_{\pm0.007}$ | $0.949_{\pm0.007}$ | $0.950_{\pm0.002}^\star$ | $0.971_{\pm0.005}$ |
| Fakeddit | Acc | $0.757_{\pm0.001}$ | $0.850_{\pm0.003}$ | $0.820_{\pm0.008}$ | $0.841_{\pm0.005}$ | $0.891_{\pm0.005}$ | $0.883_{\pm0.007}$ | $0.906_{\pm0.007}$ | $0.914_{\pm0.014}$ | $0.927_{\pm0.003}^\star$ | $0.954_{\pm0.008}$ |
| Memotion | Acc | $0.565_{\pm0.001}$ | $0.576_{\pm0.005}$ | $0.565_{\pm0.012}$ | $0.579_{\pm0.008}$ | $0.615_{\pm0.008}$ | $0.593_{\pm0.012}$ | $0.614_{\pm0.010}$ | $0.655_{\pm0.032}$ | $\underline{0.644}_{\pm0.007}$ | $0.666_{\pm0.009}$ |
| *Image + Tabular* | | | | | | | | | | | |
| CCD | AUC | $0.878_{\pm0.001}$ | $0.826_{\pm0.004}$ | $0.850_{\pm0.010}$ | $0.896_{\pm0.006}$ | $0.913_{\pm0.006}$ | $0.907_{\pm0.008}$ | $0.885_{\pm0.005}$ | $0.950_{\pm0.015}$ | $\underline{0.937}_{\pm0.005}$ | $0.972_{\pm0.005}$ |
| WikiArt | Acc | $0.679_{\pm0.001}$ | $0.762_{\pm0.004}$ | $0.750_{\pm0.010}$ | $0.789_{\pm0.006}$ | $0.793_{\pm0.006}$ | $0.800_{\pm0.008}$ | $0.762_{\pm0.008}$ | $0.809_{\pm0.015}$ | $0.825_{\pm0.005}^\star$ | $0.842_{\pm0.007}$ |
| HAM | Acc | $0.913_{\pm0.001}$ | $0.900_{\pm0.002}$ | $0.880_{\pm0.005}$ | $0.884_{\pm0.003}$ | $0.904_{\pm0.003}$ | $0.913_{\pm0.004}$ | $0.922_{\pm0.005}$ | $0.935_{\pm0.007}$ | $\underline{0.932}_{\pm0.003}$ | $0.953_{\pm0.006}$ |
| *Image + Text + Tabular* | | | | | | | | | | | |
| PetFinder | QWK | $0.294_{\pm0.001}$ | $0.205_{\pm0.012}$ | $0.280_{\pm0.035}$ | $0.310_{\pm0.021}$ | $0.320_{\pm0.021}$ | $0.323_{\pm0.034}$ | $0.401_{\pm0.015}$ | $0.428_{\pm0.069}$ | $0.454_{\pm0.004}^\star$ | $0.488_{\pm0.006}$ |
| COVID | Acc | $0.892_{\pm0.002}$ | $0.887_{\pm0.005}$ | $0.875_{\pm0.012}$ | $0.901_{\pm0.008}$ | $0.933_{\pm0.008}$ | $0.900_{\pm0.011}$ | $0.934_{\pm0.007}$ | $0.937_{\pm0.031}$ | $0.963_{\pm0.002}^\star$ | $0.982_{\pm0.009}$ |
| Artm | Acc | $0.530_{\pm0.003}$ | $0.507_{\pm0.008}$ | $0.510_{\pm0.020}$ | $0.522_{\pm0.012}$ | $0.574_{\pm0.012}$ | $0.598_{\pm0.015}$ | $0.623_{\pm0.010}$ | $0.655_{\pm0.047}$ | $\underline{0.638}_{\pm0.003}$ | $0.698_{\pm0.007}$ |
| AvG | – | 0.740 | 0.724 | 0.723 | 0.744 | 0.771 | 0.767 | 0.779 | 0.808 | **0.814**$^\star$ | **0.842** |
| MRR | – | 0.145 | 0.117 | 0.108 | 0.134 | 0.185 | 0.174 | 0.354 | 0.296 | **0.448**$^\star$ | **1.000** |

MRR) and more than doubles the strongest prototype baseline (PMX at 0.185 MRR). This 2.6-fold MRR gain demonstrates that case-based inference alone is insufficient; our optimal transport terms are necessary to drive the ranking improvements. Furthermore, the lightweight CAMP-F matches or exceeds the large-scale ($> 100M$ parameter) AutoGluon ensemble on 11 of 16 datasets across varied modality combinations. AutoGluon retains a marginal advantage ($< 0.02$) only in scenarios heavily dominated by a single modality (e.g., CCD, Airbnb). Under **end-to-end adaptation** (CAMP-E), CAMP establishes state-of-the-art performance across all 16 benchmarks. Its widest margin over AutoGluon (+0.068 AUC) occurs on PTech, confirming that CAMP's structural bias is particularly effective in low-data (749 samples), high-interaction environments where large ensembles typically overfit.

### 6.1.1. DATA SCARCITY AND IMBALANCE

The benchmark encompasses the core challenges that motivated our regularization strategy: limited training data ($< 2,000$ examples), severe class imbalance (e.g., 90/10), high cardinality (24–101), and long-tail distributions (detailed in Appendix D, Table 4). Tables 20–21 show that CAMP's advantages are most pronounced under these con-

ditions. Across the five lowest-resource tasks, CAMP-F averages 0.836 (outperforming AutoGluon's 0.830), while CAMP-E reaches 0.873. On the smallest datasets, CAMP-E yields substantial absolute gains over AutoGluon: +0.068 AUC on PTech (749 samples), +0.045 on COVID (707), and +0.043 on Artm (573). Similarly, under extreme imbalance and long tails, CAMP-E outperforms AutoGluon on all five relevant datasets (with CAMP-F winning four). The largest improvements are observed on Wine (30-class tail, +0.030) and WikiArt (27-class distribution, +0.033). Critically, Table 3 empirically validates the theoretical failure modes identified in Section 4: the allocation constraint prevents majority classes from monopolizing the bank (reducing dead prototypes from 26.6% to 2.3% and impurity from 0.22 to 0.04), anchors stabilize sparse tails, and geometry enforces margins between rare adjacent classes.

### 6.1.2. RESULTS UNDER PARTIAL OBSERVATION

In practice, a deployable CMC architecture must be robust to missing modalities. CAMP ensures this structurally: because its fusion gates and similarity queries operate strictly on the observed subset $\mathcal{M}_i$, a single trained model can evaluate any missing-data configuration without imputation heuristics or architectural modifications. Across 76 configu-

rations (Table 2), CAMP-E consistently yields the highest average performance. Despite utilizing $< 1\%$ of the trainable parameter count, CAMP-F also extends its lead over AutoGluon under partial observability, widening the performance gap from $+0.006$ (no-drop) to $+0.009$ (single-drop). This resilience suggests the gating mechanism actively redistributes evidential weight instead of collapsing onto a single stream.

While all models naturally converge near chance in uninformative "all-drop" scenarios, CAMP avoids the severe performance drops observed in baseline architectures when critical streams are omitted (Appendix F.6). Standard ensemble fusion tends to overfit to dominant modalities; dropping these causes AutoGluon to fall to $0.493$ on HAM (image-drop) and $0.301$ on PetFinder (tab-drop). In contrast, CAMP-F retains scores of $0.894$ and $0.425$ in those respective settings (and $0.890$ vs. $0.770$ on Fakeddit image-drop). CAMP's prototype bank dynamically retrieves class-consistent neighbors using whichever modality remains available. Lacking these OT-aligned prototypes, baseline methods (PR, PB, PMX) experience significant degradation, trailing CAMP-F substantially on single-drop averages ($0.598$–$0.635$ vs. $0.701$).

*Table 2.* **Graceful degradation under partial observation.** Summary over 16 datasets and 76 observed/dropped configurations. **Bold** marks the best value; single-drop and all-config averages are the primary robustness indicators.

| Model | No drop | Single drop | Worst | All configs |
|---|---|---|---|---|
| CAMP-E | **0.8416** | **0.7240** | 0.2806 | **0.6342** |
| CAMP-F | 0.8139 | 0.7009 | 0.2802 | 0.6154 |
| AG | 0.8076 | 0.6924 | **0.2812** | 0.6095 |
| TTT | 0.7772 | 0.6588 | 0.2781 | 0.5836 |
| PMX | 0.7710 | 0.6346 | 0.2759 | 0.5593 |

### 6.1.3. ABLATION STUDIES

Table 3 isolates the components of the structural objective under frozen encoders, showing a cumulative $+0.0694$ suite-average improvement over the Proto-Classic baseline. Each block produces the specific mechanistic effect anticipated in Section 4: *geometry* widens the similarity margin ($0.031 \to 0.066$) alongside a $+0.009$ score increase; *allocation* reduces dead prototypes ($26.6\% \to 5.2\%$) and lowers impurity ($0.22 \to 0.07$); *anchors* stabilize rare-case variance, yielding a $+0.019$ performance gain; and *classwise OT* produces the largest decrease in cross-modal disagreement ($0.23 \to 0.12$), which corresponds to the highest individual score improvement ($+0.029$).

These improvements are highly consistent. Every marginal addition sequentially improves all 16 datasets (Appendix F, Table 8), and leave-one-out analysis demonstrates uniform

performance degradation upon the removal of any block (Table 9), with OT omission causing the largest decline ($-0.029$). Crucially, this synergy is structural rather than purely additive: OT alignment is effective only after geometry separates the support, allocation ensures coverage, and anchors provide stability. These quantitative gains correlate closely with cleaner prototype utilization (Appendix F.3). Finally, CAMP exhibits strong robustness to hyperparameters; performance varies by only $\Delta \leq \pm 0.001$ across different prototype capacities $K$ (Figure 4), and our gated fusion outperforms concatenation and attention variants on 15 of 16 datasets (Table 18).

*Table 3.* **Mechanistic gains from the structural objective.** Frozen-encoder objective ablations. *Score* is the suite-average metric; prototype usage (Dead%), purity ($\widetilde{H}_{\text{pure}}$), cross-modal disagreement ($W_\varepsilon$), and similarity margin ($\overline{\Delta}$).

| Stage | Score ↑ | Dead% ↓ | $\widetilde{H}_{\text{pure}}$ ↓ | $W_\varepsilon$ ↓ | $\overline{\Delta}$ ↑ |
|---|---|---|---|---|---|
| Proto-Classic | 0.7445 | $0.266_{\pm 0.06}$ | $0.22_{\pm 0.04}$ | $0.40_{\pm 0.06}$ | $0.031_{\pm 0.018}$ |
| $+ \mathcal{L}_{\text{geo}}$ | 0.7538 | $0.224_{\pm 0.06}$ | $0.20_{\pm 0.04}$ | $0.37_{\pm 0.06}$ | $0.066_{\pm 0.017}$ |
| $+ \mathcal{L}_{\text{alloc}}$ | 0.7658 | $0.052_{\pm 0.02}$ | $0.07_{\pm 0.02}$ | $0.30_{\pm 0.05}$ | $0.063_{\pm 0.018}$ |
| $+ \mathcal{L}_{\text{anchor}}$ | 0.7852 | $0.038_{\pm 0.01}$ | $0.06_{\pm 0.01}$ | $0.23_{\pm 0.04}$ | $0.083_{\pm 0.017}$ |
| $+ \mathcal{L}_{\text{mm}}$ | **0.8139** | $0.023_{\pm 0.01}$ | $0.04_{\pm 0.01}$ | $0.12_{\pm 0.04}$ | $0.118_{\pm 0.020}$ |

### 6.2. Case Study of CAMP's Multimodal Explainability

To illustrate the practical utility of CAMP's explainable-by-design architecture, we analyze a multimodal case study on Airbnb price classification. Because pricing depends on a combination of tabular constraints (e.g., capacity) and qualitative textual features (e.g., neighborhood appeal), a faithful explanation must capture these cross-modal interactions rather than merely reporting global feature importance. Unlike post-hoc surrogates such as LIME (Ribeiro et al., 2016), our evidence probe (Figure 3) is strictly model-internal. It reuses the model's exact gates, distance metrics, and retrieved anchors, deriving unit-level evidence by fixing the retrieved support and re-evaluating target logits during tabular or text ablations (formalized in Appendix G and Algorithm G.1).

**Case Study.** Test Case 744 is correctly classified into Class 5 ($\$100$–$120$) with $98.34\%$ confidence. The gating module assigns $60.84\%$ of the evidential weight to tabular data and $39.16\%$ to text (Figure 3A). The query retrieves the nearest Class 5 prototypes and anchors, linking the prediction to historical listings with similar capacities and locations. Applying the fixed-support probe identifies *num_bedrooms* and *num_bathrooms* as core positive tabular features. Simultaneously, it identifies "prime location" as positive textual support, while correctly interpreting "Opposite of Sthybus station" as negative evidence. By intervening directly on the input while strictly freezing the prototype

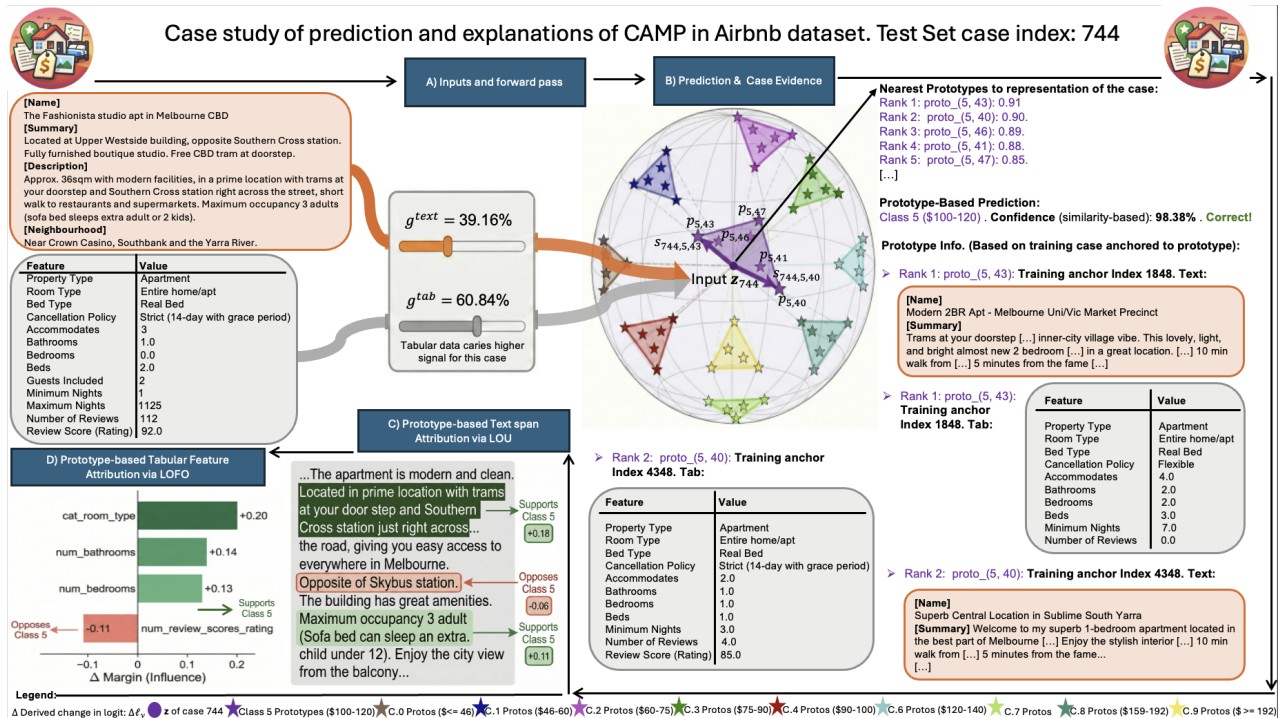

*Figure 3.* **CAMP explanation probe for Airbnb price classification (Test Case 744). (A)** Instance-wise gates combine tabular and text evidence ($g^{tab} = 60.84\%$, $g^{text} = 39.16\%$). **(B)** The fused embedding $z_{744}$ is scored against the class-partitioned prototype bank and supported by nearest Class 5 prototypes/anchors. **(C & D)** With retrieved support fixed, LOFO and LOU ablations rank tabular fields and text spans by their effect on the target logit; Appendix G formalizes the probe and faithfulness scope.

support, these attributions maintain computational faithfulness to the head's exact inference process. Ultimately, CAMP's standard outputs—modality gates, retrieved cases, and local feature impacts—provide inspectable evidence without requiring an auxiliary explanation model. While optimizing human-facing visual interfaces remains future work, the ability to extract computationally faithful evidence while maintaining state-of-the-art performance highlights the practical value of the architecture.

## 7. Conclusion

As multimodal machine learning transitions to high-stakes deployments, such as clinical triage and fraud detection, modalities rarely merely re-describe one another; they *complete* one another. Consequently, emerging regulations like the EU AI Act increasingly require inspectable evidence behind every prediction. CAMP is, to our knowledge, the first prototype framework designed for this complementary regime (CMC). It demonstrates that the presumed trade-off between scale/accuracy and faithful interpretability is not fundamental, but merely an artifact of mismatched objectives. By resolving CMC-specific failure modes through a unified structural objective (class-wise optimal transport, geometry, allocation, and anchors), CAMP delivers high-performance interpretability *for free*. A sub-1M-parameter

head on frozen encoders rivals > 100M-parameter AutoML systems across 16 datasets, degrades gracefully under missing modalities, and natively exposes its reasoning through retrieved cases and faithful local attributions.

We view CAMP as a foundational step. Its structural-bias-over-scale approach invites extensions to regression and structured prediction, while the OT-based alignment principle could serve as a drop-in coherence regularizer for other asymmetric multimodal architectures. Practically, CAMP's success in the lightweight, frozen-encoder regime makes it highly accessible for enterprise and clinical settings constrained to black-box API encoders.

**Limitations.** While our formal guarantees characterize the frozen-encoder regime, the end-to-end gains remain empirical. Furthermore, our fixed-support audit guarantees model-internal computational faithfulness, but user studies are required to validate human-facing explanation efficacy. Finally, our current evaluation targets discrete classification with strong pretrained encoders. Extending this framework to regression, continuous concept drift, dedicated batch-composition studies, and human-centered evaluation remain exciting avenues for future work.

## Impact Statement

This paper presents work whose goal is to advance the field of Machine Learning. There are many potential societal consequences of our work, none of which we feel must be specifically highlighted here.

## Disclaimer

This paper was prepared for informational purposes by the Artificial Intelligence Research group of JPMorgan Chase & Co. and its affiliates ("JP Morgan") and is not a product of the Research Department of JP Morgan. JP Morgan makes no representation and warranty whatsoever and disclaims all liability, for the completeness, accuracy or reliability of the information contained herein. This document is not intended as investment research or investment advice, or a recommendation, offer or solicitation for the purchase or sale of any security, financial instrument, financial product or service, or to be used in any way for evaluating the merits of participating in any transaction, and shall not constitute a solicitation under any jurisdiction or to any person, if such solicitation under such jurisdiction or to such person would be unlawful.

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

**Appendix organization.** Appendix A introduces notation and lightweight OT preliminaries used by all proofs. Appendix B gives *loss-by-loss* guarantees explaining which degeneracies each CAMP objective term prevents. Appendix C contains full proofs of the main theorems stated in Sec. 4. Appendices D–I keep the non-theory material from the original submission (training details, dataset suite, implementation notes, full results, ablations, and the causal-faithfulness audit).

# A. Preliminaries and notation

This appendix consolidates the theoretical statements and corresponding proofs referenced in Section 4. To ensure a self-contained narrative, we retain the notation established in the main text and briefly restate the essential definitions below.

The core theoretical contribution of our work lies not in establishing new optimal transport (OT) theorems in isolation, but in demonstrating how classical tools compose within the coherent alignment of multimodal prototypes (CAMP) setting. While our proof ingredients are standard. Relying on margin preservation for Lipschitz prototype scores, Pinsker and Markov inequalities for allocation bounds, elementary anchor stability, and Wasserstein and Kantorovich–Rubinstein inequalities. Their synthesis reveals a key insight: OT over class-wise prototype-usage measures is only meaningful when the shared prototype support is well-separated, actively utilized, class-specialized, and stable. These guarantees are derived for the prototype head within a frozen-encoder regime, complementing the empirical validation of our end-to-end fine-tuning approach.

The remainder of this appendix is organized as follows: Appendix B.1 analyzes the underlying geometry and margins, Appendix B.2 addresses support coverage and class purity, Appendix B.3 formalizes the distributional alignment of prototype usage, Appendix B.4 establishes anchor stability, and Appendix C presents the complete proofs for our CMC-specific composition statements.

**Notation and Setup**

Recall that our training set consists of partially-observed multimodal examples, denoted as $\{(\{x_i^{(m)}\}_{m \in \mathcal{M}_i}, y_i)\}_{i=1}^N$, where $\mathcal{M}_i \subseteq \mathcal{M}$ represents the observed modality set for the $i$-th sample and $y_i \in \{1, \ldots, C\}$ is its corresponding class label. Each modality encoder and its subsequent projection produce an embedding $z_i^{(m)} \in \mathbb{R}^d$. These are aggregated into a gated fusion embedding $z_i = \sum_{m \in \mathcal{M}_i} g_i^{(m)} z_i^{(m)}$ (Eq. (4)).

For the classification head, we learn $K$ prototypes $\{p_{c,k}\}_{k=1}^K \subset \mathbb{R}^d$ for each class $c$. Given an input embedding $z$, the prototype head evaluates the similarity $S(z, p_{c,k})$ against these prototypes. The class logits are then computed as $\ell_c(z) = \frac{1}{r} \sum_{k \in \text{Top}_r(\{S(z,p_{c,j})\}_{j=1}^K)} S(z, p_{c,k})$ (Eq. (5)), yielding the final prediction $\hat{y}(z) = \arg\max_c \ell_c(z)$.

To compute the OT alignment loss $\mathcal{L}_{\text{mm}}$ (Eq. (7)), we measure the prototype usage for each class $c$ and modality $m$. The assignment weights $\alpha_{i,c,k}^{(m)}$ induce an empirical usage histogram $w_{c,k}^{(m)}$, which in turn defines a discrete measure $\nu_c^{(m)} = \sum_{k=1}^K w_{c,k}^{(m)} \delta_{p_{c,k}}$ over the shared support $\{p_{c,k}\}_{k=1}^K$ (Eq. (6)). The loss $\mathcal{L}_{\text{mm}}$ then averages the Sinkhorn-regularized transport costs between pairs of these modality-specific measures.

**Assumption A.1** (Bounded embeddings and similarity regularity). Throughout the analysis, we assume the normalized regime utilized by our method, such that $\|z\|_2 \leq 1$ and $\|p\|_2 \leq 1$. We further assume that the similarity function $S(z, p)$ is bounded and $L_S$-Lipschitz with respect to its first argument, satisfying $|S(z, p) - S(z', p)| \leq L_S \|z - z'\|_2$. For the gradient analysis in Appendix B.4, we additionally require $S$ to be differentiable with respect to $p$, bounded by $\|\nabla_p S(z, p)\|_2 \leq L$.

*Remark* A.2 (Wasserstein preliminaries). Our proofs rely on two standard properties. First, for any coupling $(X, Y)$ with marginals $X \sim \mu$ and $Y \sim \nu$, Jensen's inequality implies $\|\mathbb{E}[X] - \mathbb{E}[Y]\|_2 \leq \sqrt{\mathbb{E}\|X - Y\|_2^2}$. Second, by Kantorovich–Rubinstein duality, we have $|\mathbb{E}_\mu[\varphi] - \mathbb{E}_\nu[\varphi]| \leq W_1(\mu, \nu)$ for any 1-Lipschitz function $\varphi$, alongside the standard bound $W_1 \leq W_2$ (Villani, 2009).

*Remark* A.3 (Entropic approximation gap). Under mild conditions, the Sinkhorn-regularized transport cost $W_\varepsilon$ converges to the exact squared 2-Wasserstein distance $W_2^2$ as the regularization parameter $\varepsilon \to 0$ (Cuturi, 2013; Feydy et al., 2019). For a fixed $\varepsilon$ and a predetermined number of Sinkhorn iterations, we define $\delta_\varepsilon$ as an upper bound on this approximation error:

$$\delta_\varepsilon \triangleq \left| W_\varepsilon - W_2^2 \right|. \tag{15}$$

This explicit error term is utilized in the proof of Theorem 4.1.

# B. Loss-by-loss guarantees

This section establishes local guarantees for each constituent loss block introduced in Eq. (13). To contextualize these theoretical results, each subsection begins by outlining the specific degenerate solutions it prevents, followed by formal statements and their complete proofs.

## B.1. Geometry: similarity margins and non-collapsing prototypes

Without structural regularization, the optimal transport alignment loss could be trivially minimized if the class prototypes were to collapse into a single point, a scenario that would inevitably destroy the model's discriminative power. To prevent this, the geometry block $\mathcal{L}_{\text{geo}}$ enforces a class-conditional similarity margin between correct and incorrect prototypes, alongside strict within-class prototype separation. Together, these constraints preclude redundant collapse and guarantee certified local robustness.

**Theorem B.1** (Class-conditioned geometry prevents collapse and yields a similarity margin). *If $\mathcal{L}_{pull} = \mathcal{L}_{rep} = 0$, then every training point has a similarity margin $\Delta \equiv \rho_+ - \rho_- > 0$ between its best same-class prototype and any other-class prototype. Moreover, if $S(\cdot, p)$ is $L$-Lipschitz in $z$, then for any perturbation $\delta$ with $\|\delta\|_2 < \Delta/(2L)$, the predicted class under the $r=1$ rule is unchanged. If additionally $\mathcal{L}_{pp\text{-}intra} = 0$ and $S(z, p) = \psi(\|z - p\|_2)$ with strictly decreasing $\psi$, then within each class $c$ all prototypes are separated by a non-zero distance, ruling out redundant prototype collapse.*

*Proof.* Fix any training point $i$ and define $s_i^+ \equiv \max_k s_{i,y_i,k}$ and $s_i^- \equiv \max_{c \neq y_i, k} s_{i,c,k}$. The conditions $\mathcal{L}_{\text{pull}} = \mathcal{L}_{\text{rep}} = 0$ imply $s_i^+ \geq \rho_+$ and $s_i^- \leq \rho_-$ for all $i$, hence $s_i^+ - s_i^- \geq \Delta$. Lipschitzness of $S$ then yields robustness for perturbations $\|\delta\|_2 < \Delta/(2L)$ by a standard margin-preservation argument. Finally, $\mathcal{L}_{\text{pp-intra}} = 0$ implies $S(p_{c,k}, p_{c,k'}) \leq \rho_{\text{in}}$ for $k \neq k'$; strict monotonicity of $\psi$ converts this to a non-zero Euclidean separation. $\qquad\square$

## B.2. Allocation: full support and class-specialized prototypes

Prototype-based models risk wasting representational capacity on inactive units that are never selected, or conversely, blurring class semantics if prototypes are uniformly activated across all labels. The allocation block, defined as $\mathcal{L}_{\text{alloc}} = \mathcal{L}_{\text{cov}} + \lambda_{\text{pure}} \mathcal{L}_{\text{pure}}$, mitigates these risks by enforcing near-uniform average usage to eliminate dead units, and promoting low-entropy label association to ensure strict class specialization.

**Theorem B.2** (Coverage implies no dead prototypes). *Let $\bar{\alpha} \in \Delta^{CK-1}$ be the average prototype usage distribution, defined by $\bar{\alpha}_k := \frac{1}{N} \sum_{i=1}^N \alpha_{i,k}$ over the $CK$ prototypes, and let $\mathcal{L}_{cov} = D_{\text{KL}}(\bar{\alpha}\|\mathbf{u})$ where $\mathbf{u} = \frac{1}{CK}\mathbf{1}$. If $\mathcal{L}_{cov} \leq \varepsilon$, then*

$$\min_k \bar{\alpha}_k \ \geq \ \frac{1}{CK} - \sqrt{2\varepsilon}. \tag{16}$$

*In particular, for $\varepsilon < \frac{1}{2C^2 K^2}$ every prototype receives non-trivial mass.*

*Proof.* Let $\mathbf{u} = \frac{1}{CK}\mathbf{1}$. By Pinsker's inequality,

$$\|\bar{\alpha} - \mathbf{u}\|_1 \leq \sqrt{2D_{\text{KL}}(\bar{\alpha}\|\mathbf{u})} \leq \sqrt{2\varepsilon}. \tag{17}$$

For any coordinate $k$, $\bar{\alpha}_k \geq u_k - \|\bar{\alpha} - \mathbf{u}\|_1 \geq \frac{1}{CK} - \sqrt{2\varepsilon}$, which proves Eq. (16). $\qquad\square$

**Theorem B.3** (Purity implies a dominant label for each prototype). *If $H(q_k) \leq \varepsilon$, then $\max_c q_k(c) \geq e^{-\varepsilon}$. Consequently, if the average purity loss satisfies $\mathcal{L}_{pure} \leq \varepsilon$, then for any $\delta \in (0, 1)$ at least a $(1 - \delta)$ fraction of prototypes have a dominant class probability at least $\exp(-\varepsilon/\delta)$.*

*Proof.* Let $q_{\text{max}} \equiv \max_c q_k(c)$. Since $q_k(c) \leq q_{\text{max}}$ for all $c$, we have $\log(1/q_k(c)) \geq \log(1/q_{\text{max}})$ and therefore

$$H(q_k) = \sum_c q_k(c) \log \frac{1}{q_k(c)} \geq \sum_c q_k(c) \log \frac{1}{q_{\text{max}}} = \log \frac{1}{q_{\text{max}}}.$$

Rearranging yields $q_{\text{max}} \geq e^{-H(q_k)} \geq e^{-\varepsilon}$. For the "most prototypes" claim, set $H_k \equiv H(q_k) \geq 0$ and apply Markov's inequality to $\frac{1}{CK} \sum_k H_k = \mathcal{L}_{\text{pure}} \leq \varepsilon$. $\qquad\square$

## B.3. Optimal transport: distributional evidence alignment

While strict pointwise alignment risks destroying valuable modality-specific information, the optimal transport block preserves this data by aligning modalities in distribution. By matching the class-conditioned prototype-usage measures $\nu_c^{(m)}$, this approach prevents cross-modal evidence conflict while simultaneously allowing for pointwise complementarity.

**Proposition B.4** (OT controls discrepancies of Lipschitz prototype statistics). *Let $\nu$ and $\nu'$ be two probability measures on the prototype space (e.g., $\nu_c^{(m)}$ and $\nu_c^{(m')}$). For any 1-Lipschitz statistic $\varphi$ (with respect to $\| \cdot \|_2$),*

$$\left| \mathbb{E}_\nu[\varphi(p)] - \mathbb{E}_{\nu'}[\varphi(p)] \right| \leq W_1(\nu, \nu') \leq W_2(\nu, \nu'). \tag{18}$$

*In particular, taking $\varphi(p) = u^\top p$ for any $\|u\|_2 \leq 1$ implies $\|\mathbb{E}_\nu[p] - \mathbb{E}_{\nu'}[p]\|_2 \leq W_2(\nu, \nu')$.*

*Proof.* The first inequality in Eq. (18) is exactly Kantorovich-Rubinstein duality for $W_1$. The second inequality $W_1 \leq W_2$ holds for measures on a metric space since Wasserstein distances are monotone in their order (see, e.g., (Villani, 2009)). For the mean bound, note that $\|\mathbb{E}_\nu[p] - \mathbb{E}_{\nu'}[p]\|_2 = \sup_{\|u\|_2 \leq 1} |\mathbb{E}[u^\top p] - \mathbb{E}[u^\top p']|$ and apply Eq. (18) with $\varphi(p) = u^\top p$. $\square$

**Lemma B.5** (Non-degeneracy turns low OT cost into prototype-wise agreement). *Fix a class $c$ and two modalities $m, m'$. Write the class-conditioned measures on the shared support as $\nu = \sum_{k=1}^K w_k \delta_{p_{c,k}}$ and $\nu' = \sum_{k=1}^K w_k' \delta_{p_{c,k}}$. Assume within-class prototype separation $\min_{k \neq k'} \|p_{c,k} - p_{c,k'}\|_2 \geq \Delta_c > 0$. Then*

$$\|w - w'\|_1 \leq \frac{2}{\Delta_c^2} W_2^2(\nu, \nu'). \tag{19}$$

*Proof.* Let $\pi$ be any coupling between $w$ and $w'$. The total mass that can be placed on the diagonal is at most $\sum_k \min\{w_k, w_k'\} = 1 - \frac{1}{2}\|w - w'\|_1$. Therefore the off-diagonal mass satisfies $\sum_{k \neq k'} \pi_{k,k'} \geq \frac{1}{2}\|w - w'\|_1$. By separation, $\|p_{c,k} - p_{c,k'}\|_2^2 \geq \Delta_c^2$ whenever $k \neq k'$, hence

$$\sum_{k,k'} \pi_{k,k'} \|p_{c,k} - p_{c,k'}\|_2^2 \geq \sum_{k \neq k'} \pi_{k,k'} \Delta_c^2 \geq \frac{\Delta_c^2}{2}\|w - w'\|_1.$$

Minimizing the left-hand side over couplings $\pi$ yields $W_2^2(\nu, \nu') \geq \frac{\Delta_c^2}{2}\|w - w'\|_1$ and rearranging gives Eq. (19). $\square$

## B.4. Anchors: bounded influence and bounded drift

Because optimal transport alignment operates as a global objective, it is inherently sensitive to rare outliers and transient, modality-specific artifacts. The anchor block $\mathcal{L}_{\text{anchor}}$ addresses this vulnerability by identifying and downweighting the influence of these rare samples. Concurrently, it tethers each class's prototype cloud to a slowly moving anchor, establishing the critical stability conditions required for our subsequent coherence analysis.

**Theorem B.6** (Anchors yield bounded influence for detected outliers). *Assume $S(z, p)$ is differentiable in $p$ and satisfies $\|\nabla_p S(z_i, p)\|_2 \leq L$ for all $i$ and $p$. Let $\varepsilon \equiv \frac{1}{N}\sum_{i=1}^N \omega_i$ be the fraction of detected outliers (Eq. (12)). Then:*

*(i) **Bounded influence.** For every prototype $p_{c,k}$, $\|\nabla_{p_{c,k}}\mathcal{L}_{anc}\|_2 \leq \frac{L\varepsilon}{CK}$.*

*(ii) **Bounded drift of anchors.** If $\|\bar{z}_c^{(t)}\|_2 \leq 1$ and $\|a_c^{(0)}\|_2 \leq 1$, then the EMA update $a_c^{(t)} = (1 - \beta)a_c^{(t-1)} + \beta \bar{z}_c^{(t)}$ keeps $\|a_c^{(t)}\|_2 \leq 1$ and satisfies $\|a_c^{(t)} - a_c^{(t-1)}\|_2 \leq 2\beta$ for all $t$.*

*(iii) **Anchor-consistency contracts class means.** Let $\mu_c \equiv \frac{1}{K}\sum_{k=1}^K p_{c,k}$ and consider a gradient step on $\mathcal{L}_{anc\text{-}cons} = \frac{1}{C}\sum_c \|a_c - \mu_c\|_2^2$ with step size $\eta \in (0, C/2)$. Then the class mean update obeys $\mu_c^+ = (1 - 2\eta/C)\mu_c + (2\eta/C)a_c$, hence $\|\mu_c^+ - a_c\|_2 \leq (1 - 2\eta/C)\|\mu_c - a_c\|_2$. In particular, when combined with (i), this yields bounded prototype-bank drift, supporting the stability condition used in Theorem 4.1.*

*Proof.* **(i) Bounded influence.** In $\mathcal{L}_{\text{anc}}$ (Eq. (12)), the factor $\omega_i$ gates the contribution of sample $i$. For any fixed prototype $p_{c,k}$, each summand contributes at most $L$ in gradient norm by assumption. The loss averages over $N$ samples and $CK$ prototypes, hence $\|\nabla_{p_{c,k}}\mathcal{L}_{\text{anc}}\|_2 \leq \frac{1}{N}\sum_{i=1}^N \omega_i \cdot \frac{L}{CK} = \frac{L\varepsilon}{CK}$.

**(ii) Bounded drift of anchors.** By convexity of the unit ball, if $\|a_c^{(t-1)}\|_2 \leq 1$ and $\|\bar{z}_c^{(t)}\|_2 \leq 1$, then $\|a_c^{(t)}\|_2 \leq (1-\beta)\|a_c^{(t-1)}\|_2 + \beta\|\bar{z}_c^{(t)}\|_2 \leq 1$. Moreover, $\|a_c^{(t)} - a_c^{(t-1)}\|_2 = \beta\|\bar{z}_c^{(t)} - a_c^{(t-1)}\|_2 \leq \beta(\|\bar{z}_c^{(t)}\|_2 + \|a_c^{(t-1)}\|_2) \leq 2\beta$.

**(iii) Contraction of class means under $\mathcal{L}_{\text{anc-cons}}$.** For a fixed class $c$, the gradient of $\|a_c - \mu_c\|_2^2$ with respect to each prototype $p_{c,k}$ is $\nabla_{p_{c,k}}\|a_c - \mu_c\|_2^2 = \frac{2}{K}(\mu_c - a_c)$. Therefore a gradient step on $\mathcal{L}_{\text{anc-cons}}$ updates $p_{c,k}^+ = p_{c,k} - \eta \cdot \frac{2}{CK}(\mu_c - a_c)$ and thus $\mu_c^+ = \mu_c - \eta \cdot \frac{2}{C}(\mu_c - a_c) = (1 - 2\eta/C)\mu_c + (2\eta/C)a_c$. Rearranging gives $\mu_c^+ - a_c = (1 - 2\eta/C)(\mu_c - a_c)$ and the stated contraction. This contraction, combined with (i), ensures that outlier-driven gradients cannot move the prototype configuration arbitrarily far from the anchored class geometry. $\square$

# C. Proofs of Structural Regularization

This section details the complete formal derivations for the theoretical results introduced in Section 4.

## C.1. Proof of Theorem 4.1 (Coherence via Transport)

*Proof.* This result follows as a direct specialization of Proposition B.4. Let us fix a class $c$ along with two modalities $m$ and $m'$, defining their respective measures as $\nu \equiv \nu_c^{(m)}$ and $\nu' \equiv \nu_c^{(m')}$. For any 1-Lipschitz semantic feature function $\varphi$, applying Proposition B.4 immediately yields:

$$\left|\mathbb{E}_\nu[\varphi(p)] - \mathbb{E}_{\nu'}[\varphi(p)]\right| \leq W_1(\nu, \nu') \leq W_2(\nu, \nu'),$$

which is exactly the claimed bound. $\square$

**Proposition C.1** (Modality subset stability under pairwise optimal transport alignment). *Fix a class $c$. For each modality $m \in \mathcal{M}$, let $\nu^{(m)} \equiv \nu_c^{(m)}$ denote the class-conditioned prototype-usage measure. Assume that for every pair of modalities $m, m' \in \mathcal{M}$, the Sinkhorn-regularized transport cost is bounded by $\gamma$:*

$$W_\varepsilon(\nu^{(m)}, \nu^{(m')}) \leq \gamma. \tag{20}$$

*Let $\nu^\star \equiv \nu^{(m_0)}$ serve as the measure for an arbitrary reference modality $m_0$. Then, for any nonempty subset of modalities $\mathcal{S} \subseteq \mathcal{M}$, the averaged measure $\nu^{(\mathcal{S})} \equiv \frac{1}{|\mathcal{S}|}\sum_{m \in \mathcal{S}} \nu^{(m)}$ satisfies the following bound:*

$$W_2^2(\nu^{(\mathcal{S})}, \nu^\star) \leq \gamma + \delta_\varepsilon, \tag{21}$$

*where $\delta_\varepsilon$ represents the entropic gap defined in Eq. (15). Consequently, for any 1-Lipschitz statistic $\varphi$, it holds that $\left|\mathbb{E}_{\nu^{(\mathcal{S})}}[\varphi(p)] - \mathbb{E}_{\nu^\star}[\varphi(p)]\right| \leq \sqrt{\gamma + \delta_\varepsilon}$.*

*Proof.* Let us designate an arbitrary reference modality $m_0$ and define $\nu^\star \equiv \nu^{(m_0)}$. By combining the assumption in Eq. (20) with the entropic gap inequality, we can bound the exact squared Wasserstein distance for every modality $m$ as $W_2^2(\nu^{(m)}, \nu^\star) \leq W_\varepsilon(\nu^{(m)}, \nu^\star) + \delta_\varepsilon \leq \gamma + \delta_\varepsilon$.

Now, consider a nonempty subset $\mathcal{S} \subseteq \mathcal{M}$ and its associated averaged measure $\nu^{(\mathcal{S})} = \frac{1}{|\mathcal{S}|}\sum_{m \in \mathcal{S}} \nu^{(m)}$. By leveraging the convexity of the squared Wasserstein distance $W_2^2(\cdot, \nu^\star)$ with respect to its first argument (since it is defined as the infimum of linear functionals over couplings), we obtain:

$$W_2^2(\nu^{(\mathcal{S})}, \nu^\star) \leq \frac{1}{|\mathcal{S}|}\sum_{m \in \mathcal{S}} W_2^2(\nu^{(m)}, \nu^\star) \leq \gamma + \delta_\varepsilon,$$

which proves Eq. (21). Finally, applying Proposition B.4 alongside the standard inequality $W_1 \leq W_2$ yields the stated bound for the Lipschitz statistic. $\square$

# D. Dataset details

We evaluate our method on the AutoGluon benchmark suite for CMC (Shi et al., 2021; Tang et al., 2024b;a), restricted to the 16 datasets utilized throughout this paper. Each dataset represents a supervised classification task where every instance corresponds to a row in a data table featuring one or more modalities, including text, images, and structured attributes.

| Dataset | Modalities | Train | Test | Feat. (T/C/N) | Task (Metric) | Target | Key Structural Challenge |
|---|---|---|---|---|---|---|---|
| **Tabular and Text** | | | | | | | |
| IMDB | Tab+Text | 800 | 200 | 4 / 0 / 7 | Binary (AUC) | Drama classification | Low data |
| Fake Job | Tab+Text | 12,725 | 3,182 | 3 / 2 / 0 | Binary (AUC) | Fraudulent posting | Extreme class imbalance (95/5) |
| Kick | Tab+Text | 86,502 | 21,626 | 3 / 3 / 3 | Binary (AUC) | Funding success | Large-scale baseline |
| Jigsaw | Tab+Text | 100,000 | 25,000 | 1 / 2 / 27 | Binary (AUC) | Toxicity detection | Extreme class imbalance (90/10) |
| Wine | Tab+Text | 84,123 | 21,031 | 3 / 0 / 2 | 30-class (Acc) | Grape variety | High cardinality, long-tail |
| Airbnb | Tab+Text | 18,316 | 4,579 | 28 / 37 / 24 | 24-class (Acc) | Price bin | High cardinality, ordinal |
| **Image and Text** | | | | | | | |
| PTech | I+T | 749 | 200 | 1 / 0 / 0 | Binary (AUC) | Smear persuasion | Very low data |
| Food101 | I+T | 13,613 | 4,547 | 1 / 0 / 0 | 101-class (Acc) | Food category | High cardinality |
| Fakeddit | I+T | 16,277 | 5,723 | 1 / 0 / 0 | 6-class (Acc) | Fake news class | Multi-way classification |
| Memotion | I+T | 6,992 | 1,878 | 1 / 0 / 0 | 4-class (Acc) | Meme sentiment | Sentiment skew, moderate data |
| **Image and Tabular** | | | | | | | |
| CCD | I+Tab | 1,126 | 374 | 0 / 2 / 0 | Binary (AUC) | Accident involvement | Low data |
| WikiArt | I+Tab | 15,278 | 5,084 | 0 / 2 / 0 | 27-class (Acc) | Artist category | High cardinality, long-tail |
| HAM | I+Tab | 8,010 | 2,005 | 0 / 3 / 1 | 7-class (Acc) | Skin lesion type | Medical class imbalance |
| **Tri-modal** | | | | | | | |
| PetFinder | I+T+Tab | 11,994 | 2,999 | 1 / 15 / 5 | 5-class (QWK) | Adoption speed | Ordinal classification |
| COVID | I+T+Tab | 707 | 222 | 2 / 9 / 3 | 4-class (Acc) | Pneumonia category | Low data, medical context |
| Artm | I+T+Tab | 573 | 177 | 3 / 3 / 1 | 4-class (Acc) | Art movement | Very low data |

*Table 4.* **Benchmark suite and stressor profile.** Quantitative splits, modality composition, task metric, target, and the principal statistical challenge for each dataset. Feature counts are listed as Text/Categorical/Numeric (T/C/N), and the suite follows the AutoGluon multimodal benchmarks (Shi et al., 2021; Tang et al., 2024b;a).

This matches our CMC problem definition (Sec. 3.1): models must predict labels from arbitrary subsets of loosely coupled modalities, where each modality provides a distinct view of the same underlying entity (e.g., an image of a medical lesion accompanied by patient metadata, or a property listing description paired with structured amenities).

Across all 16 tasks, the constituent modalities serve as different measurement channels for the same underlying subject. These encompass unstructured text (e.g., reviews, descriptions, clinical notes), images (e.g., photos, posters, X-rays), and structured covariates (e.g., demographics, listing attributes). The learning problem differs from standard image and caption matching. The text fields are typically authored independently of the images, and the structured fields consist of curated attributes that cannot be deterministically inferred from the raw text or pixels alone. These properties instantiate the characteristic CMC setting of weakly coupled channels offering complementary evidence for the target label.

From the broader AutoGluon catalogs, we retain the datasets possessing intact download links and an unambiguous mapping to our problem definition. We discard a small number of benchmark tasks where access links were broken at the time of experimentation (specifically the prod, salary, and channel datasets from (Shi et al., 2021)) or where the provided modalities effectively act as near-duplicates of the same underlying signal (e.g., two views derived from the exact same text source); this is the case with the AEP dataset from (Tang et al., 2024b), which fails to instantiate true complementary channels.

### D.1. Preprocessing pipeline

We use the standardized dataset snapshots and train/test splits provided by the original benchmark releases (Shi et al., 2021; Tang et al., 2024b;a). When early stopping or model selection requires a validation split, we carve it from the training set using a label-stratified split. Our preprocessing pipeline applies the following modality-specific procedures:

For tabular fields, all numeric features are centered and rescaled using training-set statistics, and any missing numeric values are imputed with the training-set mean. Categorical columns are handled as discrete values; both natively missing entries

and inference-time unseen categories are explicitly mapped to a unified unknown category token (Shi et al., 2021). For text fields, missing entries are simply treated as empty strings. In datasets with multiple text columns, we concatenate the individual fields in a fixed order, interspersed with separator tokens, before passing them to the backbone tokenizer. The resulting sequences are padded or truncated to a globally chosen maximum length, and we do not apply any stochastic text augmentations. Finally, for image inputs, each table row is associated with a single, representative image file path provided by the benchmark snapshot. These images are decoded, converted to RGB, resized to match the specific backbone's required input resolution, and normalized using the backbone's default statistical normalization. Consistent with the text pipeline, no stochastic visual augmentations are applied during training.

### D.2. Dataset descriptions, sources, and licenses

This subsection records the provenance, usage terms, and benchmark construction choices for every dataset used in the 16-task suite. Each entry states the prediction problem, the complementary evidence supplied by the modalities, and the source and license or usage statement followed in our experiments. The Tab+Text datasets use the processed snapshots of Shi et al. (2021); image-containing datasets follow the sourcing and split protocol of Tang et al. (2024b).

TABULAR AND TEXT DATASETS

For the six Tab+Text datasets, we use the processed benchmark snapshots distributed by Shi et al. (2021). These snapshots are released under CC BY-NC-SA as part of that benchmark, while the raw sources retain the terms stated by their original curators.

**IMDB (imdb).**
Movie records contain free-text fields, such as synopses and keywords, together with structured attributes such as ratings and release years. The target is whether a film is labeled as drama. The task is complementary because narrative cues in the text and reception or production metadata provide different evidence for the genre label. Source: Kaggle IMDB dataset (PromptCloudHQ). License/usage: CC BY-NC-SA for the processed benchmark snapshot as part of the AutoGluon benchmark release (Shi et al., 2021).

**Fake Job (fake).**
Job advertisements are represented by text fields, including title, description, and requirements, and by structured posting metadata such as employment type and telecommuting flags. The target is whether the posting is fraudulent. The task requires both language cues and metadata cues, which can each independently indicate fraud. Source: Kaggle dataset derived from the Employment Scam Aegean Dataset (shivamb; Vidros et al., 2017). License/usage: CC0 for the raw source; processed benchmark snapshot under CC BY-NC-SA (Shi et al., 2021).

**Kickstarter (kick).**
Kickstarter projects pair descriptive text with structured fundraising and project attributes. The target is whether a project reaches its funding goal. The text describes the proposal, whereas the tabular features encode financial and categorical context that is not reducible to the text. Source: Kaggle funding-success dataset (codename007). License/usage: processed benchmark snapshot under CC BY-NC-SA (Shi et al., 2021).

**Jigsaw (jigsaw).**
Online comments are paired with structured annotations and identity-related fields to predict toxicity. The comment itself provides linguistic evidence, while tabular annotations capture subgroup and metadata signals that are important in this highly imbalanced task. Source: Kaggle Jigsaw Unintended Bias challenge and related release (Kaggle, a; Google & Hugging Face; Borkan et al., 2019). License/usage: processed benchmark snapshot under CC BY-NC-SA (Shi et al., 2021).

**Wine (wine).**
Wine reviews combine tasting-note text with numeric attributes such as score and price; the target is the grape variety. Text describes sensory evidence, while tabular attributes provide rating and market context that helps resolve long-tail variety distinctions. Source: Kaggle Wine Reviews (zynicide). License/usage: CC BY-NC-SA 4.0 for the raw source; processed benchmark snapshot under CC BY-NC-SA (Shi et al., 2021).

**Airbnb (airbnb).**
Melbourne Airbnb listings contain rich listing text and structured property attributes. The target is a discretized price

label. The text conveys qualitative location and presentation signals, whereas structured features such as room type, capacity, bathrooms, and amenities provide constraints that cannot be reliably inferred from the text alone. Source: Melbourne Airbnb Open Data (tylerx). License/usage: CC0 for the raw source; processed benchmark snapshot under CC BY-NC-SA (Shi et al., 2021).

IMAGE AND TEXT DATASETS

For image-text datasets, the raw-source usage statement and the split protocol follow the AutoGluon multimodal benchmark descriptions (Tang et al., 2024b). We retain the benchmark's single-image/single-text representation for each row.

**PTech (ptech).**
Internet memes are represented by an image and associated text, and the target is whether the smear persuasion technique is present. The visual composition supplies affective or metaphorical context that changes how the text should be interpreted. Source: SemEval-2021 Task 6 corpus (Dimitrov et al., 2021; di-dimitrov & others). License/usage: released for general research use (CC0 1.0); the benchmark construction combines the original train and validation splits, removes duplicates, and keeps the original test split (Tang et al., 2024b).

**Food101 (food101).**
Food images are paired with recipe or web-page text, and the target is one of 101 food categories. Visual evidence captures plating and appearance, while text can identify ingredients that are visually ambiguous. Source: UPMC-Food101 Kaggle release (Wang et al., 2015; gianmarco96). License/usage: reported as public by the AutoGluon benchmark; the benchmark randomly selects 20% of the original data from each category to form new train/test splits (Tang et al., 2024b).

**Fakeddit (fakeddit).**
Reddit posts are represented by an attached image and title text, and the target is the fine-grained fake-news category. The title provides the claim or framing, while the image provides the purported visual evidence; mismatches between them are central to the classification problem. Source: Fakeddit (Nakamura et al., 2020; entitize). License/usage: cite the dataset when used; the benchmark uses the 6-way category setting and samples 3% of the original training set and 10% of the original test set (Tang et al., 2024b).

**Memotion (memotion).**
Meme images and embedded text are used to classify sentiment. The sentiment often emerges from the interaction between a neutral image and sarcastic text, or conversely from a visually expressive image and short text. Source: SemEval-2020 Memotion task (Sharma et al., 2020). License/usage: allowed to be used in papers upon citation; the benchmark follows the original task release and split protocol (Tang et al., 2024b).

IMAGE AND TABULAR DATASETS

**CCD (ccd).**
Dashcam frames are paired with structured traffic or environmental metadata to predict whether the ego-vehicle is involved in an accident. The image gives the immediate visual configuration, while the metadata records contextual factors such as weather or scene conditions. Source: Car Crash Dataset / CCD Kaggle release (asefjamilajwad; Bao et al., 2020). License/usage: MIT License; the benchmark uses the image corresponding to the accident frame and splits the data at a 3:1 train/test ratio (Tang et al., 2024b).

**WikiArt (wikiart).**
Artwork images are paired with structured metadata to classify the artist. The image captures palette, texture, and brush-work, while metadata constrains the historical and stylistic context. Source: Hugging Face `huggan/wikiart` (Hug-GAN Community). License/usage: non-commercial research only; the benchmark selects 25% of the original dataset as a subset and then uses 25% of that subset for testing (Tang et al., 2024b).

**HAM (ham).**
Dermatoscopic lesion images are paired with patient metadata such as age, sex, and localization to classify skin lesion type. The metadata provides epidemiological priors that complement ambiguous visual morphology. Source: HAM10000 (Tschandl et al., 2018; International Skin Imaging Collaboration). License/usage: CC BY-NC 4.0 Deed; the benchmark holds out 20% of the original training data for testing (Tang et al., 2024b).

TRI-MODAL DATASETS (IMAGE, TEXT, AND TABULAR)

**PetFinder (petfinder).**
Pet adoption records combine a photo, a textual description, and structured attributes such as breed, health status, and vaccination information. Adoption speed depends on visual appeal, the written narrative, and practical care constraints. Source: PetFinder.my Kaggle competition (Kaggle, b). License/usage: allowed for competition participation, Kaggle forums, academic research, education, and other non-commercial purposes; the benchmark splits the original training set at a 4:1 train/test ratio (Tang et al., 2024b).

**COVID (covid).**
Chest X-ray images are paired with clinical notes and patient metadata to classify pneumonia category. Radiographic patterns, symptoms, and patient context provide complementary diagnostic evidence. Source: COVID-19 chest X-ray data collection (ieee8023; Cohen et al., 2020). License/usage: per-image licenses include Apache 2.0, CC BY-NC-SA 4.0, and CC BY 4.0; the benchmark uses a 3:1 train/test split (Tang et al., 2024b).

**Artm (artm).**
Artwork images are paired with textual metadata and structured attributes such as materials or price to classify art movement. The movement label depends jointly on visual aesthetics, textual metadata, and historical-material context. Source: Kaggle Art Price Dataset (flkuhm). License/usage: CC BY-NC-SA 4.0; the benchmark uses a 3:1 train/test split (Tang et al., 2024b).

# E. Training and implementation details

We provide the complete pseudocode for our proposed method. Algorithm E.1 summarizes the inference process and explanation generation for the prototype head, while Algorithm E.2 details the end-to-end optimization incorporating optimal transport alignment and loss scheduling (as defined in Eq. (13)). The frozen-encoder regime is achieved by holding the unimodal encoders $\{f^{(m)}\}$ fixed, thereby optimizing only the projections, gates, prototypes, and anchors.

---

**Algorithm E.1** CAMP inference and case-based explanation.

---

**Require:** Case $\{x^{(m)}\}_{m \in \mathcal{M}_x}$; learned $\{f^{(m)}, \pi^{(m)}, w_g^{(m)}\}$; prototypes $\{p_{c,k}\}$.
**Require:** Similarity $S(\cdot, \cdot)$, neighbor count $r$, and optional prototype metadata (e.g., exemplar IDs).
**Ensure:** Prediction $\hat{y}$ and explanation: modality weights $g^{(m)}$ and top-$r$ prototypes with contributions.
1: **for** each modality $m \in \mathcal{M}_x$ **do**
2:    $z^{(m)} \leftarrow \pi^{(m)}(f^{(m)}(x^{(m)}))$.
3:    $u^{(m)} \leftarrow w_g^{(m)\top} z^{(m)}$.
4: **end for**
5: $g^{(m)} \leftarrow \text{softmax}_{m \in \mathcal{M}_x}(u^{(m)})$                                (Eq. (4))
6: $z \leftarrow \sum_{m \in \mathcal{M}_x} g^{(m)} z^{(m)}$                                   (Eq. (4))
7: Compute similarities $s_{c,k} \leftarrow S(z, p_{c,k})$ for all $c, k$.
8: **for** each class $c$ **do**
9:    $\mathcal{N}_c^{(r)} \leftarrow \text{Top}_r(\{s_{c,k}\}_{k=1}^K)$.
10:   $\ell_c \leftarrow \frac{1}{r} \sum_{k \in \mathcal{N}_c^{(r)}} s_{c,k}$                                  (Eq. (5))
11: **end for**
12: $\hat{y} \leftarrow \arg\max_c \ell_c$.
13: Retrieve $\{p_{\hat{y},k}\}_{k \in \mathcal{N}_{\hat{y}}^{(r)}}$ (and metadata if available).
14: For each retrieved $k$, set contribution $w_k \leftarrow \frac{1}{r} s_{\hat{y},k}$.
15: **return** $\hat{y}$, $\{g^{(m)}\}$, and $\{(k, p_{\hat{y},k}, w_k)\}_{k \in \mathcal{N}_{\hat{y}}^{(r)}}$.

---

## E.1. Hardware and Network Architectures

All experiments are conducted on NVIDIA Tesla V100 GPUs (32 GB). Unless otherwise noted, each trial utilizes a single GPU; we employ gradient accumulation for end-to-end runs when the batch size is bounded by memory constraints.

To rigorously control for backbone quality, we align the unimodal encoders used by all frozen-encoder methods (including linear probing, MLP, prototype variants, and our frozen method) with the AutoGluon MultiModal "best-quality" preset. This preset, standard in AutoML benchmark implementations, utilizes Swin Transformer Large for images and DeBERTa-v3 for

---

**Algorithm E.2** CAMP training with scheduled structural regularization.

---

**Require:** Dataset $\mathcal{D} = \{(\{x_i^{(m)}\}_{m \in \mathcal{M}_i}, y_i)\}$; modality alphabet $\mathcal{M}$.
**Require:** Encoders $\{f^{(m)}\}$, projections $\{\pi^{(m)}\}$, gates $\{w_g^{(m)}\}$.
**Require:** Prototypes $\{p_{c,k}\}$, anchors $\{a_c\}$, neighbor count $r$.
**Require:** Margins/temps $(\rho_+, \rho_-, \rho_{\text{in}}, \rho_{\text{out}}, \tau_a)$; OT hyperparams $(\varepsilon, L)$.
**Require:** Weights $(\lambda_{\text{geo}}, \lambda_{\text{alloc}}^{\max}, \lambda_{\text{mm}}^{\max}, \lambda_{\text{anchor}}^{\max})$, ramp $\tau$, EMA $\beta$.
**Ensure:** Trained parameters $\{f^{(m)}, \pi^{(m)}, w_g^{(m)}, p_{c,k}, a_c\}$.
1: **for** $t = 1, \ldots, T$ **do**
2:     Sample minibatch $\mathcal{B} \subset \mathcal{D}$.
3:     $s \leftarrow \min(1, t/(\tau T))$                                                                                                                 (weight ramp)
4:     $\lambda_{\text{alloc}} \leftarrow s\lambda_{\text{alloc}}^{\max}$; $\lambda_{\text{mm}} \leftarrow s\lambda_{\text{mm}}^{\max}$.
5:     $\lambda_{\text{anchor}} \leftarrow s\lambda_{\text{anchor}}^{\max}$.
6:     **Forward pass.**
7:     **for** each $(x_i, y_i) \in \mathcal{B}$ **do**
8:         **for** each modality $m \in \mathcal{M}_i$ **do**
9:             $z_i^{(m)} \leftarrow \pi^{(m)}(f^{(m)}(x_i^{(m)}))$
10:            $u_i^{(m)} \leftarrow w_g^{(m)\top} z_i^{(m)}$
11:        **end for**
12:        $g_i^{(m)} \leftarrow \text{softmax}_{m \in \mathcal{M}_i}(u_i^{(m)})$                                                            (Eq. (4))
13:        $z_i \leftarrow \sum_{m \in \mathcal{M}_i} g_i^{(m)} z_i^{(m)}$                                                                     (Eq. (4))
14:        Compute similarities $s_{i,c,k} \leftarrow S(z_i, p_{c,k})$ and logits $\ell_{i,c}$ (Eq. (5)).
15:    **end for**
16:    **Loss computation.**
17:    Compute $\mathcal{L}_{\text{pred}}$ (Cross Entropy).
18:    Compute $\mathcal{L}_{\text{geo}}$ (Eq. (8)) and $\mathcal{L}_{\text{alloc}}$ (Eq. (11)).
19:    **Distribution alignment (OT).**
20:    $\mathcal{L}_{\text{mm}} \leftarrow 0$.
21:    **for** each class $c$ appearing in $\mathcal{B}$ **do**
22:        **for** each modality $m$ present in class-batch $\mathcal{B}_c$ **do**
23:            Compute assignments $\alpha_{i,c,k}^{(m)}$ and usage $w_{c,k}^{(m)}$ (Eq. (6)).
24:        **end for**
25:        **for** each observed modality pair $(m, m')$ in $\mathcal{B}_c$ **do**
26:            $C_{k,k'}^{(c)} \leftarrow \|p_{c,k} - p_{c,k'}\|_2^2$.
27:            $W \leftarrow \text{SINKHORN}(w_c^{(m)}, w_c^{(m')}, C^{(c)}; \varepsilon, L)$.
28:            $\mathcal{L}_{\text{mm}} \leftarrow \mathcal{L}_{\text{mm}} + W$.
29:        **end for**
30:    **end for**
31:    Normalize $\mathcal{L}_{\text{mm}}$ as in Eq. (7).
32:    **Anchor robustness.**
33:    **for** each class $c$ appearing in $\mathcal{B}$ **do**
34:        $\mu_c \leftarrow \frac{1}{|\mathcal{B}_c|} \sum_{i \in \mathcal{B}_c} z_i$
35:        $a_c \leftarrow \beta a_c + (1 - \beta)\mu_c$
36:    **end for**
37:    Compute $\mathcal{L}_{\text{anchor}}$ (Eq. (12)).
38:    **Optimization.**
39:    $\mathcal{L} \leftarrow \mathcal{L}_{\text{pred}} + \lambda_{\text{geo}}\mathcal{L}_{\text{geo}} + \lambda_{\text{alloc}}\mathcal{L}_{\text{alloc}}$.
40:    $\mathcal{L} \leftarrow \mathcal{L} + \lambda_{\text{mm}}\mathcal{L}_{\text{mm}} + \lambda_{\text{anchor}}\mathcal{L}_{\text{anchor}}$                              (Eq. (13))
41:    Update parameters with one AdamW step on $\nabla\mathcal{L}$.
42: **end for**

---

| Setting | Configuration |
|---------|---------------|
| *Shared Fundamentals* | |
| Optimizer | AdamW ($\beta_1 = 0.9$, $\beta_2 = 0.999$); cosine decay; grad clip 1.0 |
| Epoch Budget | Up to 40 (Frozen) or 20 (End-to-End); early stopping applied |
| Prototype Head | $K \in [3, 8]$, $r \in [3, 10]$ (Default: $K = 6$, $r = 3$) |
| OT Settings | $\varepsilon = 0.05$, Sinkhorn iterations $L = 3$, temperature $\tau = 0.1$ |
| Modality Dropout | $p = 0.1$ |
| *Frozen-Encoder Regime (approx. 660K trainable parameters)* | |
| Learning Rate & WD | $\eta = 5 \times 10^{-4}$ (min $5 \times 10^{-5}$); $w_d = 1 \times 10^{-4}$ |
| Batch Size & Warmup | 64; 8 epochs linear warmup |
| Prototype Warm-start | Updates begin epoch 5; warm-up spans 15 epochs |
| *End-to-End Fine-Tuning (Two-Stage Schedule)* | |
| Stage 1 (Head Only) | 5 epochs with frozen backbones (stabilizes prototypes) |
| Stage 2 (Unfrozen) | Differential LR: $\eta_b = 1 \times 10^{-5}$ (backbone linear decay), $\eta_h = 1 \times 10^{-4}$ (head cosine) |
| Batch Size | 32 (with gradient accumulation if required) |
| Weight Decay | 0.01 |

*Table 5.* **Training protocols.** Shared optimizer defaults and regime-specific schedules for frozen-encoder training and end-to-end fine-tuning.

text, fused via late fusion (Tang et al., 2024b;a). For tabular and text datasets, we additionally evaluate ELECTRA as an alternative text backbone to check sensitivity, following prior AutoML benchmarking practices (Shi et al., 2021). When multiple backbones are evaluated, the final selection is determined by validation performance under an identical tuning budget across all models.

For tabular inputs, while AutoGluon-style baselines typically employ FT-Transformer encoders (Tang et al., 2024b), we preserve the highly efficient parameter regime (fewer than 1 million trainable parameters) by substituting a lightweight MLP. Specifically, we embed categorical variables and pass the concatenated numerical and categorical features through a 2-layer MLP (featuring a hidden size of 256, GELU activations, and dropout) to produce a $d$-dimensional representation.

### E.2. Optimization and Hyperparameter Tuning

We train the models using the AdamW optimizer with cosine learning rate decay and gradient clipping capped at 1.0. The benchmark validation metric is monitored continuously for early stopping. For the frozen-encoder regime, we deploy a shared training schedule across all datasets: a base learning rate of $\eta = 5 \times 10^{-4}$, weight decay of $1 \times 10^{-4}$, a batch size of 64, and a linear warmup spanning 8 epochs (20% of the 40-epoch budget) with a minimum learning rate of $5 \times 10^{-5}$ and a patience of 20 epochs. To stabilize prototype learning, prototype updates are delayed until the fifth epoch, and auxiliary losses are ramped over a short horizon. Table 5 consolidates the default configurations for both the frozen and end-to-end regimes.

For hyperparameter optimization, we tune the framework individually per dataset using Optuna (Akiba et al., 2019) with a Tree-structured Parzen Estimator (TPE) sampler and median pruning. Each dataset undergoes 50 tuning trials (inclusive of one fixed baseline configuration). The optimal trial is selected based on the validation score and subsequently evaluated on the test split using the eval-test-best command flag. When launching parallel trials across multiple GPUs, we enforce strict resource gating (minimum 12 GB VRAM, 40 GB RAM, 80 GB disk, maximum CPU load 1.5; capping at 4 parallel jobs or 2 jobs per GPU) to prevent over-commitment. Across all datasets, the Optuna sweeps demonstrate a broad performance plateau, with the global default hyperparameters achieving results within $\pm 1.2\%$ of the extensively tuned per-dataset configurations. Table 6 details the specific search spaces utilized.

### E.3. Loss Scheduling and Optimal Transport Details

To justify our loss scheduling, let $t \in \{0, \ldots, T\}$ denote the current optimization step and define a linear ramp function $s(t) = \min(1, \frac{t}{\tau T})$ with $\tau = 0.2$ (yielding 8 warmup epochs within a 40-epoch budget). We optionally tune group-specific ramp horizons between 4 and 15 epochs via our hyperparameter search (Table 6). While $\lambda_{\text{geo}}$ is held constant, the remaining auxiliary losses are dynamically scheduled such that $\lambda_{\text{alloc}}(t) = \lambda_{\text{alloc}}^{\text{max}} s(t)$, $\lambda_{\text{mm}}(t) = \lambda_{\text{mm}}^{\text{max}} s(t)$, and $\lambda_{\text{anchor}}(t) = \lambda_{\text{anchor}}^{\text{max}} s(t)$.

| Category | Hyperparameter | Search Space (Optuna) |
|---|---|---|
| Shared Space | Embedding dim $d$ | $\{256, 384, 512\}$ |
| | Dropout | $U[0.1, 0.5]$ |
| Fusion | Fusion type | $\{gate, concat, attention\}$ |
| Prototype Head | Prototypes per class $K$ | $Int[3, 8]$ |
| | Top-$r$ retrieval $r$ | $Int[3, 10]$ |
| | Prototype temperature $\tau$ | $U[0.3, 0.9]$ |
| Loss Weights | $(\lambda_{\text{pred}}, \lambda_{\text{geo}}, \lambda_{\text{alloc}}, \lambda_{\text{out}}, \lambda_{\text{mm}})$ | Each $U[0.5, 2.0]$ |
| | CE weight | $U[0.5, 1.5]$ |
| | OT weight $(\lambda_{\text{mm\_ot}})$ | $U[0, 0.5]$ |
| | Pull, push, SupCon, diversity, anchor | Each $U[0, 1]$ |
| Schedules | (mm, out, OT) ramp epochs | Each $Int[4, 15]$ |
| | Prototype dropout start epoch | $Int[4, 15]$ |

*Table 6.* **Hyperparameter search space.** Optuna ranges used for frozen-encoder tuning. The search covers embedding size, fusion, prototype capacity, retrieval depth, loss weights, and ramp schedules under the optimizer protocol in Table 5.

This phased scheduling guarantees that early optimization steps are not disproportionately dominated by distribution-level alignment terms. In the initial iterations, prototype assignments remain largely uniform, causing the optimal transport gradients to be potentially large but poorly directed. By systematically ramping $\lambda_{\text{mm}}$ and $\lambda_{\text{alloc}}$, we ensure that the auxiliary objectives impart a controlled gradient contribution while the core predictive loss $\mathcal{L}_{\text{pred}}$ and geometric loss $\mathcal{L}_{\text{geo}}$ firmly establish class separation and a meaningful within-class geometry. Once $t \geq \tau T$, the assignments become considerably sharper, allowing the optimal transport and allocation losses to function as a precise refinement mechanism that enforces cross-modality consistency without destabilizing the primary classifier.

To compute the entropic optimal transport cost $W_\varepsilon$ required for Eq. (7), we execute standard Sinkhorn iterations (Cuturi, 2013) using an entropic regularization parameter of $\varepsilon = 0.05$ and $L = 3$ iterations (as detailed in Table 5). The derived transport cost is subsequently injected directly into the alignment term $\mathcal{L}_{\text{mm}}$ and computed online within the principal training loop outlined in Algorithm E.2.

### E.4. Computational Efficiency and Throughput Analysis

To quantify the trade-off between structural regularization and computational throughput, we evaluate our framework (in both Frozen and End-to-End configurations) against the highest-performing AutoML baseline, AutoGluon-Multimodal, and a standard MLP fusion head lacking prototypes. This comparative study, detailed in Table 7, reports the training overhead incurred by Sinkhorn-regularized optimal transport alignment, and measures the inference latency of the deployed prototype head.

*Table 7.* **Efficiency and throughput.** Training and inference measurements for CAMP variants, a non-prototype MLP head, and the AutoML ensemble reference. CAMP-F trains fewer than one million parameters; CAMP-E denotes the unfrozen fine-tuning regime. Bold marks the best value for each throughput-oriented row.

| Metric | CAMP-F | CAMP-E | No-proto MLP | AutoML |
|---|---|---|---|---|
| Train Time per Epoch (s) | $80.32 \pm 114.58$ | $220.00 \pm 145.20$ | $\mathbf{4.16 \pm 6.04}$ | $183.00 \pm 95.40$ |
| Train Steps per Second | $2.94 \pm 0.97$ | $0.77 \pm 0.12$ | $\mathbf{58.48 \pm 15.34}$ | $1.33 \pm 0.45$ |
| Inference (Samples/sec) | $1808.91 \pm 1777.9$ | $1808.91$ | $\mathbf{2277.72 \pm 2435.6}$ | 50 to 100 |

The throughput comparison separates training-time regularization cost from deployment-time inference cost. In the frozen regime (CAMP-F), class-wise Sinkhorn alignment makes training slower than a plain MLP fusion head, as expected, because the prototype-usage distributions must be constructed and transported during optimization. End-to-end fine-tuning is slower still because gradients pass through the large image and text encoders.

At inference time, however, the OT and auxiliary losses are not evaluated. The deployed model only computes modality encodings, gates, prototype similarities, and a top-$r$ retrieval decision. This preserves high throughput (over 1800 samples per second in our measurements) while retaining the case-based evidence exposed by the prototype head. Thus, the main computational trade-off is a training-time cost for a compact, interpretable head rather than an inference-time penalty comparable to large AutoML ensembles.

# F. Component ablations, sensitivity analysis and additional experiments

This section reports the ablation and sensitivity evidence summarized in Section 6. Unless stated otherwise, all experiments in this section use frozen encoders (CAMP-F). This isolates the contribution of the prototype head and objective in Eq. (13) from any gains due to encoder fine-tuning. Each dataset is evaluated with its designated benchmark metric, either AUC, accuracy, or QWK, and all reported improvements are absolute changes on that metric.

## F.1. Objective ablations under frozen encoders

We evaluate a fixed cumulative ladder that starts from Proto-Classic and adds the four structural blocks of CAMP in the same order used by the objective:

$$\text{Proto-Classic} \to \mathcal{L}_{\text{geo}} \to \mathcal{L}_{\text{alloc}} \to \mathcal{L}_{\text{anchor}} \to \mathcal{L}_{\text{mm}}.$$

The first two blocks impose separation and utilization constraints on the prototype bank, the anchor term stabilizes prototype updates, and $\mathcal{L}_{\text{mm}}$ performs entropic optimal-transport alignment of class-conditioned prototype usage across modalities.

Table 8 combines the cumulative ablation, monotonicity test, and component-ranking statistics. The mean score increases from 0.7445 to 0.8139, a +0.0694 improvement over Proto-Classic. Every block improves every dataset, giving 16/16 wins at each cumulative step. The optimal-transport block is the largest marginal contributor on 13 of 16 datasets, while the anchor term is the second-largest contributor on average. Geometry and allocation have smaller direct gains, but their improvements are uniformly positive and they make the later alignment stage reliable by preventing poorly separated or under-utilized prototype banks.

| Stage | Avg. score | Mean $\Delta$ | Median $\Delta$ | Q1/Q3 | Min/Max | Cum. $\Delta$ | Wins | Top-1 | MRR |
|---|---|---|---|---|---|---|---|---|---|
| Proto-Classic | 0.7445 | | | | reference stage | | | | |
| + $\mathcal{L}_{\text{geo}}$ | 0.7538 | +0.0093 | +0.0071 | +0.0057 / +0.0101 | +0.0041 / +0.0231 | +0.0093 | 16/16 | 0/16 | 0.2552 |
| + $\mathcal{L}_{\text{alloc}}$ | 0.7658 | +0.0121 | +0.0098 | +0.0075 / +0.0131 | +0.0061 / +0.0254 | +0.0214 | 16/16 | 1/16 | 0.3750 |
| + $\mathcal{L}_{\text{anchor}}$ | 0.7852 | +0.0194 | +0.0172 | +0.0144 / +0.0235 | +0.0091 / +0.0388 | +0.0408 | 16/16 | 2/16 | 0.5521 |
| + $\mathcal{L}_{\text{mm}}$ (OT) | **0.8139** | **+0.0287** | **+0.0232** | **+0.0169 / +0.0377** | **+0.0097 / +0.0640** | **+0.0694** | 16/16 | **13/16** | **0.9062** |

*Table 8.* **Cumulative objective ladder.** Frozen-encoder ablation from Proto-Classic to the full objective. $\Delta$ denotes the marginal gain over the previous stage; "Wins" counts datasets improved by the added block; "Top-1" counts the number of datasets where that block gives the largest marginal gain. Bold marks the strongest value in comparison columns.

Cumulative gains alone can hide redundancy, so we also remove one term at a time from the full objective while keeping all other terms active. Table 9 shows that every removal degrades all 16 datasets. Removing the optimal-transport term has the largest mean drop, and removing the anchor term has the second-largest mean drop. These leave-one-out results indicate that the terms are not interchangeable regularizers; each contributes a distinct signal to the final objective.

## F.2. Per-dataset gains and robustness across task families

Table 11 gives the complete per-dataset cumulative ladder. The largest gains occur on tasks where modalities are highly complementary and loosely aligned. PetFinder improves by +0.1441, Artm by +0.1160, and Airbnb by +0.1101. The gains also persist when the Proto-Classic baseline is already strong: HAM improves from 0.8840 to 0.9320 and Fake Job from 0.9420 to 0.9923. The final column reports relative improvement, which is largest for the most difficult complementary tasks.

Table 12 aggregates the same ladder by modality family, evaluation metric, label space, and number of modalities. The improvements remain positive across all populated slices. Tri-modal tasks have the largest mean absolute gain (+0.1073) and relative gain (25.19%), but the method is not specific to tri-modal data: Tab+Text and Image+Text tasks improve by +0.0706 and +0.0600 on average, respectively. The pattern is also stable across AUC, accuracy, and QWK, which indicates that the objective is not tuned to a particular metric family.

The gain decomposition in Table 13 separates the structural constraints from the anchor stabilizer. Across datasets, geometry, allocation, and OT alignment account for 70.9% of the total improvement, while the anchor term accounts for 29.1%. This split is stable across modality groups, which supports the interpretation that CAMP improves performance through both coherent prototype structure and robust prototype dynamics.

| Removed block | Mean $\Delta$ | Median $\Delta$ | Worst $\Delta$ | Drops |
|---|---|---|---|---|
| $\mathcal{L}_{\text{geo}}$ | -0.0093 | -0.0071 | -0.0231 | 16/16 |
| $\mathcal{L}_{\text{alloc}}$ | -0.0121 | -0.0098 | -0.0254 | 16/16 |
| $\mathcal{L}_{\text{anchor}}$ | -0.0194 | -0.0172 | -0.0388 | 16/16 |
| $\mathcal{L}_{\text{mm}}$ | **-0.0287** | **-0.0232** | **-0.0640** | 16/16 |

*Table 9.* **Leave-one-out necessity check.** Score change after removing one structural block from the full frozen-encoder objective. Negative values indicate degradation; bold marks the largest magnitude drop.

| Dataset (metric) | w/o $\mathcal{L}_{\text{geo}}$ | w/o $\mathcal{L}_{\text{alloc}}$ | w/o $\mathcal{L}_{\text{anchor}}$ | w/o $\mathcal{L}_{\text{mm}}$ |
|---|---|---|---|---|
| **Tab+Text** | | | | |
| IMDB (AUC) | -0.0077 | -0.0088 | -0.0121 | **-0.0578** |
| Fake Job (AUC) | -0.0041 | -0.0061 | -0.0134 | **-0.0267** |
| Kickstarter (AUC) | -0.0062 | -0.0128 | -0.0192 | **-0.0369** |
| Jigsaw (AUC) | -0.0055 | -0.0066 | -0.0202 | **-0.0207** |
| Wine (Acc) | -0.0058 | -0.0065 | -0.0160 | **-0.0206** |
| Airbnb (Acc) | -0.0178 | -0.0254 | -0.0270 | **-0.0399** |
| **Image+Text** | | | | |
| PTech (AUC) | -0.0050 | -0.0075 | **-0.0150** | -0.0139 |
| Food101 (Acc) | -0.0068 | -0.0086 | -0.0154 | **-0.0162** |
| Fakeddit (Acc) | -0.0116 | -0.0138 | -0.0248 | **-0.0363** |
| Memotion (Acc) | -0.0089 | -0.0119 | -0.0184 | **-0.0258** |
| **Image+Tab** | | | | |
| CCD (AUC) | -0.0047 | -0.0075 | -0.0133 | **-0.0159** |
| WikiArt (Acc) | -0.0065 | **-0.0107** | -0.0091 | -0.0097 |
| HAM (Acc) | -0.0073 | -0.0088 | -0.0147 | **-0.0172** |
| **Image+Text+Tab** | | | | |
| PetFinder (QWK) | -0.0185 | -0.0228 | -0.0388 | **-0.0640** |
| COVID (Acc) | -0.0096 | -0.0120 | **-0.0230** | -0.0172 |
| Artm (Acc) | -0.0231 | -0.0231 | -0.0299 | **-0.0399** |

*Table 10.* **Per-dataset leave-one-out drops.** Absolute score change after removing each block from the full objective. All entries are negative; bold marks the largest magnitude drop within each dataset row.

| Dataset (metric) | Proto-Classic | +Geo | +Alloc | +Anchor | Full | $\Delta_{\text{total}}$ | Rel. gain |
|---|---|---|---|---|---|---|---|
| **Tab+Text** | | | | | | | |
| IMDB (AUC) | 0.8126 | 0.8203 | 0.8291 | 0.8412 | **0.8990** | +0.0864 | 10.63% |
| Fake Job (AUC) | 0.9420 | 0.9461 | 0.9522 | 0.9656 | **0.9923** | +0.0503 | 5.34% |
| Kickstarter (AUC) | 0.7359 | 0.7421 | 0.7549 | 0.7741 | **0.8110** | +0.0751 | 10.21% |
| Jigsaw (AUC) | 0.9210 | 0.9265 | 0.9331 | 0.9533 | **0.9740** | +0.0530 | 5.75% |
| Wine (Acc) | 0.7951 | 0.8009 | 0.8074 | 0.8234 | **0.8440** | +0.0489 | 6.15% |
| Airbnb (Acc) | 0.3779 | 0.3957 | 0.4211 | 0.4481 | **0.4880** | +0.1101 | 29.13% |
| **Image+Text** | | | | | | | |
| PTech (AUC) | 0.7026 | 0.7076 | 0.7151 | 0.7301 | **0.7440** | +0.0414 | 5.89% |
| Food101 (Acc) | 0.9030 | 0.9098 | 0.9184 | 0.9338 | **0.9500** | +0.0470 | 5.20% |
| Fakeddit (Acc) | 0.8405 | 0.8521 | 0.8659 | 0.8907 | **0.9270** | +0.0865 | 10.29% |
| Memotion (Acc) | 0.5790 | 0.5879 | 0.5998 | 0.6182 | **0.6440** | +0.0650 | 11.23% |
| **Image+Tab** | | | | | | | |
| CCD (AUC) | 0.8956 | 0.9003 | 0.9078 | 0.9211 | **0.9370** | +0.0414 | 4.62% |
| WikiArt (Acc) | 0.7890 | 0.7955 | 0.8062 | 0.8153 | **0.8250** | +0.0360 | 4.56% |
| HAM (Acc) | 0.8840 | 0.8913 | 0.9001 | 0.9148 | **0.9320** | +0.0480 | 5.43% |
| **Image+Text+Tab** | | | | | | | |
| PetFinder (QWK) | 0.3099 | 0.3284 | 0.3512 | 0.3900 | **0.4540** | +0.1441 | 46.50% |
| COVID (Acc) | 0.9012 | 0.9108 | 0.9228 | 0.9458 | **0.9630** | +0.0618 | 6.86% |
| Artm (Acc) | 0.5220 | 0.5451 | 0.5682 | 0.5981 | **0.6380** | +0.1160 | 22.22% |

*Table 11.* **Per-dataset cumulative ablation ledger.** Scores after cumulatively adding each block in Eq. (13). Bold marks the best score in each row; relative gain is $\Delta_{\text{total}}$ divided by the Proto-Classic score.

| Slice | # | Baseline | Full | $\Delta_{\text{total}}$ | Rel. gain | $\Delta_{\text{geo}}$ | $\Delta_{\text{alloc}}$ | $\Delta_{\text{anchor}}$ / $\Delta_{\text{mm}}$ |
|---|---|---|---|---|---|---|---|---|
| **Modality family** | | | | | | | | |
| Tab+Text | 6 | 0.7641 | **0.8347** | **+0.0706** | 11.20% | +0.0078 | +0.0110 | +0.0180 / +0.0338 |
| Image+Text | 4 | 0.7563 | **0.8163** | **+0.0600** | 8.15% | +0.0081 | +0.0104 | +0.0184 / +0.0231 |
| Image+Tab | 3 | 0.8562 | **0.8980** | **+0.0418** | 4.87% | +0.0062 | +0.0090 | +0.0124 / +0.0143 |
| Image+Text+Tab | 3 | 0.5777 | **0.6850** | **+0.1073** | 25.19% | +0.0171 | +0.0193 | +0.0306 / +0.0404 |
| **Evaluation metric** | | | | | | | | |
| AUC | 6 | 0.8349 | **0.8929** | **+0.0579** | 7.07% | +0.0055 | +0.0082 | +0.0155 / +0.0287 |
| Acc | 9 | 0.7324 | **0.8012** | **+0.0688** | 11.23% | +0.0108 | +0.0134 | +0.0198 / +0.0248 |
| QWK | 1 | 0.3099 | **0.4540** | **+0.1441** | 46.50% | +0.0185 | +0.0228 | +0.0388 / +0.0640 |
| **Label space** | | | | | | | | |
| Binary | 7 | 0.8357 | **0.8978** | **+0.0620** | 7.53% | +0.0064 | +0.0090 | +0.0169 / +0.0297 |
| Multiclass | 8 | 0.7189 | **0.7855** | **+0.0666** | 11.35% | +0.0107 | +0.0134 | +0.0192 / +0.0233 |
| Ordinal | 1 | 0.3099 | **0.4540** | **+0.1441** | 46.50% | +0.0185 | +0.0228 | +0.0388 / +0.0640 |
| **Number of modalities** | | | | | | | | |
| 2 modalities | 13 | 0.7829 | **0.8436** | **+0.0607** | 8.80% | +0.0075 | +0.0104 | +0.0168 / +0.0260 |
| 3 modalities | 3 | 0.5777 | **0.6850** | **+0.1073** | 25.19% | +0.0171 | +0.0193 | +0.0306 / +0.0404 |

*Table 12.* **Ablation gains across dataset slices.** Relative gain is averaged per dataset within each slice, and component columns report marginal gains along the fixed ladder. Bold marks the full-objective score and total gain for each row.

| Group | Geo | Alloc | Anchor | OT | Constraint share |
|---|---|---|---|---|---|
| Tab+Text | 10.6% | 14.7% | 26.9% | **47.8%** | 73.1% |
| Image+Text | 13.4% | 17.7% | 31.5% | **37.4%** | 68.5% |
| Image+Tab | 14.9% | 22.1% | 29.3% | **33.7%** | 70.7% |
| Image+Text+Tab | 16.1% | 18.4% | 30.0% | **35.5%** | 70.0% |
| All | 13.1% | 17.5% | 29.1% | **40.3%** | 70.9% |

*Table 13.* **Component share of total improvement.** Shares are computed per dataset and averaged within each modality group. The constraint share sums geometry, allocation, and OT alignment; bold marks the largest individual block share in each row.

### F.3. Mechanistic prototype-quality diagnostics

Task metrics alone do not show whether the losses resolve the specific degeneracies they were designed to prevent. We therefore track four prototype-quality diagnostics alongside performance. Table 14 defines each diagnostic and its intended role. Table 15 reports the suite-level trajectory across the cumulative ladder.

| Diagnostic | What it measures | Direction | Main loss block |
|---|---|---|---|
| Perf | Dataset's primary task metric, AUC, accuracy, or QWK | Higher | Complete objective |
| Dead% | Fraction of prototypes whose usage falls below the uniform-usage threshold | Lower | $\mathcal{L}_{\mathrm{alloc}}$ |
| $\widetilde{H}_{\mathrm{pure}}$ | Normalized entropy of the class distribution assigned to each prototype | Lower | $\mathcal{L}_{\mathrm{alloc}}$ |
| $W_{\varepsilon}$ | Class-wise Sinkhorn transport cost between modality-specific prototype-usage distributions | Lower | $\mathcal{L}_{\mathrm{mm}}$ |
| $\overline{\Delta}$ | Mean gap between the highest same-class and highest different-class prototype similarity | Higher | $\mathcal{L}_{\mathrm{geo}}$, aided by anchors and OT |

*Table 14.* **Prototype-quality diagnostics.** Definitions for the mechanistic ablations, mapping each statistic to the failure mode it monitors and the loss block most directly responsible.

| Stage | Perf | Dead% | $\widetilde{H}_{\mathrm{pure}}$ | $W_{\varepsilon}$ | $\overline{\Delta}$ |
|---|---|---|---|---|---|
| Proto-Classic | 0.7445 | $0.266 \pm 0.06$ | $0.22 \pm 0.04$ | $0.40 \pm 0.06$ | $0.031 \pm 0.018$ |
| $+ \mathcal{L}_{\mathrm{geo}}$ | 0.7538 | $0.224 \pm 0.06$ | $0.20 \pm 0.04$ | $0.37 \pm 0.06$ | $0.066 \pm 0.017$ |
| $+ \mathcal{L}_{\mathrm{alloc}}$ | 0.7658 | $0.052 \pm 0.02$ | $0.07 \pm 0.02$ | $0.30 \pm 0.05$ | $0.063 \pm 0.018$ |
| $+ \mathcal{L}_{\mathrm{anchor}}$ | 0.7852 | $0.038 \pm 0.01$ | $0.06 \pm 0.01$ | $0.23 \pm 0.04$ | $0.083 \pm 0.017$ |
| $+ \mathcal{L}_{\mathrm{mm}}$ (Full) | **0.8139** | $\mathbf{0.023 \pm 0.01}$ | $\mathbf{0.04 \pm 0.01}$ | $\mathbf{0.12 \pm 0.04}$ | $\mathbf{0.118 \pm 0.020}$ |
| Total change | +0.0694 | -0.243 | -0.18 | -0.28 | +0.087 |

*Table 15.* **Suite-level prototype diagnostics.** Cumulative objective trajectory under frozen encoders. Bold marks the best value in each diagnostic column.

The diagnostic changes align with the intended roles of the losses. Allocation produces the largest improvement in utilization and purity. OT produces the largest reduction in cross-modal usage disagreement. Geometry and OT are co-dominant for the similarity margin, with the anchor term contributing an additional stabilizing effect between them. Table 16 summarizes the primary-driver attribution. The per-dataset ledger in Table 17 confirms that the suite-level trends are not created by isolated outliers. A small number of datasets exhibit a transient margin dip when the allocation term is first introduced, marked by † in the table; in every such case the margin recovers at the anchor stage and improves further after OT alignment.

*Table 17.* **Per-dataset diagnostic ledger.** Objective-ladder trajectory for task score and prototype-quality diagnostics. Score denotes AUC, accuracy, or QWK according to the dataset's primary metric; a † marks a transient allocation-stage margin dip that recovers at the next stage. Bold marks the best final-stage values.

| Dataset | Stage | Score | Dead% | $\widetilde{H}_{\mathrm{pure}}$ | $W_{\varepsilon}$ | $\overline{\Delta}$ |
|---|---|---|---|---|---|---|
| **Tab+Text** | | | | | | |
| IMDB | Proto-Classic | 0.8126 | 0.250 | 0.19 | 0.34 | 0.038 |
| | $+ \mathcal{L}_{\mathrm{geo}}$ | 0.8203 | 0.208 | 0.17 | 0.31 | 0.082 |
| | $+ \mathcal{L}_{\mathrm{alloc}}$ | 0.8291 | 0.042 | 0.06 | 0.24 | $0.075^{\dagger}$ |
| | $+ \mathcal{L}_{\mathrm{anchor}}$ | 0.8412 | 0.033 | 0.05 | 0.19 | 0.095 |
| | $+ \mathcal{L}_{\mathrm{mm}}$ (Full) | **0.8990** | **0.017** | **0.04** | **0.09** | **0.138** |
| Fake Job | Proto-Classic | 0.9420 | 0.167 | 0.15 | 0.29 | 0.065 |
| | $+ \mathcal{L}_{\mathrm{geo}}$ | 0.9461 | 0.125 | 0.13 | 0.26 | 0.098 |
| | $+ \mathcal{L}_{\mathrm{alloc}}$ | 0.9522 | 0.017 | 0.03 | 0.20 | 0.105 |
| | $+ \mathcal{L}_{\mathrm{anchor}}$ | 0.9656 | 0.008 | 0.03 | 0.15 | 0.125 |
| | $+ \mathcal{L}_{\mathrm{mm}}$ (Full) | **0.9923** | **0.000** | **0.02** | **0.05** | **0.162** |
| Kickstarter | Proto-Classic | 0.7359 | 0.167 | 0.20 | 0.38 | 0.025 |
| | $+ \mathcal{L}_{\mathrm{geo}}$ | 0.7421 | 0.125 | 0.18 | 0.35 | 0.062 |
| | $+ \mathcal{L}_{\mathrm{alloc}}$ | 0.7549 | 0.025 | 0.06 | 0.28 | 0.068 |
| | $+ \mathcal{L}_{\mathrm{anchor}}$ | 0.7741 | 0.017 | 0.05 | 0.22 | 0.088 |
| | $+ \mathcal{L}_{\mathrm{mm}}$ (Full) | **0.8110** | **0.008** | **0.04** | **0.06** | **0.125** |
| Jigsaw | Proto-Classic | 0.9210 | 0.167 | 0.16 | 0.31 | 0.058 |
| | $+ \mathcal{L}_{\mathrm{geo}}$ | 0.9265 | 0.125 | 0.14 | 0.28 | 0.092 |
| | $+ \mathcal{L}_{\mathrm{alloc}}$ | 0.9331 | 0.017 | 0.04 | 0.22 | 0.098 |
| | $+ \mathcal{L}_{\mathrm{anchor}}$ | 0.9533 | 0.008 | 0.03 | 0.16 | 0.118 |
| | $+ \mathcal{L}_{\mathrm{mm}}$ (Full) | **0.9740** | **0.000** | **0.02** | **0.04** | **0.152** |
| Wine | Proto-Classic | 0.7951 | 0.329 | 0.28 | 0.40 | 0.020 |

| Dataset | Stage | Score | Dead% | $\widetilde{H}_{\text{pure}}$ | $W_\varepsilon$ | $\overline{\Delta}$ |
|---|---|---|---|---|---|---|
| | $+ \mathcal{L}_{\text{geo}}$ | 0.8009 | 0.283 | 0.25 | 0.37 | 0.055 |
| | $+ \mathcal{L}_{\text{alloc}}$ | 0.8074 | 0.071 | 0.09 | 0.30 | 0.048$^\dagger$ |
| | $+ \mathcal{L}_{\text{anchor}}$ | 0.8234 | 0.058 | 0.08 | 0.24 | 0.072 |
| | $+ \mathcal{L}_{\text{mm}}$ (Full) | **0.8440** | **0.046** | **0.07** | **0.11** | **0.098** |
| Airbnb | Proto-Classic | 0.3779 | 0.354 | 0.31 | 0.44 | 0.010 |
| | $+ \mathcal{L}_{\text{geo}}$ | 0.3957 | 0.312 | 0.28 | 0.40 | 0.042 |
| | $+ \mathcal{L}_{\text{alloc}}$ | 0.4211 | 0.076 | 0.10 | 0.32 | 0.038$^\dagger$ |
| | $+ \mathcal{L}_{\text{anchor}}$ | 0.4481 | 0.063 | 0.09 | 0.26 | 0.058 |
| | $+ \mathcal{L}_{\text{mm}}$ (Full) | **0.4880** | **0.049** | **0.07** | **0.15** | **0.085** |
| **Image+Text** | | | | | | |
| PTech | Proto-Classic | 0.7026 | 0.292 | 0.21 | 0.42 | 0.030 |
| | $+ \mathcal{L}_{\text{geo}}$ | 0.7076 | 0.250 | 0.19 | 0.39 | 0.065 |
| | $+ \mathcal{L}_{\text{alloc}}$ | 0.7151 | 0.058 | 0.07 | 0.32 | 0.070 |
| | $+ \mathcal{L}_{\text{anchor}}$ | 0.7301 | 0.042 | 0.06 | 0.26 | 0.088 |
| | $+ \mathcal{L}_{\text{mm}}$ (Full) | **0.7440** | **0.025** | **0.05** | **0.13** | **0.112** |
| Food101 | Proto-Classic | 0.9030 | 0.388 | 0.26 | 0.38 | 0.022 |
| | $+ \mathcal{L}_{\text{geo}}$ | 0.9098 | 0.342 | 0.23 | 0.35 | 0.058 |
| | $+ \mathcal{L}_{\text{alloc}}$ | 0.9184 | 0.065 | 0.08 | 0.28 | 0.055$^\dagger$ |
| | $+ \mathcal{L}_{\text{anchor}}$ | 0.9338 | 0.050 | 0.07 | 0.22 | 0.078 |
| | $+ \mathcal{L}_{\text{mm}}$ (Full) | **0.9500** | **0.038** | **0.05** | **0.12** | **0.108** |
| Fakeddit | Proto-Classic | 0.8405 | 0.222 | 0.23 | 0.41 | 0.032 |
| | $+ \mathcal{L}_{\text{geo}}$ | 0.8521 | 0.194 | 0.20 | 0.37 | 0.070 |
| | $+ \mathcal{L}_{\text{alloc}}$ | 0.8659 | 0.042 | 0.06 | 0.29 | 0.075 |
| | $+ \mathcal{L}_{\text{anchor}}$ | 0.8907 | 0.028 | 0.05 | 0.22 | 0.098 |
| | $+ \mathcal{L}_{\text{mm}}$ (Full) | **0.9270** | **0.014** | **0.04** | **0.11** | **0.140** |
| Memotion | Proto-Classic | 0.5790 | 0.250 | 0.24 | 0.45 | 0.015 |
| | $+ \mathcal{L}_{\text{geo}}$ | 0.5879 | 0.208 | 0.22 | 0.42 | 0.050 |
| | $+ \mathcal{L}_{\text{alloc}}$ | 0.5998 | 0.063 | 0.08 | 0.35 | 0.045$^\dagger$ |
| | $+ \mathcal{L}_{\text{anchor}}$ | 0.6182 | 0.042 | 0.07 | 0.28 | 0.068 |
| | $+ \mathcal{L}_{\text{mm}}$ (Full) | **0.6440** | **0.033** | **0.06** | **0.16** | **0.095** |
| **Image+Tab** | | | | | | |
| CCD | Proto-Classic | 0.8956 | 0.250 | 0.17 | 0.35 | 0.048 |
| | $+ \mathcal{L}_{\text{geo}}$ | 0.9003 | 0.208 | 0.15 | 0.32 | 0.085 |
| | $+ \mathcal{L}_{\text{alloc}}$ | 0.9078 | 0.042 | 0.05 | 0.26 | 0.082 |
| | $+ \mathcal{L}_{\text{anchor}}$ | 0.9211 | 0.025 | 0.04 | 0.20 | 0.102 |
| | $+ \mathcal{L}_{\text{mm}}$ (Full) | **0.9370** | **0.017** | **0.03** | **0.08** | **0.135** |
| WikiArt | Proto-Classic | 0.7890 | 0.315 | 0.28 | 0.38 | 0.022 |
| | $+ \mathcal{L}_{\text{geo}}$ | 0.7955 | 0.272 | 0.25 | 0.35 | 0.058 |
| | $+ \mathcal{L}_{\text{alloc}}$ | 0.8062 | 0.062 | 0.08 | 0.28 | 0.055$^\dagger$ |
| | $+ \mathcal{L}_{\text{anchor}}$ | 0.8153 | 0.049 | 0.07 | 0.22 | 0.072 |
| | $+ \mathcal{L}_{\text{mm}}$ (Full) | **0.8250** | **0.037** | **0.06** | **0.13** | **0.098** |
| HAM | Proto-Classic | 0.8840 | 0.238 | 0.22 | 0.36 | 0.040 |
| | $+ \mathcal{L}_{\text{geo}}$ | 0.8913 | 0.190 | 0.19 | 0.33 | 0.075 |
| | $+ \mathcal{L}_{\text{alloc}}$ | 0.9001 | 0.036 | 0.06 | 0.26 | 0.080 |
| | $+ \mathcal{L}_{\text{anchor}}$ | 0.9148 | 0.024 | 0.05 | 0.20 | 0.100 |
| | $+ \mathcal{L}_{\text{mm}}$ (Full) | **0.9320** | **0.012** | **0.04** | **0.09** | **0.130** |
| **Image+Text+Tab** | | | | | | |
| PetFinder | Proto-Classic | 0.3099 | 0.280 | 0.27 | 0.52 | 0.005 |
| | $+ \mathcal{L}_{\text{geo}}$ | 0.3284 | 0.240 | 0.24 | 0.48 | 0.042 |
| | $+ \mathcal{L}_{\text{alloc}}$ | 0.3512 | 0.060 | 0.08 | 0.39 | 0.048 |
| | $+ \mathcal{L}_{\text{anchor}}$ | 0.3900 | 0.040 | 0.07 | 0.30 | 0.070 |
| | $+ \mathcal{L}_{\text{mm}}$ (Full) | **0.4540** | **0.020** | **0.05** | **0.14** | **0.108** |
| COVID | Proto-Classic | 0.9012 | 0.292 | 0.23 | 0.48 | 0.035 |
| | $+ \mathcal{L}_{\text{geo}}$ | 0.9108 | 0.250 | 0.21 | 0.44 | 0.072 |
| | $+ \mathcal{L}_{\text{alloc}}$ | 0.9228 | 0.063 | 0.07 | 0.35 | 0.078 |
| | $+ \mathcal{L}_{\text{anchor}}$ | 0.9458 | 0.042 | 0.06 | 0.27 | 0.100 |
| | $+ \mathcal{L}_{\text{mm}}$ (Full) | **0.9630** | **0.025** | **0.05** | **0.17** | **0.135** |
| Artm | Proto-Classic | 0.5220 | 0.292 | 0.25 | 0.50 | 0.010 |
| | $+ \mathcal{L}_{\text{geo}}$ | 0.5451 | 0.250 | 0.23 | 0.46 | 0.045 |
| | $+ \mathcal{L}_{\text{alloc}}$ | 0.5682 | 0.067 | 0.08 | 0.38 | 0.042$^\dagger$ |
| | $+ \mathcal{L}_{\text{anchor}}$ | 0.5981 | 0.050 | 0.07 | 0.30 | 0.062 |
| | $+ \mathcal{L}_{\text{mm}}$ (Full) | **0.6380** | **0.033** | **0.06** | **0.19** | **0.098** |

## F.4. Fusion, encoder, and prototype-capacity sensitivity

The main model uses the gated fusion operator in Eq. (4). Table 18 compares it against concatenation and a single attention-based fusion block under identical losses and optimization. Gated fusion is best on 15 of 16 datasets, but the median gain over concatenation is only +0.005. Thus, the ablation gains are not explained by an unusually strong fusion head. The gate provides a stable and parameter-efficient way to combine modalities, while the main improvements come from the prototype objective.

| Diagnostic | Primary observed stage | Strongest secondary effect |
|---|---|---|
| Dead% | **71% at $+\mathcal{L}_{\text{alloc}}$** | 17% at $+\mathcal{L}_{\text{geo}}$ |
| $\widetilde{H}_{\text{pure}}$ | **72% at $+\mathcal{L}_{\text{alloc}}$** | 11% at $+\mathcal{L}_{\text{geo}}$ and 11% at $+\mathcal{L}_{\text{mm}}$ |
| $W_\varepsilon$ | **39% at $+\mathcal{L}_{\text{mm}}$** | 25% at $+\mathcal{L}_{\text{alloc}}$ |
| $\overline{\Delta}$ | **40% at $+\mathcal{L}_{\text{geo}}$ and 40% at $+\mathcal{L}_{\text{mm}}$** | 22% at $+\mathcal{L}_{\text{anchor}}$ |

*Table 16.* **Primary diagnostic drivers.** Stage-wise attribution of each diagnostic change from Proto-Classic to the full objective. Bold marks the primary observed driver for each diagnostic.

| Dataset | Metric | Concat | Attention | Gated (ours) |
|---|---|---|---|---|
| IMDB | AUC | 0.877 | 0.880 | **0.886** |
| Fake Job | AUC | 0.989 | 0.988 | **0.993** |
| Kickstarter | AUC | 0.810 | 0.808 | **0.811** |
| Jigsaw | AUC | 0.965 | 0.967 | **0.970** |
| Wine | Acc | **0.847** | 0.840 | 0.844 |
| Airbnb | Acc | 0.501 | 0.503 | **0.506** |
| PTech | AUC | 0.739 | 0.741 | **0.744** |
| Food101 | Acc | 0.947 | 0.949 | **0.952** |
| Fakeddit | Acc | 0.912 | 0.914 | **0.917** |
| Memotion | Acc | 0.599 | 0.601 | **0.604** |
| CCD | AUC | 0.921 | 0.923 | **0.930** |
| WikiArt | Acc | 0.820 | 0.822 | **0.825** |
| HAM | Acc | 0.927 | 0.929 | **0.932** |
| PetFinder | QWK | 0.439 | 0.441 | **0.444** |
| COVID | Acc | 0.938 | 0.940 | **0.943** |
| Artm | Acc | 0.658 | 0.660 | **0.663** |
| Avg. | | 0.806 | 0.807 | **0.810** |
| Wins | | 1/16 | 0/16 | **15/16** |

*Table 18.* **Fusion head sensitivity.** Frozen-encoder comparison in which loss weights and optimization are fixed and only the fusion operator changes. Bold marks the best value in each row.

Table 19 swaps frozen text, image, and tabular encoders while keeping the prototype head, objective, and training protocol fixed. Stronger encoders improve the average score, showing that the prototype constraints and OT alignment can exploit higher-quality embeddings. Lightweight encoders reduce the score, as expected, but do not require a different head. The main benchmark comparisons remain fair because they fix encoders to the benchmark-standard settings; encoder scaling is an orthogonal deployment choice rather than the source of the objective-level gains.

We also vary the number of prototypes per class. This stress test uses a single global configuration for all 16 datasets rather than dataset-specific tuning: embedding dimension $d = 384$, dropout 0.3, gated fusion, learning rate $5 \times 10^{-4}$, weight decay $10^{-4}$, batch size 64, 8 warmup epochs, prototype updates starting at epoch 5 with a 15-epoch ramp, and fixed structural weights $\lambda_{\text{mm}} = 1.5$, $\lambda_{\text{geo}} = 1.2$, $\lambda_{\text{alloc}} = 1.0$, $\lambda_{\text{out}} = 1.0$, and $\lambda_{\text{pred}} = 1.0$. We vary $K \in \{1, 2, 4, 8, 16, 32\}$ and report

$$\Delta_{i,K} = \text{Metric}_i(K) - \text{Metric}_i(K = 1), \tag{22}$$

where the metric is the dataset's primary AUC, accuracy, or QWK score. Figure 4 shows that the mean aggregate delta remains within a narrow $\pm 0.001$ band across the range. This stability indicates that performance is not the result of tuning the structural bottleneck $K$ to individual datasets. We use $K = 8$ as the default because it balances expressivity with compact case-based explanations.

| Setting | Text / Image / Tab backbones | Avg. score | Δ vs. baseline |
|---|---|---|---|
| Baseline | ELECTRA + Swin-L + MLP | 0.835 | +0.000 |
| Best overall | DeBERTa-V3 + ConvNeXt-L + FT-Trans. | **0.869** | **+0.032** |
| Balanced | DeBERTa-V3 + ViT-B/16 + FT-Trans. | 0.860 | +0.019 |
| Lightweight | RoBERTa + ResNet-50 + MLP | 0.811 | -0.017 |
| Minimalist | DistilBERT + ResNet-50 + MLP | 0.790 | -0.039 |

*Table 19.* **Backbone sensitivity.** Frozen-encoder variants with the prototype head, objective, and training protocol fixed. Bold marks the best average score and corresponding gain.

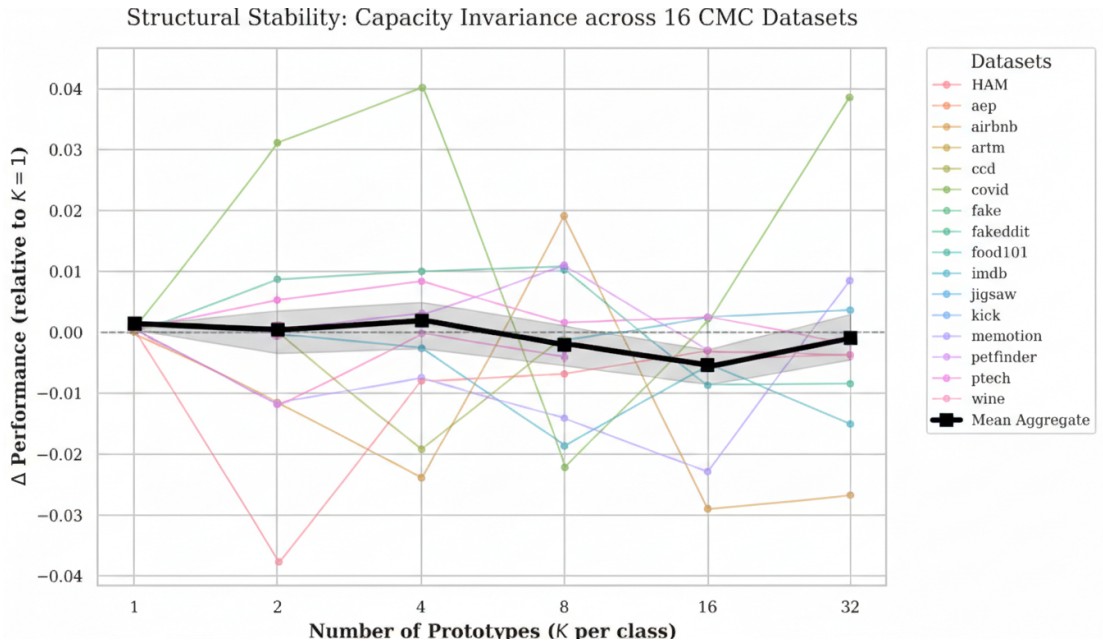

*Figure 4.* **Prototype-count sensitivity.** Performance deltas relative to $K = 1$ across all 16 CMC datasets. Individual lines correspond to datasets; the bold line is the mean aggregate delta, and the shaded region is the standard error of the mean.

### F.5. Low-data and imbalanced benchmark slices

The benchmark already contains several low-data, imbalanced, and high-cardinality tasks. Table 20 isolates these slices to make their difficulty explicit. The slice means are computed over the same primary metrics used in the main results, all scaled to $[0, 1]$. CAMP remains competitive in each regime: CAMP-E improves over AutoGluon in all three slices, and CAMP-F is also above AutoGluon on the low-data and severe-imbalance summaries.

| Slice | # | Included datasets | CAMP-F | CAMP-E | AG |
|---|---|---|---|---|---|
| Low data, $N_{\mathrm{train}} < 2000$ | 5 | IMDB, PTech, CCD, COVID, Artm | 0.836 | **0.873** | 0.830 |
| Severe imbalance or long tail | 6 | Fake Job, Jigsaw, Wine, Airbnb, WikiArt, HAM | 0.842 | **0.863** | 0.838 |
| High-cardinality classification | 4 | Wine, Airbnb, Food101, WikiArt | 0.777 | **0.804** | 0.775 |

*Table 20.* **Challenge-slice performance.** Unweighted mean primary metric for low-data, imbalanced/long-tail, and high-cardinality subsets. Bold marks the best value in each row.

| Dataset | Challenge description | Metric | CAMP-F | CAMP-E | AG |
|---|---|---|---|---|---|
| IMDB | 800 training examples | AUC | 0.899 | **0.911** | 0.878 |
| PTech | 749 training examples | AUC | 0.744 | **0.800** | 0.732 |
| CCD | 1,126 training examples | AUC | 0.937 | **0.972** | 0.950 |
| COVID | 707 training examples | Acc | 0.963 | **0.982** | 0.937 |
| Artm | 573 training examples | Acc | 0.638 | **0.698** | 0.655 |
| Fake Job | ≈95/5 binary imbalance | AUC | 0.992 | **0.997** | 0.974 |
| Jigsaw | ≈90/10 binary imbalance | AUC | 0.974 | **0.986** | 0.967 |
| Wine | 30-class long-tail label space | Acc | 0.844 | **0.872** | 0.842 |
| Airbnb | 24 classes with sparse extreme bins | Acc | 0.488 | **0.529** | 0.498 |
| WikiArt | 27-class long-tail label space | Acc | 0.825 | **0.842** | 0.809 |
| HAM | 7-class medical imbalance | Acc | 0.932 | **0.953** | 0.935 |
| Food101 | 101-class Image+Text classification | Acc | 0.950 | **0.971** | 0.949 |

*Table 21.* **Dataset-level challenge-slice scores.** Primary metric for the difficult regimes already present in the benchmark. Bold marks the best value in each row.

### F.6. Partial-observation and missing-modality robustness

Because Eq. (4) computes gates over the observed modality set, the same trained model can be evaluated under partial observations without imputation or retraining. We stress-test this property by dropping modalities at test time. For two-modality datasets, we evaluate the full input, each single-modality removal, and the all-drop floor. For three-modality datasets, we evaluate all non-empty observed subsets and the all-drop floor. The all-drop rows are included for completeness, but they correspond to prior-like or chance-level behavior and should not be interpreted as deployment settings.

Table 22 summarizes robustness over all datasets and drop configurations. CAMP-E has the best no-drop average, single-drop average, and overall mean across all configurations. CAMP-F has the second-best overall mean and exceeds AutoGluon on the single-drop average. AutoGluon has a small advantage only on the heavily compressed worst-case average, where all-drop floors dominate the statistic.

Taken together, the cumulative ladder, leave-one-out ablations, mechanistic diagnostics, fusion comparison, backbone swaps, prototype-count stress test, benchmark challenge slices, and partial-observation study support a consistent interpretation. Geometry and allocation make the prototype bank separated and usable, anchors stabilize learning, and class-wise OT alignment is the dominant mechanism for coherent multimodal complementarity.

For completeness, the next four tables report the missing-modality grids used to compute the aggregates above. Rows labeled "none" correspond to the fully observed benchmark setting. Rows labeled with modalities list the precise inputs removed at test time.

| Aggregate scores | | | | |
|---|---|---|---|---|
| Model | No drop | Single drop | Worst case | All configs |
| CAMP-E | **0.8416** | **0.7240** | 0.2806 | **0.6342** |
| CAMP-F | 0.8139 | 0.7009 | 0.2802 | 0.6154 |
| AG | 0.8076 | 0.6924 | **0.2812** | 0.6095 |
| TTT | 0.7772 | 0.6588 | 0.2781 | 0.5836 |
| PMX | 0.7710 | 0.6346 | 0.2759 | 0.5593 |
| PR | 0.7231 | 0.6347 | 0.2744 | 0.5517 |
| MLP | 0.7666 | 0.6135 | 0.2747 | 0.5461 |
| XGB | 0.7397 | 0.5913 | 0.2733 | 0.5295 |
| LP | 0.7241 | 0.5988 | 0.2724 | 0.5259 |
| PB | 0.7446 | 0.5979 | 0.2672 | 0.5200 |

| Counts over the missing-modality grid | | | | |
|---|---|---|---|---|
| Model | ≥ AG, all anchors | ≥ AG, missing anchors | Best or joint-best, all | Best or joint-best, missing |
| CAMP-F | 42/76 | 31/60 | 0 | 0 |
| CAMP-E | **62/76** | **46/60** | 57 | 41 |
| AG | 76/76 | 60/60 | 8 | 8 |
| MLP | | | 4 | 4 |
| TTT | | | 3 | 3 |
| XGB | | | 2 | 2 |
| PR | | | 2 | 2 |
| PMX | | | 1 | 1 |

| Family-level counts against AutoGluon | | | | |
|---|---|---|---|---|
| Family | CAMP-F ≥ AG, all | CAMP-E ≥ AG, all | CAMP-F ≥ AG, missing | CAMP-E ≥ AG, missing |
| Tab+Text | 15/24 | **20/24** | 10/18 | **14/18** |
| Image+Text | 11/16 | **12/16** | 8/12 | **8/12** |
| Image+Tab | 5/12 | **10/12** | 4/9 | **7/9** |
| Image+Text+Tab | 11/24 | **20/24** | 9/21 | **17/21** |

*Table 22.* **Missing-modality robustness summary.** "All configs" averages over the full grid of observed and dropped modality patterns. The AG comparison columns are meaningful for CAMP-F and CAMP-E; AG is included as the reference row. Bold marks the best value in each aggregate column and best non-reference counts.

*Table 23.* **Missing-modality grid: Tab+Text datasets.** Bold marks the best value in each row.

| Dataset | Drop | XGB | LP | PR | PB | PMX | MLP | TTT | AG | CAMP-F | CAMP-E |
|---|---|---|---|---|---|---|---|---|---|---|---|
| IMDB | none | 0.8060 | 0.8010 | 0.7950 | 0.8130 | 0.8430 | 0.8390 | 0.8860 | 0.8780 | 0.8990 | **0.9110** |
| IMDB | tab | 0.7631 | 0.7382 | 0.7602 | 0.7519 | 0.7593 | 0.7640 | 0.8302 | 0.8253 | 0.8293 | **0.8391** |
| IMDB | text | 0.7626 | 0.7599 | 0.7907 | 0.7913 | 0.8164 | 0.7960 | 0.8554 | 0.8491 | 0.8621 | **0.8722** |
| IMDB | tab+text | 0.5030 | 0.4936 | 0.4980 | 0.4975 | 0.5075 | 0.4956 | **0.5086** | 0.4996 | 0.5011 | 0.5021 |
| Fake Job | none | 0.9480 | 0.9150 | 0.9200 | 0.9420 | 0.9700 | 0.9510 | 0.9770 | 0.9740 | 0.9920 | **0.9970** |
| Fake Job | tab | 0.9340 | 0.8698 | 0.9250 | 0.9126 | 0.9380 | 0.8930 | **0.9491** | 0.9476 | 0.9410 | 0.9453 |
| Fake Job | text | 0.7808 | 0.8238 | 0.9159 | 0.8753 | 0.8983 | 0.8503 | 0.9041 | 0.9052 | 0.9160 | **0.9202** |
| Fake Job | tab+text | 0.4980 | 0.4998 | 0.5007 | 0.5014 | 0.5002 | 0.4975 | **0.5025** | 0.5005 | 0.5002 | 0.5006 |
| Kickstarter | none | 0.7630 | 0.7450 | 0.7100 | 0.7360 | 0.7660 | 0.7630 | 0.7670 | 0.7990 | 0.8110 | **0.8440** |
| Kickstarter | tab | 0.6914 | 0.6975 | 0.6809 | 0.6887 | 0.7175 | 0.7068 | 0.7238 | 0.7582 | 0.7492 | **0.7772** |
| Kickstarter | text | 0.6687 | 0.6438 | 0.7057 | 0.6283 | 0.6501 | 0.6573 | 0.6743 | 0.7115 | 0.7825 | **0.8114** |
| Kickstarter | tab+text | 0.4990 | 0.4998 | 0.4997 | 0.4994 | 0.4997 | 0.5007 | 0.5001 | **0.5009** | 0.4994 | 0.5003 |
| Jigsaw | none | 0.9410 | 0.9120 | 0.9050 | 0.9210 | 0.9540 | 0.9550 | 0.9450 | 0.9670 | 0.9740 | **0.9860** |
| Jigsaw | tab | 0.7814 | 0.8528 | 0.8238 | 0.7962 | 0.8215 | 0.8550 | 0.8550 | 0.8820 | 0.9187 | **0.9290** |
| Jigsaw | text | 0.9264 | 0.9170 | 0.8806 | 0.9082 | 0.9417 | **0.9423** | 0.9180 | 0.9415 | 0.9088 | 0.9191 |
| Jigsaw | tab+text | 0.5007 | 0.5005 | 0.5008 | 0.4996 | 0.5004 | 0.4997 | 0.4992 | 0.5004 | 0.5008 | **0.5018** |
| Wine | none | 0.8020 | 0.7370 | 0.7800 | 0.7950 | 0.8180 | 0.8090 | 0.8210 | 0.8420 | 0.8440 | **0.8720** |
| Wine | tab | 0.6669 | 0.6600 | **0.7850** | 0.7118 | 0.7415 | 0.6903 | 0.7490 | 0.7740 | 0.2383 | 0.2487 |
| Wine | text | 0.3443 | 0.2667 | 0.7386 | 0.3323 | 0.3614 | 0.3538 | 0.5060 | 0.5445 | 0.7202 | **0.7426** |
| Wine | tab+text | 0.2053 | 0.1956 | 0.2020 | 0.2043 | 0.2077 | 0.2064 | 0.2082 | 0.2113 | 0.2116 | **0.2129** |
| Airbnb | none | 0.4450 | 0.3610 | 0.3550 | 0.3780 | 0.4010 | 0.3950 | 0.3830 | 0.4980 | 0.4880 | **0.5290** |
| Airbnb | tab | 0.3300 | 0.3000 | 0.2950 | 0.3100 | 0.3300 | 0.3200 | 0.3150 | 0.3850 | 0.3900 | **0.4213** |
| Airbnb | text | 0.4050 | 0.3400 | 0.3350 | 0.3550 | 0.3750 | 0.3700 | 0.3350 | 0.4350 | 0.4250 | **0.4583** |
| Airbnb | tab+text | 0.1520 | 0.1390 | 0.1380 | 0.1420 | 0.1450 | 0.1440 | 0.1430 | 0.1600 | 0.1580 | **0.1601** |

*Table 24.* **Missing-modality grid: Image+Text datasets.** Bold marks the best value in each row.

| Dataset | Drop | XGB | LP | PR | PB | PMX | MLP | TTT | AG | CAMP-F | CAMP-E |
|---|---|---|---|---|---|---|---|---|---|---|---|
| PTech | none | 0.7260 | 0.6880 | 0.6900 | 0.7030 | 0.7160 | 0.7200 | 0.7180 | 0.7320 | 0.7440 | **0.8000** |
| PTech | image | 0.5585 | 0.6127 | 0.6276 | 0.6202 | 0.6357 | 0.6360 | 0.6280 | **0.6470** | 0.5679 | 0.6076 |
| PTech | text | **0.7310** | 0.6905 | 0.6950 | 0.6905 | 0.7136 | 0.7194 | 0.6910 | 0.7065 | 0.6291 | 0.6736 |
| PTech | image+text | 0.5020 | 0.5036 | 0.5049 | 0.4958 | 0.5031 | 0.4987 | 0.4944 | 0.4990 | 0.5068 | **0.5088** |
| Food101 | none | 0.8960 | 0.9130 | 0.8850 | 0.9030 | 0.9250 | 0.9160 | 0.8920 | 0.9490 | 0.9500 | **0.9710** |
| Food101 | image | 0.5460 | 0.5805 | 0.5875 | 0.5530 | 0.5925 | 0.5485 | 0.5770 | 0.6515 | 0.6700 | **0.6847** |
| Food101 | text | 0.8760 | 0.8940 | 0.8680 | 0.8830 | 0.9060 | 0.8950 | 0.8740 | 0.9320 | 0.9340 | **0.9533** |
| Food101 | image+text | **0.0114** | 0.0102 | 0.0104 | 0.0086 | 0.0088 | 0.0103 | 0.0106 | **0.0114** | 0.0103 | 0.0086 |
| Fakeddit | none | 0.7570 | 0.8500 | 0.8200 | 0.8410 | 0.8910 | 0.8830 | 0.9060 | 0.9140 | 0.9270 | **0.9540** |
| Fakeddit | image | 0.5908 | 0.7110 | 0.7136 | 0.6551 | 0.7086 | 0.6398 | 0.7530 | 0.7695 | 0.8898 | **0.9134** |
| Fakeddit | text | 0.5543 | 0.7186 | 0.6837 | 0.7492 | 0.8162 | 0.7569 | 0.7890 | 0.8035 | 0.8541 | **0.8778** |
| Fakeddit | image+text | 0.2552 | 0.2692 | 0.2647 | 0.2678 | 0.2753 | 0.2741 | 0.2776 | 0.2788 | 0.2807 | **0.2821** |
| Memotion | none | 0.5650 | 0.5760 | 0.5650 | 0.5790 | 0.6150 | 0.5930 | 0.6140 | 0.6550 | 0.6440 | **0.6660** |
| Memotion | image | 0.5245 | 0.5810 | 0.5700 | 0.5840 | **0.6200** | 0.5198 | 0.5510 | 0.5955 | 0.5977 | 0.6164 |
| Memotion | text | 0.5395 | 0.5215 | 0.4853 | 0.5790 | 0.6150 | 0.2587 | 0.5690 | 0.6125 | 0.6204 | **0.6403** |
| Memotion | image+text | 0.2547 | 0.2564 | 0.2547 | 0.2569 | 0.2622 | 0.2590 | 0.2621 | **0.2682** | 0.2666 | 0.2680 |

*Table 25.* **Missing-modality grid: Image+Tab datasets.** Bold marks the best value in each row.

| Dataset | Drop | XGB | LP | PR | PB | PMX | MLP | TTT | AG | CAMP-F | CAMP-E |
|---|---|---|---|---|---|---|---|---|---|---|---|
| CCD | none | 0.8780 | 0.8260 | 0.8500 | 0.8960 | 0.9130 | 0.9070 | 0.8850 | 0.9500 | 0.9370 | **0.9720** |
| CCD | image | 0.5021 | 0.5722 | **0.7191** | 0.5620 | 0.5557 | 0.5830 | 0.5880 | 0.6695 | 0.5908 | 0.6135 |
| CCD | tab | 0.8782 | 0.8297 | 0.8550 | 0.8954 | 0.9124 | 0.9016 | 0.8670 | 0.9330 | 0.9314 | **0.9638** |
| CCD | image+tab | 0.5047 | 0.5008 | 0.5064 | 0.4956 | 0.4934 | **0.5071** | 0.4977 | 0.5029 | 0.5007 | 0.5025 |
| WikiArt | none | 0.6790 | 0.7620 | 0.7500 | 0.7890 | 0.7930 | 0.8000 | 0.7620 | 0.8090 | 0.8250 | **0.8420** |
| WikiArt | image | 0.1190 | 0.2300 | 0.2740 | 0.2290 | 0.2610 | 0.2580 | 0.2580 | 0.3330 | 0.3770 | **0.3860** |
| WikiArt | tab | 0.6490 | 0.7335 | 0.7245 | 0.7590 | 0.7645 | 0.7685 | 0.7350 | 0.7835 | 0.8010 | **0.8165** |
| WikiArt | image+tab | 0.0373 | 0.0410 | 0.0404 | 0.0421 | 0.0423 | 0.0426 | 0.0410 | 0.0430 | 0.0437 | **0.0441** |
| HAM | none | 0.9130 | 0.9000 | 0.8800 | 0.8840 | 0.9040 | 0.9130 | 0.9220 | 0.9350 | 0.9320 | **0.9530** |
| HAM | image | 0.3930 | 0.4060 | 0.4380 | 0.3640 | 0.4100 | 0.3670 | 0.4540 | 0.4930 | 0.8936 | **0.9119** |
| HAM | tab | 0.8980 | 0.8858 | 0.8672 | 0.8690 | 0.8898 | 0.8972 | 0.9085 | 0.9223 | 0.9144 | **0.9337** |
| HAM | image+tab | 0.2584 | 0.2564 | 0.2534 | 0.2540 | 0.2570 | 0.2584 | 0.2597 | 0.2617 | 0.2612 | **0.2622** |

*Table 26.* **Missing-modality grid: Image+Text+Tab datasets.** Bold marks the best value in each row.

| Dataset | Drop | XGB | LP | PR | PB | PMX | MLP | TTT | AG | CAMP-F | CAMP-E |
|---|---|---|---|---|---|---|---|---|---|---|---|
| PetFinder | none | 0.2940 | 0.2050 | 0.2800 | 0.3100 | 0.3200 | 0.3230 | 0.4010 | 0.4280 | 0.4540 | **0.4880** |
| PetFinder | image | 0.2740 | 0.1860 | 0.2630 | 0.2900 | 0.3010 | 0.3020 | 0.3830 | 0.4110 | 0.4420 | **0.4720** |
| PetFinder | tab | 0.1310 | 0.1142 | 0.1758 | 0.1496 | 0.1411 | 0.0820 | 0.2660 | 0.3005 | 0.4253 | **0.4555** |
| PetFinder | text | 0.2150 | 0.1989 | 0.2821 | 0.2339 | 0.2569 | 0.3207 | 0.3380 | 0.3685 | 0.4062 | **0.4354** |
| PetFinder | image+tab | 0.1181 | 0.0394 | 0.1299 | 0.1338 | 0.1391 | 0.0800 | 0.2400 | 0.2749 | 0.4233 | **0.4511** |
| PetFinder | image+text | 0.1981 | 0.1154 | 0.1979 | 0.2138 | 0.2281 | 0.2220 | 0.3120 | 0.3429 | 0.4042 | **0.4280** |
| PetFinder | tab+text | 0.0000 | 0.0000 | 0.1145 | 0.0000 | 0.0350 | 0.0338 | 0.1950 | **0.2324** | 0.1319 | 0.1408 |
| PetFinder | image+tab+text | -0.0037 | -0.0020 | 0.0229 | -0.0020 | 0.0070 | -0.0031 | 0.0351 | **0.0485** | 0.0276 | 0.0212 |
| COVID | none | 0.8920 | 0.8870 | 0.8750 | 0.9010 | 0.9330 | 0.9000 | 0.9340 | 0.9370 | 0.9630 | **0.9820** |
| COVID | image | 0.8920 | 0.8829 | 0.8615 | 0.9012 | 0.9285 | 0.8865 | 0.8980 | 0.9030 | 0.9203 | **0.9363** |
| COVID | tab | 0.6307 | 0.8920 | 0.7579 | 0.7300 | 0.6717 | **0.9050** | 0.8440 | 0.8520 | 0.8379 | 0.8533 |
| COVID | text | 0.7884 | 0.5541 | 0.8029 | 0.2435 | 0.6582 | 0.5667 | 0.8170 | 0.8265 | 0.8259 | **0.8412** |
| COVID | image+tab | 0.6307 | 0.8879 | 0.7629 | 0.7255 | 0.6717 | **0.8915** | 0.7901 | 0.8001 | 0.8429 | 0.8533 |
| COVID | image+text | 0.7884 | 0.4910 | 0.7894 | 0.2300 | 0.6402 | 0.5712 | 0.7631 | 0.7746 | 0.8309 | **0.8411** |
| COVID | tab+text | 0.5181 | 0.4974 | 0.6678 | 0.0400 | 0.5861 | 0.2604 | 0.7091 | **0.7236** | 0.5982 | 0.6066 |
| COVID | image+tab+text | 0.1252 | 0.1247 | 0.1235 | 0.0450 | 0.1293 | 0.1260 | 0.1294 | 0.1297 | 0.1323 | **0.1325** |
| Artm | none | 0.5300 | 0.5070 | 0.5100 | 0.5220 | 0.5740 | 0.5980 | 0.6230 | 0.6550 | 0.6380 | **0.6980** |
| Artm | image | 0.3800 | 0.3645 | 0.3825 | 0.3720 | 0.4315 | 0.4405 | 0.4880 | 0.5275 | 0.5180 | **0.5625** |
| Artm | tab | 0.5100 | 0.4880 | 0.4930 | 0.5020 | 0.5550 | 0.5770 | 0.6050 | 0.6380 | 0.6220 | **0.6750** |
| Artm | text | 0.4600 | 0.4405 | 0.4505 | 0.4520 | 0.5075 | 0.5245 | 0.5600 | 0.5955 | 0.5820 | **0.6324** |
| Artm | image+tab | 0.3500 | 0.3359 | 0.3559 | 0.3421 | 0.4016 | 0.4081 | 0.4581 | 0.4980 | 0.4898 | **0.5192** |
| Artm | image+text | 0.3000 | 0.2884 | 0.3134 | 0.2921 | 0.3541 | 0.3556 | 0.4131 | 0.4555 | 0.4498 | **0.4772** |
| Artm | tab+text | 0.4300 | 0.4119 | 0.4239 | 0.4221 | 0.4776 | 0.4921 | 0.5301 | 0.5660 | 0.5538 | **0.5881** |
| Artm | image+tab+text | 0.0723 | 0.0704 | 0.0706 | 0.0717 | 0.0761 | 0.0782 | 0.0803 | **0.0831** | 0.0816 | 0.0822 |

## G. Explanation extraction for multimodal prototype-based models

This appendix formalizes the explanation protocol used in the main-body Airbnb case study. The goal is to verify that the displayed evidence is tied directly to the deployed prototype head and it is applicable to any other case. We did not provide more case studies due to time constrains but the method still holds as per the following theoretical guratantees. The returned modality gates, retrieved prototypes, and training anchors are not surrogate objects fitted after the fact; they are the quantities used by the classifier to form its prediction.

For fine-grained input evidence, we define a fixed-support evidence probe. The probe ranks local input units by their exact effect on the retrieved-prototype score while keeping the original retrieved prototype support fixed. This produces token-, feature-, or region-level evidence without replacing the case-based explanation by an auxiliary model.

The probe establishes model-internal faithfulness for the trained prototype head under fixed retrieved support. It does not claim that the resulting visualization is the best possible interface for human decision makers, and user-facing studies of prototype concepts and evidence presentation remain future work.

### G.1. Setup and Evidence Units

Consider a case $x = (\{x^{(m)}\}_{m \in \mathcal{M}_x})$ with observed modalities $\mathcal{M}_x$. Let $z(x)$ denote the fused embedding computed by gated late fusion (Eq. (4)), and let $\ell(x) \in \mathbb{R}^C$ be the prototype-head logits (Eq. (5)). For each modality $m$, define a set of evidence units $\mathcal{U}^{(m)}(x)$: tokens for text, input features for tabular data, and spatial cells on a $G \times G$ grid for images. For a unit $u \in \mathcal{U}^{(m)}(x)$, the ablation operator $\mathsf{Abl}(x; m, u)$ produces a perturbed case $x_{-u}$ by applying the corresponding modality-level intervention: token masking for text, replacement by a fixed baseline such as a column mean or unknown category for tabular features, or grid-cell occlusion for images. Encoders and fusion are re-run on the perturbed case, and no surrogate model is fitted.

### G.2. Fixed-Support Prototype Score

Ablating an input unit can change which prototypes are returned by the top-$r$ operator in Eq. (5). For the purpose of explaining the evidence already displayed for a reference prediction, we therefore hold the retrieved support fixed. Let $\mathcal{S}_c(x) \equiv \mathcal{N}_c^{(r)}(x)$ be the class-$c$ top-$r$ prototype indices selected by Eq. (5), with ties broken deterministically. Given a reference case $x$, define the fixed-support logit for any perturbed case $x'$ as

$$\tilde{\ell}_c\big(x'; \mathcal{S}(x)\big) \; = \; \frac{1}{r} \sum_{k \in \mathcal{S}_c(x)} S\big(z(x'), \, p_{c,k}\big). \tag{23}$$

For the predicted class $\hat{y} = \arg\max_c \ell_c(x)$, the unit-level probe score is the target-logit drop

$$\Delta_{m,u} \; = \; \tilde{\ell}_{\hat{y}}\big(x; \mathcal{S}(x)\big) - \tilde{\ell}_{\hat{y}}\big(\mathsf{Abl}(x; m, u); \mathcal{S}(x)\big). \tag{24}$$

Positive values indicate evidence that supports the displayed class under the original retrieved prototypes; negative values indicate evidence whose removal increases the fixed-support target logit.

### G.3. Explanation Extraction algorithm

---

**Algorithm G.1** Fixed-support evidence probe.

---

**Require:** Case $x$, trained CAMP parameters (encoders $\{f^{(m)}\}$, projections $\{\pi^{(m)}\}$, gates, prototypes $\{p_{c,k}\}$), neighbor count $r$, grid size $G$.
1:  Compute modality embeddings $\{z^{(m)}\}$, gates $\{g^{(m)}\}$, and fused embedding $z(x)$ (Eq. (4)).
2:  Compute logits $\ell(x)$ and predicted class $\hat{y}$ (Eq. (5)).
3:  Record prototype supports $\mathcal{S}_c(x) \leftarrow \mathcal{N}_c^{(r)}(x)$ for all $c$ (Eq. (5)).
4:  Set reference fixed-support logit $\ell_{\hat{y}}^{\text{fix}} \leftarrow \tilde{\ell}_{\hat{y}}(x; \mathcal{S}(x))$ (Eq. (23)).
5:  **for** each observed modality $m \in \mathcal{M}_x$ **do**
6:      **for** each evidence unit $u \in \mathcal{U}^{(m)}(x)$ **do**
7:          Construct perturbed case $x_{-u} \leftarrow \mathsf{Abl}(x; m, u)$.
8:          Re-run encoders and fusion to obtain $z(x_{-u})$.
9:          Compute fixed-support logit $\tilde{\ell}_{\hat{y}}(x_{-u}; \mathcal{S}(x))$ (Eq. (23)).
10:         Score unit impact $\Delta_{m,u} \leftarrow \ell_{\hat{y}}^{\text{fix}} - \tilde{\ell}_{\hat{y}}(x_{-u}; \mathcal{S}(x))$.
11:     **end for**
12: **end for**
**Ensure:** Ranked evidence units by $\Delta_{m,u}$, retrieved support $\mathcal{S}_{\hat{y}}(x)$, and modality weights $\{g^{(m)}\}$.

---

### G.4. Faithfulness Guarantee

**Proposition G.1** (Exact head reconstruction at the reference case). *Let $\mathcal{S}_c(x)$ be the top-r prototype indices selected by Eq. (5) on a reference case $x$. Then for all classes $c$, the fixed-support logits match the deployed logits at $x$: $\tilde{\ell}_c(x; \mathcal{S}(x)) = \ell_c(x)$. In floating-point implementations, this equality holds up to numerical tolerance.*

*Proof.* By definition, $\mathcal{S}_c(x)$ contains exactly the prototype indices used by Eq. (5) to compute $\ell_c(x)$. Substituting the same set into Eq. (23) yields the same average similarity. $\square$

**Theorem G.2** (Fixed-support interventional faithfulness). *Fix a reference case $x$ and support $\mathcal{S}(x)$. Consider the deterministic computation induced by the trained model: encoders and projections compute $\{z^{(m)}\}$, fusion computes $z$, and the fixed-support head computes $\tilde{\ell}(\cdot; \mathcal{S}(x))$ via Eq. (23). For any modality $m$ and evidence unit $u \in \mathcal{U}^{(m)}(x)$, the probe score $\Delta_{m,u}$ equals the interventional effect of ablating $u$ on the predicted-class fixed-support logit:*

$$\Delta_{m,u} \;=\; \tilde{\ell}_{\hat{y}}(x; \mathcal{S}(x)) - \tilde{\ell}_{\hat{y}}(x_{-u}; \mathcal{S}(x)) \;=\; \tilde{\ell}_{\hat{y}}\big(\mathrm{do}(u \leftarrow \mathrm{orig})\big) - \tilde{\ell}_{\hat{y}}\big(\mathrm{do}(u \leftarrow \mathrm{abl})\big). \tag{25}$$

*Thus, ranking units by $\Delta_{m,u}$ identifies the units whose removal most decreases the model's predicted-class evidence within the fixed retrieved support.*

*Proof.* Under fixed support, the mapping from input units to $\tilde{\ell}_{\hat{y}}$ is a deterministic composition of the encoder, fusion module, and prototype head. The intervention $\mathrm{do}(u \leftarrow \mathrm{abl})$ is implemented by constructing $x_{-u} = \mathsf{Abl}(x; m, u)$ and re-evaluating the same structural computation while keeping $\mathcal{S}(x)$ fixed. The difference in Eq. (25) is therefore exactly the fixed-support interventional effect on the deployed head. $\square$

**Remarks and Limitations.**  The fixed-support constraint probes the influence of input units relative to the prototypes originally returned to explain the prediction. This is the appropriate check when the question is whether the displayed prototypes, gates, and local evidence are actual decision drivers for the current instance. A complementary analysis can also report whether $\mathcal{N}_{\hat{y}}^{(r)}(x_{-u})$ differs from $\mathcal{S}_{\hat{y}}(x)$; such changes indicate that the model would retrieve a different set of prototypes under the intervention.

We focus the qualitative discussion on the main-body example and do not duplicate dataset-specific case studies in the appendix. The fixed-support evidence probe is model-exact under the stated support constraint (Theorem G.2) and can be applied to any CMC input once the corresponding evidence units and ablation operators are defined.

