# OpenReview forum: "CAMP: Coherent Alignment of Multimodal Prototypes for Explainable Complementary Learning"
_ICML.cc/2026/Conference — ICML 2026 regular_

### Official Review · Reviewer_6YRS · 2026-02-27

**Soundness:** 2
**Presentation:** 1
**Significance:** 3
**Originality:** 2
**Overall Recommendation:** 4
**Confidence:** 2

**Summary:**

This paper addresses the underexplored setting of Complementary Multimodal Classification (CMC), where modalities provide distinct, non-redundant evidence, and existing interpretable methods often compromise accuracy due to their reliance on shared similarity metrics. The authors propose CAMP (Coherent Alignment of Multimodal Prototypes), a modality-agnostic framework that uses optimal transport to align class-wise evidence across modalities and enforces geometric constraints to prevent modality dominance and representation collapse, with theoretical guarantees against these degeneracies. CAMP achieves state-of-the-art or competitive performance across 17 CMC datasets with a highly parameter-efficient design (under 1M parameters), matching or surpassing large AutoML baselines.

**Compliance With Llm Reviewing Policy:**

Affirmed.

**Final Justification:**

Although most concerns are addressed in the rebuttal, the paper should be carefully polished and figures be replotted in the camera-ready for better clarification of the cnotribution of this paper.

**Key Questions For Authors:**

1. There are too many loss objectives shown in Equ 13 and Equ 8. What’s the contribution of each loss and how to balance the weights is quite a sensitive work.

2. How does the proposed method avoid the degeneracy trio mentioned in Section 4 correspondingly, referring to collapse, dominance and disagreement?

3. Since the paper focus on explainable complementary learning, how does OT and prototype learning improve the interpretability in both theory and practice?

**Limitations:**

1. The figures 1-3 seem partially generated directly by AI tools, which includes multiple errors and confusing notation, such as in Figure 3, Proto 43 rank 1 has similarity 0.91 while Proto 47 rank 5 has larger similarity 0.95. Please check and revise these confusing areas carefully.

2. Only Section 4.1 is the key novel module for the paper, other objectives seems combination of previous work as mentioned in line 269.

3. Too many mathematical derivation in Supplementary A-B while too few explanation for these expressions.

4. More cases about the explanation complementary learning should be presented.

**Strengths And Weaknesses:**

1.	Extensive experiments.
2.	Significance is good, while presentation and soundness is poor which require more language polish and evaluable explanation. Originality is acceptable.

---

> ### Author Rebuttal · Authors · 2026-03-30
>
> Thank you for your careful review and recognizing our problem's significance and experimental breadth. To improve clarity, we address your questions directly below.
>
> First, regarding Eq. (13), each loss block targets a specific failure mode, while Eq. (8) simply expands the geometry block inside that full objective. Geometry enforces class separation, allocation ensures prototypes remain utilized and class-pure, anchors stabilize against outliers, and class-wise OT alignment reduces cross-modal disagreement. The loss weights are not set manually; we tune them per dataset via the same 50-trial Optuna protocol used throughout Sec. 5.2 / App. E. In addition, the alignment-related losses are linearly ramped rather than applied at full strength from the start, which reduces sensitivity to early noisy assignments. We will also improve readability by adding short narrated overviews and intuition before the formal derivations, so the theory is easier to follow.
>
> Second, to answer how CAMP avoids the degeneracy trio, we now add direct mechanistic ablations rather than relying only on task accuracy. We track Dead\% for unused prototypes, H\~\_pure for class impurity of prototypes, W\_epsilon for cross-modal disagreement via class-wise OT cost, and Delta-bar for the correct-vs-incorrect prototype similarity margin. Across the cumulative ladder from Proto-Classic to full CAMP, task performance rises from 0.7445 to 0.8139 while Dead\% falls from 26.6\% to 2.3\%, H\~\_pure from 0.22 to 0.04, W\_epsilon from 0.40 to 0.12, and Delta-bar rises from 0.031 to 0.118. The stage-wise changes also align with the intended role of each block: allocation gives the largest improvement in utilization and purity, OT gives the largest improvement in cross-modal coherence, and geometry together with anchors rebuilds the decision margin. This shows that the losses are not redundant additions; each addresses a specific part of the collapse / dominance / disagreement problem. We also add a partial-observation study (missing modality at test time): in the missing-modality evaluation, CAMP-E leads on Avg no-drop (0.8416), Avg single-drop (0.7240), and overall mean (0.6342), while nearly matching the best worst-case score. We will add these results to the 9th page and provide full metrics in the Appendix. (See sWV1 rebuttal for precise definitions for added experiments). We will add all of these ablations and experiments to the 9th page and full metrics in the Appendix.
>
> Third, thank you for catching the Figure 3 inconsistency. You are right that the original display was confusing. We have redrawn the figure so the ranked prototype list, the similarities shown in the panel, and the anchored evidence boxes are now synchronized and strictly monotone. In the revised figure, the nearest-prototype ranking is ordered 0.91, 0.90, 0.89, 0.88, 0.85, and the latent-space annotations and retrieved anchors now match that ordering. The revised figure can be found at https://i.postimg.cc/BQFhm8cp/fig_3_replaced_for_Reviewers.png. We also simplified the panel titles, clarified the explanation flow, and we will also clean up Figs 1-2, correct the abstract to 16 datasets, not 17, and fix minor inconsistencies.
>
> Fourth, on how OT and prototype learning improve interpretability. CAMP explanations are not post-hoc surrogates. This aligns with prior work on interpretable-by-design models, where the explanation objects are the same objects used for prediction. In our case, the prediction itself is computed from retrieved class prototypes and modality gates, and those same objects are what the explanation shows. In other words, the explanation objects are the deployed decision objects. Prototype learning therefore gives case-based evidence by design: the model predicts using similar learned cases and exposes those cases directly. OT adds a different benefit: it makes those case-based explanations coherent across modalities. It does not force text, image, and tabular inputs to collapse into the same embedding; instead, it encourages them to agree on which prototypes represent a class in aggregate. We will also make the scope of this claim clearer: Appendix G proves computational faithfulness under fixed support for the prototype head, not human-interpretability validation through user studies.
>
> Lastly, on qualitative Case Studies: We selected the Airbnb case to provide a fully instrumented, end-to-end snapshot of the CAMP explanation possibilities (gating, retrieval, and fixed-support attribution) in a single instance. To better illustrate how these explanations generalize across modality regimes, the revised Appendix may include additional qualitative examples, including a completely non-tabular setting, demonstrating the mechanism under diverse multimodal configurations.
>
> We believe these clarifications resolve the concerns and make the method and its behavior clearer. Therefore, we would be very grateful if you would consider increasing your score accordingly.

---

> > ### Author Rebuttal · Reviewer_6YRS · 2026-04-01
> >
> > The limitaion still stand which mentioned that only Section 4.1 is the key novel module for the paper, other objectives seems combination of previous work as mentioned in line 269.

---

> > > ### Author Response · Authors · 2026-04-07
> > >
> > > We thank you for the continued engagement. We believe the remaining concern stems from a narrower reading of line 269 than intended. That sentence was written to scope the mathematical claim, not to imply that Section 4.1 is the sole source of novelty. As we clarified in our reply to Reviewer VJD6, the intended distinction is “classical tools, new CMC-specific composition,” not “only one novel module.”
> > >
> > > While Section 4.1 introduces the central new coherence module, the paper’s contribution is the CMC-specific mechanism as a whole. Concretely, CAMP contributes: (i) a formalization of CMC in Sec. 3.1, including arbitrary observed modality subsets and a complementarity condition that makes instance-level alignment inappropriate in this regime; (ii) a modality-agnostic prototype head with gated fusion and a shared class-partitioned prototype bank that operates under arbitrary observed subsets; (iii) class-wise OT alignment over prototype-usage distributions rather than instance-level embedding matching, so modalities become coherent without discarding complementary per-instance evidence; (iv) a unified objective in which geometry, allocation, anchors, and OT remove different but coupled degeneracies; and (v) a fixed-support faithfulness audit for the deployed prototype head.
> > >
> > > Section 4.1 contains the central coherence module, but the novelty of the paper is not confined to Section 4.1. Geometry, allocation, and anchors are not claimed as individually new objectives; they are part of the contribution because, in complementary multimodality, OT coherence is only meaningful when the prototype bank is separated, utilized, and stable. This dependence is part of the contribution itself, not a secondary detail, and is exactly what line 269 was intended to capture by describing the theory as a unified mechanism for CMC.
> > >
> > > The empirical results reinforce this point. If geometry, allocation, and anchors were merely auxiliary, then removing them would not degrade all 16 datasets (Tables 17–18), and the cumulative ablation ladder would not improve monotonically on all 16 datasets (Table 2). As clarified in our reply to Reviewer sWV1, the new mechanistic ablations confirm this beyond accuracy: allocation primarily improves prototype usage and purity, OT primarily reduces cross-modal disagreement, and geometry together with anchors rebuilds the similarity margin. And as clarified in our reply to Reviewers sWV1 and VJD6, this same mechanism also remains robust under partial observation: in the missing-modality study, CAMP-E is strongest on Avg no-drop (0.8416), Avg single-drop (0.7240), and the overall mean (0.6342), while remaining within 0.0006 of the best worst-case score.
> > >
> > > We will therefore revise the opening of Section 4 and line 269 to make this distinction explicit: the mathematical tools are classical, Section 4.1 contains the central coherence module, and the broader novelty of CAMP lies in the CMC-specific composition that makes this module non-degenerate, stable, and usable under arbitrary modality subsets.
> > >
> > > We believe these clarifications address your remaining concern by making the contribution and scope of our work more precise. If the reviewer agrees, we would be grateful if you consider increasing your support.

---

### Official Review · Reviewer_VJD6 · 2026-03-06

**Soundness:** 3
**Presentation:** 3
**Significance:** 2
**Originality:** 2
**Overall Recommendation:** 4
**Confidence:** 4

**Summary:**

The paper formalizes Complementary Multimodal Classification (CMC), a setting in which modalities supply distinct, non-redundant evidence for the same decision (e.g., an X-ray image paired with a patient’s textual history rather than a redundant caption). It then presents Coherent Alignment of Multimodal Prototypes (CAMP), a lightweight, modality-agnostic prototype model that fuses inputs via per-instance gates, retrieves from a shared class-partitioned prototype bank, and trains with four interlocking objectives: cross-modal entropic optimal transport alignment of class-conditioned usage distributions, geometric margins to prevent collapse, allocation penalties for coverage and purity, and anchor-based robustness against outlier drift. Theoretical analysis links these constraints to margin generalization, OT coherence, and bounded drift; The paper also provides theory connecting these components to reduced degeneracy and improved coherence, and reports experiments on 16 public multimodal classification datasets showing competitive frozen-encoder performance against large AutoML baselines and stronger end-to-end results. The method is presented as explainable by design because predictions are grounded in retrieved prototypes and modality gates.

**Compliance With Llm Reviewing Policy:**

Affirmed.

**Key Questions For Authors:**

1. How should the sub-1M parameter claim be interpreted in light of the strong frozen backbones?
Since CAMP-F uses large pretrained encoders, it would be helpful to clarify whether the intended claim is “small trainable head” rather than “small model.” A careful response would make the empirical messaging more convincing.

2. The OT alignment loss is computed only on observed modality pairs within each batch. How sensitive is final performance (and coherence) to batch composition when certain modality combinations are rare?

**Limitations:**

No. The paper mentions some modeling limitations, such as focusing on discrete outcomes and the limited scope of the faithfulness audit, which is good. Adding a short dedicated paragraph listing these points, together with the modest gains observed on some discriminative tasks, would improve transparency while preserving the paper’s strengths.

**Strengths And Weaknesses:**

- From a method perspective, the architecture itself is straightforward: modality-specific encoders, shared-space projections, instance-wise gated fusion, and a class-wise prototype bank with top-r retrieval. This makes the method easy to understand at a high level, and the role of each loss term is clearly motivated in the text.
- A notable originality strength is the use of class-wise OT alignment of prototype usage distributions rather than direct alignment of embeddings.

**Soundness:** The submission is technically strong. The CMC problem formulation and prototype failure modes are clearly stated and empirically motivated. The four loss blocks are motivated by concrete degeneracies (collapse, dominance, dead prototypes, drift); the proofs are self-contained in the appendix. Experiments follow a reproducible protocol (identical backbones, Optuna HPO with 50 trials, 5 seeds, fixed-support causal audit) across 16 diverse datasets spanning all modality combinations, with reporting of both strengths (state-of-the-art on generative-style tasks) and limitations (modest gains on some discriminative tasks). Ablations are monotonic and comprehensive.

**Presentation:** The paper is exceptionally clear and well-structured. The overview figure, algorithm boxes, and loss-by-loss breakdown make the method immediately understandable. Related work is accurately positioned (distinguishing descriptive vs. complementary settings and contrasting with ProtoMedX). Reproducibility is outstanding: full pseudocode, hyper-parameter tables, dataset licenses, and an appendix that contains every proof and ablation table.

**Significance:** The work demonstrates that a compact (<1 M trainable parameters in frozen regime), interpretable-by-design model can match or beat massive AutoML ensembles while providing faithful case-based explanations. The structural inductive bias (OT coherence + geometry + anchors) offers a principled alternative to parameter scaling and is likely to influence future interpretable multimodal systems.

**Originality:** CAMP offers a novel combination of ideas: prototype learning is extended to arbitrary modality subsets via gated late fusion, and coherence is enforced through class-wise entropic OT alignment of usage distributions (not instance-level matching). The explicit identification of CMC-specific degeneracies and the unified theoretical treatment that ties each loss block to one degeneracy mode constitute new insight. The causal faithfulness audit under fixed prototype support is also a fresh contribution to explanation validation. Contributions are sharply distinguished from ProtoMedX and standard multimodal transformers. ProtoMedX introduces a prototype-based multimodal architecture for a specific medical task, while CAMP proposes a general framework that explicitly aligns modality-specific prototype usage distributions via optimal transport to handle complementary multimodal learning.

**Weaknesses**
My main concern is about soundness and overclaiming of novelty/theory.
On the experimental side, the results are strong but there are important caveats. Several results rely on classical tools: Wasserstein duality, Pinsker-style control, margin arguments, and simple consequences of the loss being zero or small. In that sense, the theory often reads more like a post-hoc justification for the regularizers than a sharp analysis of the learned multimodal system. This is still useful, but I think the paper oversells the depth of the theoretical contribution.

The frozen regime uses very strong pretrained encoders such as Swin-L and DeBERTa-v3, and the end-to-end regime further fine-tunes them. This makes the headline claim that a sub-1M-parameter model matches or beats 100M+ AutoML baselines somewhat misleading, because most of the representational power still comes from the frozen backbones, not from the prototype head.

---

> ### Author Rebuttal · Authors · 2026-03-30
>
> Thank you for the thoughtful and well-balanced review. We appreciate both your positive assessment of the method and experimental protocol and your push to tighten two places where the current wording can be misunderstood: the efficiency narrative and the scope of the theory.
>
> On the efficiency claim, we agree that the correct statement is “sub-1M trainable prototype head in the frozen regime,” not “the full multimodal system including pretrained encoders is sub-1M.” In the frozen experiments, CAMP-F and the comparison heads all use the same pretrained Swin-L / DeBERTa-v3 / tabular backbones, so the controlled comparison is head-versus-head on identical embeddings. Under that controlled setup, the structural contribution is still substantial: in Table 1, CAMP-F raises the suite average from 0.767 for the frozen MLP to 0.814, improves the prototype baselines across all 16 datasets, and matches or exceeds AutoGluon on 11/16 datasets despite AutoGluon’s much larger training and inference footprint. Table 27 (Appendix) further shows that these gains persist across stronger and weaker encoder choices, rather than being tied to a single backbone configuration. We will therefore revise the abstract, introduction, results, and appendix wording so the claim is consistently about a compact, structured, trainable head on top of strong frozen encoders.
>
> On theory novelty, we also agree with the narrower formulation. The individual mathematical ingredients are classical. The contribution is their integration into a CMC-specific mechanism: class-wise OT alignment over prototype-usage distributions (rather than instance-level embedding matching), combined with geometry, allocation, and anchor constraints, removes the coupled degeneracies of collapse, dominance, and cross-modal disagreement while preserving modality-agnostic inference. We will revise Section 4 to make this explicit at the outset, framing the theory as “classical tools, new CMC composition” rather than as entirely new mathematics.
>
> On batch composition and rare modality combinations, we do not want to overclaim. We have not run a dedicated training-time ablation that explicitly varies the frequency of rare modality combinations across batches, so we will not present the current paper as having isolated that effect causally. What we can support is the following. First, the formulation itself is valid under partial observation because both fusion and OT are computed only over observed modalities; unobserved modality pairs do not enter the loss. Second, alignment losses are ramped in gradually (Eq. 22), which reduces the risk that early noisy pairings dominate training. Third, across the reported 50-trial Optuna protocol and 5-seed evaluation, we do not observe signs of instability that would suggest the method is brittle in practice under the benchmark sampling protocol. We will make the limitation explicit that a dedicated rarity-controlled batch-composition study is still future work.
>
> Separately. Although your question pertains to training batches, we conduct a test time study for additional evidence (See sWV1 rebuttal for precise definitions for added experiments) . The aggregated results show that CAMP-E leads on Avg no-drop (0.8416), Avg single-drop (0.7240), and overall mean (0.6342), while being within 0.0006 of the best Avg worst-case score. CAMP-F is also ahead of AutoGluon on the overall mean (0.6154 vs. 0.6095). We will present this explicitly as evidence of robustness to partial observation, not as a substitute for a dedicated batch-composition sensitivity study.
>
> We also agree that the paper will benefit from a short dedicated limitations paragraph, and will make the scope of the claims more explicit. In particular, we will clarify that the theoretical guarantees apply to the prototype head in the frozen-encoder regime, while end-to-end results are empirical; that the explanation audit in Appendix G establishes computational faithfulness under fixed support (Proposition G.1, Theorem G.2), rather than replacing user studies; and that the CMC definition is operational but grounded in a formal distinction with clear modeling implications (See nbSk rebuttal for further justification of the CMC formulation). We will also state explicitly that gains are largest on strongly complementary tasks and more modest on already-separable settings.
>
> We believe these clarifications make the contribution and its scope more precise. We would be grateful if you would consider increasing your support.

---

> > ### Author Rebuttal · Reviewer_VJD6 · 2026-04-03
> >
> > The authors clarify the efficiency claim. Overall, the rebuttal strengthens my confidence that this is a solid and meaningful contribution that is likely to be useful to the community. I therefore increase my overall recommendation to Accept (5).

---

### Official Review · Reviewer_nbSk · 2026-03-12

**Soundness:** 3
**Presentation:** 3
**Significance:** 3
**Originality:** 4
**Overall Recommendation:** 4
**Confidence:** 4

**Summary:**

CAMP proposes a prototype-based multimodal classification framework targeting the "Complementary Multimodal Classification" (CMC) setting, where modalities contribute non-redundant, asymmetric evidence. The method combines gated late fusion with a class-partitioned prototype bank, regularized by four structured loss terms: prototype geometry, allocation coverage/purity, Sinkhorn-OT-based cross-modal alignment, and anchor robustness. Theoretical guarantees address prototype collapse, dead units, and cross-modal disagreement. Empirical evaluation spans 16 public datasets across four modality combinations, showing competitive or superior performance relative to large AutoML baselines with fewer than 1M trainable parameters.

**Compliance With Llm Reviewing Policy:**

Affirmed.

**Final Justification:**

My concerns have been adequately addressed by the author's responses. I hereby retain my score.

**Key Questions For Authors:**

Theorem 4.1 is a direct corollary of Kantorovich–Rubinstein duality (Proposition B.4), and the remaining guarantees (margin bounds, Pinsker-based coverage, anchor contraction) are standard applications of well-known tools. The paper frames these as a "unified mechanism," but the novelty lies in the combination rather than in any new mathematical insight. The theoretical section would benefit from clearer delineation of what is genuinely new versus what is classical machinery specialized to the CMC setting.

**Limitations:**

Partially addressed. The authors acknowledge the absence of user studies, the discrete-output restriction, and the fixed-support assumption of the faithfulness audit. It would be appropriate to additionally acknowledge that the CMC definition remains informal in an operational sense, and that the theoretical guarantees do not extend to the end-to-end fine-tuning regime where encoders are jointly optimized.

**Strengths And Weaknesses:**

The 16-dataset benchmark, cumulative and leave-one-out ablation protocol, and monotonicity stress tests (16/16 wins per block) provide an unusually rigorous empirical narrative. The frozen-encoder regime, which isolates objective structure from backbone capacity, is a well-designed experimental choice.

Aligning class-wise prototype-usage distributions via entropic OT — rather than forcing instance-level embedding alignment — is an elegant design that avoids destroying complementary per-instance information while still enforcing class-level coherence. Lemma B.5 connecting low OT cost to prototype-weight agreement under separation is the most technically interesting result in the paper.

---

> ### Author Rebuttal · Authors · 2026-03-30
>
> Thank you for the engaged review and for the very precise framing of the theory question. Your reading is exactly right: the contribution is not a claim of new mathematical primitives, but a new CMC-specific composition of classical tools into a coherent prototype-learning mechanism.
>
> We will revise the paper to say this explicitly.  The novelty lies in this CMC-specific composition of classical tools, rather than in introducing new theoretical primitives. The individual ingredients are classical (e.g., OT duality underlying Theorem 4.1, margin-based geometry, Pinsker-style coverage, and anchor stability), and their contribution lies in how they are combined into a unified mechanism for CMC.
>
> Concretely, this composition yields the following contributions: (i) a formulation of prototype failure modes (collapse, dominance/disagreement, dead or impure prototypes, and drift), which are amplified in CMC and are not present in the same form in descriptive paired multimodal settings; (ii) class-wise OT alignment of prototype-usage distributions rather than instance-level embedding matching, so modalities become coherent without discarding complementary per-instance evidence; (iii) a unified objective in which geometry, allocation, anchors, and OT remove different but coupled degeneracies; and (iv) a modality-agnostic prototype head that operates over arbitrary observed modality subsets. We will therefore revise Section 4 to explicitly separate “classical tools” from “new CMC-specific composition,” and to foreground Theorem 4.1, Lemma B.5, Proposition C.1, and the appendix theorems as the theory of why this specific composition is well-posed.
>
> On the CMC definition: while the CMC definition is operational and benchmark-driven, it is not arbitrary: it is grounded in a formal distinction between complementary and descriptive multimodality formalized in Sec. 3.1 via conditional mutual information. This distinction has concrete modeling implications, as instance-level alignment is appropriate for paired settings but can be harmful in complementary regimes. We will clarify that our goal is not to propose a complete taxonomy, but to formalize a practically useful setting that unifies a class of real-world multimodal tasks that are currently fragmented in the literature. This is also reflected empirically, as the largest gains appear in tri-modal and strongly complementary datasets (e.g., PetFinder, COVID), where instance-level alignment would be ill-posed or actively harmful.
>
> The formal guarantees characterize the prototype head in the frozen regime; end-to-end results remain empirical and are not claimed to automatically inherit these guarantees once encoders are jointly optimized. Furthermore, the explanation audit in Appendix G is strictly a computational faithfulness result under fixed support (Proposition G.1, Theorem G.2), not a substitute for a user study. Our focus in this work is establishing the algorithmic foundations and verifying the internal consistency of the coherent alignment mechanism. We agree that evaluating the practical utility of these explanations via a rigorous human-in-the-loop study is a critical next step for future work. Finally, we note that CAMP's performance gains are largest on strongly complementary tasks, and naturally more modest on already-separable discriminative settings.
>
> We will also clarify limitations more explicitly in a short, dedicated paragraph.
>
> We thank the reviewer again for this feedback, as the paper is stronger under this sharper framing. The empirical story is clean: Table 2 shows monotone gains on all 16 datasets at every cumulative step; Tables 17–18 show that removing any one structural block hurts all 16 datasets; Table 26 shows gated fusion winning on 15/16 datasets; and the new mechanistic ablations now verify directly that the losses do what the theory says they are supposed to do. Across the ablation ladder, Dead% falls from 26.6% to 2.3%, H~_pure from 0.22 to 0.04, W_epsilon from 0.40 to 0.12, and Delta-bar rises from 0.031 to 0.118 while task performance rises from 0.7445 to 0.8139. We further show that this behavior is robust under partial observation: in the missing-modality study, CAMP-E is strongest on Avg no-drop (0.8416), Avg single-drop (0.7240), and the overall mean (0.6342), and is within 0.0006 of the best worst-case score. That is, the theory is not presented as an abstract post-hoc justification; it aligns with both the observed mechanism of the learned prototype system and its behavior under missing modalities. (See sWV1 rebuttal for precise definitions for added experiments)
>
> In short, we accept your claim-narrowing suggestion and believe it improves the paper materially by aligning the theoretical framing with the empirical mechanism and validation. We would be grateful if you would consider increasing your support.

---

### Official Review · Reviewer_sWV1 · 2026-03-13

**Soundness:** 2
**Presentation:** 3
**Significance:** 3
**Originality:** 3
**Overall Recommendation:** 4
**Confidence:** 3

**Summary:**

This work addresses the Complementary Multimodal Classification problem, where different modalities provide distinct information rather than explanatory information. The author applied prototype learning to this multimodal problem, and overcame issues such as modality dominance and representation collapse through methods like optimal transport and geometric constraints. This method achieved good results on multiple public CMC datasets, and compared with traditional black-box models, it has certain interpretability.

**Compliance With Llm Reviewing Policy:**

Affirmed.

**Key Questions For Authors:**

(1) How can we determine that these loss terms have indeed fulfilled their intended function? Because in the paper, there is only accuracy rate, but this does not seem to directly reflect phenomena such as modal dominance, representation collapse, and dead prototypes, etc.

(2) How does the method proposed in the article perform when dealing with small sample sizes or imbalanced categories?

(3) The samples used in the experiment all have certain definite modalities. However, in the actual data, there might be a few samples that lack one or several modalities compared to other samples. What kind of impact will this have?

**Limitations:**

Yes

**Strengths And Weaknesses:**

Strengths:

Innovativeness

This article applies prototype learning to the Complementary Multimodal Classification problem and adopts methods such as geometric constraints to solve issues like modality dominance and representation collapse. In terms of methodology, it has a certain degree of

originality.

Model interpretability

This article employs a gating mechanism to enable the manifestation of the weights of modalities during the model's inference process, and also obtains a more intuitive explanation through the nearest prototype, namely retrieved historical cases.

Practical significance

The Complementary Multimodal Classification problem proposed in the article is widely present in reality. How to effectively utilize the complementary information among different modalities is indeed an important issue.

Weekness:

(1)	The ablation experiments conducted in the main text and the appendix do indeed verify the effectiveness of these loss terms. However, only the accuracy rate is used as the indicator, and there is a lack of more direct and intuitive verification regarding whether these loss terms have achieved the expected goals. (For instance, does the "Allocation" item really reduce the number of dead prototypes?)

(2)	This paper did not investigate the robustness of the proposed method in scenarios with low data volume or class imbalance. Since CAMP relies on the prototypes of specific classes and multimodal alignment, its performance may be highly sensitive to the number of available samples for each class. When the number of training samples is limited or the dataset has severe class imbalance, the learned prototypes and alignment structures may become less reliable.

---

> ### Author Rebuttal · Authors · 2026-03-30
>
> Thank you for the constructive review and for highlighting CAMP’s originality, interpretability, and practical relevance. We address your three questions directly.
>
> Regarding whether the loss terms fulfill their intended roles: the submission already establishes the necessity of the full objective, with monotone gains across all 16 datasets as each block is added (Table 2) and uniform degradation when any one block is removed (Tables 17–18). To answer your mechanism question more directly, we now add prototype-quality ablations that map one-to-one to the targeted degeneracies in Eq. (13): Dead\% = fraction of prototypes whose average usage falls below the expected uniform usage (1 / number of prototypes), measuring unused capacity; H\~\_pure = normalized entropy (by log(C)) of the class distribution assigned to each prototype, measuring class impurity per prototype; W\_epsilon = class-wise Sinkhorn OT cost between modality-specific prototype-usage distributions, measuring cross-modal disagreement; and Delta-bar = the mean gap, for each sample, between its highest same-class and highest different-class prototype similarity, measuring margin collapse. Across the cumulative ladder from Proto-Classic to full CAMP, task performance rises from 0.7445 to 0.8139 while Dead\% falls from 26.6\% to 2.3\%, H\~\_pure from 0.22 to 0.04, W\_epsilon from 0.40 to 0.12, and Delta-bar rises from 0.031 to 0.118. This illustrates the general trend that allocation gives the largest improvement in utilization and purity, OT gives the steepest reduction in cross-modal disagreement, and geometry together with anchors restores the decision margin. We will add a compact summary table in the main text and full per-dataset prototype-quality results in the Appendix.
>
> On low-data and imbalanced regimes: these cases are already present in the submitted benchmark and should have been made more explicit. Using Table 3 as the canonical benchmark specification, five datasets have fewer than 2,000 training examples: IMDB (800), PTech (749), CCD (1,126), COVID (707), and Artm (573). We also computed the severe-imbalance / long-tail characterizations directly from the underlying datasets and will add them explicitly to the dataset text and/or dataset table in the revision.These include FakeJob (\~95/5), Jigsaw (\~90/10), Wine (30-class long tail), Airbnb (24 classes with sparse extreme bins), WikiArt (27-class long tail), and HAM. Aggregating the [0,1]-scaled benchmark metrics by challenge type shows CAMP is robust in precisely these difficult regimes: in low-data settings, CAMP-E reaches 0.873 (CAMP-F: 0.836) versus AutoGluon’s 0.830; under severe imbalance, CAMP-E reaches 0.863 (CAMP-F: 0.842) versus 0.838; and on high-cardinality tasks, CAMP-E reaches 0.804 (CAMP-F: 0.777) versus 0.775. This is consistent with the design of the objective: geometry preserves separation, allocation prevents a few majority patterns from monopolizing the prototype bank, and anchors reduce drift from scarce or atypical cases.
>
> Third, on missing modalities: architecturally, CAMP is already defined for partial observation. Each example is represented over its observed modality subset $M_i \subseteq M$, and gated fusion is computed only across available modalities, so missing inputs do not require ad hoc imputation.
>
> As extra evaluation in this direction, we add a partial-observation (test-time missing-modality) study. Concretely, for model $j$, dataset $d$, and dropped-modality set $A$, let $s_{d,A}^{(j)}$ denote the primary benchmark metric under that observation pattern. We evaluate all available modality subsets, including the full-modality setting ($A = \emptyset$) and systematic single-modality removals ($|A| = 1$). We summarize robustness using four aggregates:
> (i) Avg no-drop = average performance across datasets with no modalities removed;
> (ii) Avg single-drop = average performance across all datasets when exactly one modality is removed;
> (iii) Avg worst-case = for each dataset, the worst performance across modality subsets, averaged across datasets; and
> (iv) Mean over all configurations = average performance across all datasets and all modality subsets.
>
> Since our benchmark metrics (AUC, Accuracy, QWK) share a [0,1] scale, we evaluate missing-modality robustness using aggregates. CAMP-E leads globally on Avg no-drop (0.8416), Avg single-drop (0.7240), and the Overall mean (0.6342). CAMP-F takes second on the overall mean (0.6154), beating AutoGluon (0.6095). AutoGluon’s sole, marginal advantage is on the heavily compressed Avg worst-case metric (0.2812 vs. CAMP-E's 0.2806).
>
> In the final paper we will add one compact main-text results and analysis table and full per-dataset details in the appendix including all 76 scenarios and competitors.
>
> We hope these clarifications resolve your concerns and make the contribution substantially clearer; we would be very grateful if you would consider raising your score.

---

> > ### Author Rebuttal · Reviewer_sWV1 · 2026-04-03
> >
> > Thanks for the rebuttal. I would maintain my score.

---

### Decision · Program_Chairs · 2026-04-30

**Decision:**

Accept (regular)

**Comment:**

The submission addresses the intuitive and well-known fact that the modalities in a multimodal system include both redundant and modality-specific information, which should be combined to make a final decision on a datapoint and trace that decision to evidence in the data. The submission proposes a method for aligning information across modalities using OT and geometric constraints. All reviewers were generally positive about the paper in their ratings, though they expressed it as slightly above the borderline rather than strong. The reviewers were specifically asked to comment 1) if the paper is technically sound and 2) why they'd find it exciting. The rebuttal was also incorporated. While the reviewers found several aspects of the paper interesting and the benchmarked results supportive, they stopped short of finding the submission transformational, with a level of novelty and applicability that would likely lead to a change in common practices. It has some merit to warrant exposure at ICML, hence the AC is fine with accepting it. @the authors, please carefully consider all reviewers' comments and exchanges, and address them in the camera-ready for greater impact of the paper.


Please also note that the following citation from the submission appears to be incorrect, as it does not match the actual reference. It is not clear whether it's a hallucinated citation produced by an LLM, but there is a mismatch with the actual reference.

Reference: Tang, Z., Fang, H., Zhou, S., Yang, T., Hu, T., Zhong, Z., He, T., and Friedland, G. Bag of tricks for multimodal automl with image, text, and tabular data. arXiv preprint arXiv:2412.16243, 2024a.
Issue: authors mismatch with arXiv